# DIGRAC: Digraph Clustering Based on Flow Imbalance

**Yixuan He**
Department of Statistics
University of Oxford
yixuan.he@stats.ox.ac.uk

**Gesine Reinert**
Department of Statistics
University of Oxford &
The Alan Turing Institute, London, UK
reinert@stats.ox.ac.uk

**Mihai Cucuringu**
Department of Statistics and Mathematical Institute
University of Oxford &
The Alan Turing Institute, London, UK
mihai.cucuringu@stats.ox.ac.uk

## Abstract

Node clustering is a powerful tool in the analysis of networks. We introduce a graph neural network framework, named DIGRAC, to obtain node embeddings for directed networks in a self-supervised manner, including a novel probabilistic imbalance loss, which can be used for network clustering. Here, we propose *directed flow imbalance* measures, which are tightly related to directionality, to reveal clusters in the network even when there is no density difference between clusters. In contrast to standard approaches in the literature, in this paper, directionality is not treated as a nuisance, but rather contains the main signal. DIGRAC optimizes directed flow imbalance for clustering without requiring label supervision, unlike existing graph neural network methods, and can naturally incorporate node features, unlike existing spectral methods. Extensive experimental results on synthetic data, in the form of directed stochastic block models, and real-world data at different scales, demonstrate that our method, based on flow imbalance, attains state-of-the-art results on directed graph clustering when compared against 10 state-of-the-art methods from the literature, for a wide range of noise and sparsity levels, graph structures, and topologies, and even outperforms supervised methods.

## 1 Introduction

Revealing an underlying community structure of *directed* networks (*digraphs*) is an important problem in many applications, see for example [1] and [2], such as detecting influential social groups [3] and analyzing migration patterns [4]. While most existing methods that could be applied to directed clustering use local edge densities as main signal and directionality (i.e, edge orientation) as additional signal, we argue that even in the absence of any edge density differences, directionality can play a vital role in directed clustering as it can reveal latent properties of network flows. The underlying intuition is that homogeneous clusters of nodes form *meta-nodes* in a *meta-graph*, with the meta-graph directing the flow between clusters; directed core-periphery structure is such an example [5]. Loosely speaking, a meta-node is a collection of nodes, and a meta-graph is a graph on such meta-nodes, with weighted edges collecting the overall sum of edge weights between the meta-nodes. Fig. 1(a) is an example of flow imbalance between two clusters, here on an unweighted network for simplicity: while 80% of the edges flow from the *Transient* cluster to the *Sink* cluster, only 20% flow in the other direction. As a real-world example, Fig. 1(b) shows the strongest flow imbalances between clusters detected by our method in a network of US migration flow [4]; most edges flow from the red cluster (label 1) to the blue one (label 2). Figures 1(c-d) show examples on a synthetic meta-graph. We could also think of a social network in which a set of fake accounts $\mathcal{A}$ have been

Y. He et al., DIGRAC: Digraph Clustering Based on Flow Imbalance. *Proceedings of the First Learning on Graphs Conference (LoG 2022)*, PMLR 198, Virtual Event, December 9–12, 2022.

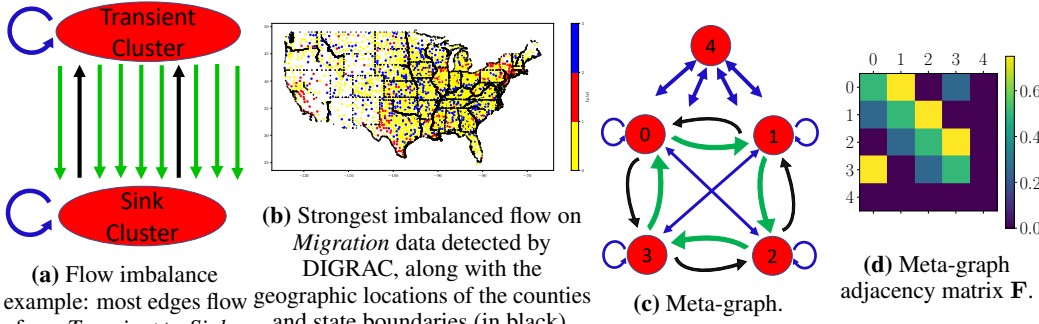

**(a)** Flow imbalance example: most edges flow from *Transient* to *Sink*.

**(b)** Strongest imbalanced flow on *Migration* data detected by DIGRAC, along with the geographic locations of the counties and state boundaries (in black).

**(c)** Meta-graph.

**(d)** Meta-graph adjacency matrix **F**.

**Figure 1:** Visualization of cut flow imbalance and meta-graph: (a) 80% of edges flow from Transient to Sink, while 20% of edges flow in the opposite direction; (b) top pair imbalanced flow on *Migration* data [4]: most edges flow from red (1) to blue (2); (c) & (d) are for a Directed Stochastic Block Model with a cycle meta-graph with ambient nodes, for a total of 5 clusters. Most edges flow in direction $0 \to 1 \to 2 \to 3 \to 0$, while few flow in the opposite direction. Cluster 4 is the ambient cluster. In (a) and (c), blue lines indicate flows with random, equally likely directions; these flows do not exist in the meta-graph adjacency matrix **F**. For (d), the lighter the color, the stronger the flow.

created, and these target another subset $\mathcal{B}$ of real accounts by sending them messages. Most likely, there would be many more messages from $\mathcal{A}$ to $\mathcal{B}$ than from $\mathcal{B}$ to $\mathcal{A}$, hinting that $\mathcal{A}$ is most likely comprised of fake accounts.

Thus, instead of finding relatively dense groups of nodes in digraphs with a relatively small amount of flow between the groups, as in [6–11], our main goal is to recover clusters with *strongly imbalanced flow* among them, in the spirit of [12, 13], where directionality is the main signal. This task is not addressed by most methods for node clustering in digraphs, including community detection methods. Those methods that do lay emphasis on directionality are usually spectral methods, for which incorporating features is non-trivial, or graph neural network (GNN) methods that require labeling information. An exception is the network community detection method InfoMap [14] which uses directed random walks; however, it still relies on some edge density information within clusters, as a walk is more likely to happen when the density is higher. [15] and [16] employ Markov chains, but we pick InfoMap as a representative for methods based on information theory/Markov chains.

Here we introduce DIGRAC, a GNN framework to obtain node embeddings for clustering digraphs (allowing weighted edges and self-loops but no multiple edges). In a self-supervised manner, a novel *probabilistic imbalance loss* is proposed to act on the digraph induced by all training nodes. The global imbalance score, one minus whom is the self-supervised loss function, is aggregated from pairwise normalized cut imbalances. The method is end-to-end in combining embedding generation and clustering without an intermediate step. To the best of our knowledge, this is the first GNN method which derives node embeddings for digraphs that directly maximizes flow imbalance between pairs of clusters. With an emphasis on the use of a direction-based flow imbalance objective, experimental results on synthetic data and real-world data at different scales demonstrate that our method can achieve leading performance for a wide range of network densities and topologies.

DIGRAC's main novelty is the ability to cluster based on direction-based flow imbalance, instead of using classical criteria such as maximizing relative densities within clusters. Compared with prior methods that focus on directionality, DIGRAC can easily consider node features and also does not require known node clustering labels. DIGRAC complements existing approaches in various aspects: (1) Our results show that DIGRAC complements classical community detection by detecting alternative patterns in the data, such as meta-graph structures, which are otherwise not detectable by existing methods. This aspect of detecting novel structures in directed graphs has also been emphasized in [12]. (2) DIGRAC complements existing spectral methods, through the possibility of including exogenous information, in the form of node-level features or labels, thus borrowing their strength. (3) DIGRAC complements existing GNN methods by introducing an imbalance-based objective. (4) DIGRAC introduces imbalance measures for evaluation when ground-truth is unavailable.

DIGRAC's applicability extends beyond settings where the input data is a digraph: with time series data as input, the digraph construction mechanism can accommodate any procedure that encodes a pairwise directional association between the corresponding time series, such as lead-lag relationships and Granger causality [17], with applications such as in the analysis of information flow in brain networks [18], biology [19], finance [20, 21] and earth sciences [22]. DIGRAC could also facilitate tasks in ranking [23] and anomaly detection [24–26], as it allows one to extrapolate from *local* pairwise (directed) interactions to a *global* structure inference, in the high-dimensional low signal-to-noise ratio regime.

**Main contributions.** Our main contributions are as follows.

1. We propose a GNN framework for self-supervised end-to-end node clustering on (possibly attributed and weighted) digraphs explicitly taking into account the directed flow imbalance.

2. We propose a family of probabilistic global imbalance scores to serve as the self-supervised loss function and evaluation objective, including one based on hypothesis testing for directionality signal. To the best of our knowledge, this is the first method directly maximizing flow imbalance for node clustering in digraphs using GNNs.

3. We extend our method to the semi-supervised setting when label information is available.

## 2   Related Work

Directed clustering has been explored by non-GNN methods. [27] performs directed clustering that hinges on symmetrizations of the adjacency matrix, but is not scalable as it requires large matrix multiplications. [28] proposes a spectral co-clustering algorithm for asymmetry discovery that relies on in-degree and out-degree. Whenever direction is the sole information, such as in a complete network with a lead-lag structure derived from time series [20], a purely degree-based method cannot detect the clusters. While [29] produces two partitions of the node set, one based on out-degree and one based on in-degree, our partition simultaneously takes both directions into account. The *directed graph Laplacians* introduced by [2] are only applicable to strongly connected digraphs, which is rarely the case in sparse networks arising in applications. InfoMap by [14] assumes that there is a "map" underlying the network, similar to a meta-graph in DIGRAC. InfoMap aims to minimize the expected description length of a random walk and is recommended for networks where edges encode patterns of movement among nodes. While related to DIGRAC, InfoMap still relies on some amount of density-based signal being present within each of the modules. [12] seeks to uncover clusters characterized by a strongly imbalanced flow circulating among them, based on eigenvectors of the Hermitian matrix $(\mathbf{A} - \mathbf{A}^T) \cdot i$, where $\mathbf{A}$ is the (normalized) adjacency matrix and $i$ the imaginary unit. [12] is a purely spectral-based method and is not able to naturally incorporate any available node features or label information; in contrast, DIGRAC is a GNN-based method that is naturally able to account for such information. Moreover, [12] is not driven by an optimization function, but only proposes evaluation metrics that capture the imbalance of the pairs of clusters. In contrast, inspired by [12], in DIGRAC a family of novel imbalance loss functions is proposed, with a probabilistic interpretation, rendering DIGRAC a fully trainable end-to-end pipeline. Furthermore, the rich class of imbalance evaluation and training objectives/losses proposed in this paper go far beyond the evaluation metrics considered in [12]. [13] uncovers higher-order structural information among clusters in digraphs, while maximizing the imbalance of the edge directions, but its definition of the flow ratio restricts the underlying meta-graph to a path.

GNNs have been applied to digraph node classification, which is similar to digraph clustering but requires known clustering labels. [30] uses first and second-order proximity, and constructs three Laplacians, but the method is space and speed-inefficient. [31] simplifies [30], builds a directed Laplacian based on PageRank, and aggregates information dependent on higher-order proximity. Building on [12, 32], [33] constructs a Hermitian matrix that encodes undirected geometric structure in the magnitude of its entries, and directional information in their phase. [34] introduces a digraph data augmentation method called *Laplacian perturbation* and conducts digraph contrastive learning. [35] proposes a spectral-based graph convolution network for digraphs, yet is restricted to strongly connected digraphs that are usually not realistic. [36] utilizes convolution-like anisotropic filters based on local subgraph structures (motifs) for semi-supervised node classification tasks in digraphs, but relies on pre-defined structures and fails to handle complex networks.

In particular, [30, 31, 33, 34, 36] all require known labels, which are not generally available for real-world data. [2, 12, 13, 27, 28] could not trivially incorporate node attributes or node labels. In contrast, we propose an efficient GNN-based method that maximizes a probablistic flow imbalance objective, in a self-supervised manner, and which can naturally analyze attributed weighted digraphs.

To avoid potential misunderstanding, we briefly mention several related works that we are aware of but do not compare against in our experiments in the main text. While DIGRAC addresses the task of partitioning the nodes into disjoint sets, [37] locates a certain community within a network. In particular, [37] proposes a local algorithm while this paper proposes a global one. OSLOM by [38] is very flexible but based on a density heuristic and hence a comparison to DIGRAC on networks without density signal would not be fair, to begin with. [39] introduces directionality in the Louvain algorithm. This algorithm optimizes a modularity-type function that compares the number of edges within communities to the expected number of edges under a specified model. It is thus an approach that aims to find denser-than-expected groups of vertices. When all groups have the same density, as in our synthetic data sets, and the only structure lies in the directionality of the edges, this method simply cannot be expected to perform well. The Leiden algorithm in [40] also builds on the Louvain method, again optimizing a modularity-type function that compares the number of edges within communities to the expected number of edges under a specified model. It is a powerful method for that task, but cannot be fairly compared to DIGRAC which is tailored to find imbalances. As confirmed by our experiments in Appendix (App.) E, comparing these methods to DIGRAC is not appropriate.

We also do not compare DIGRAC against graph pooling methods [41], which are inspired by pooling in CNNs and developed to discard information that is superfluous for the task at hand, as a partition of the nodes which can be interpreted as clustering is only a byproduct. Moreover, graph pooling methods are usually developed only for undirected networks. While graph matching as in [42–44] and [45] can be viewed as a clustering method of networks, matching the graph of interest to a disconnected graph by connecting each node in the observed graph with an isolated node of the disconnected graph, this approach is not developed for directed networks. The underlying idea of these papers is complementary to the meta-graph idea which underpins DIGRAC; in the meta-graph, the components are connected, and estimating the directionality of these connections is the main focus. Hence this work addresses a very different task. We emphasize that these are all excellent methods, but they address different objectives and tasks. DIGRAC is tailored to detect an imbalance signal in directed networks, and such a signal cannot be present in an undirected network. As it is based on imbalance, DIGRAC will not be able to detect a signal in an undirected network, thus rendering it not applicable to undirected networks.

## 3  The DIGRAC Framework

**Problem definition.** Denote a (possibly weighted) digraph with node attributes as $\mathcal{G} = (\mathcal{V}, \mathcal{E}, w, \mathbf{X})$, with $\mathcal{V}$ the set of nodes, $\mathcal{E}$ the set of directed edges or links, and $w \in [0, \infty)^{|\mathcal{E}|}$ the set of edge weights. $\mathcal{G}$ may have self-loops, but no multiple edges. The number of nodes is $n = |\mathcal{V}|$, and $\mathbf{X} \in \mathbb{R}^{n \times d_{\text{in}}}$ is a matrix whose rows encode the nodes' attributes. Such a network can be represented by the attribute matrix $\mathbf{X}$ and the adjacency matrix $\mathbf{A} = (A_{ij})_{i,j \in \mathcal{V}}$, with $\mathbf{A}_{ij} = 0$ if no edge exists from $v_i$ to $v_j$; if there is an edge $e$ from $v_i$ to $v_j$, we set $A_{ij} = w_e$, the edge weight.

Digraphs often lend themselves to interpreting weighted directed edges as flows, with a meta-graph on clusters of vertices describing the overall flow directions; see Fig. 1. A clustering is a partition of the set of nodes into $K$ disjoint sets (clusters) $\mathcal{V} = \mathcal{C}_0 \cup \mathcal{C}_1 \cup \cdots \cup \mathcal{C}_{K-1}$ (ideally, $K \geq 2$). Intuitively, nodes within a cluster should be similar to each other with respect to flow directions, while nodes across clusters should be dissimilar. In a self-supervised setting, only the number of clusters $K$ is given. In a semi-supervised setting, for each of the $K$ clusters, a fraction set $\mathcal{V}^{\text{seed}} \subseteq \mathcal{V}^{\text{train}} \subset \mathcal{V}$ of the set $\mathcal{V}^{\text{train}}$ of all training nodes is selected to serve as the set of seed nodes, for which the cluster membership labels are known before training. The goal of semi-supervised clustering is to assign each node $v \in \mathcal{V}$ to a cluster containing some known seed nodes, without knowledge of the underlying flow meta-graph. The corresponding self-supervised clustering task does not use seed nodes.

### 3.1 Self-Supervised Loss for Clustering

Our self-supervised loss function is inspired by [12], aiming to cluster the nodes by maximizing a normalized form of cut imbalance across clusters. We first define probabilistic versions of cuts, imbalance flows, and probabilistic volumes. For $K$ clusters, the *assignment probability matrix* $\mathbf{P} \in \mathbb{R}^{n \times K}$ has as row $i$ the probability vector $\mathbf{P}_{(\mathbf{i},:)} \in \mathbb{R}^K$ with entries denoting the probabilities of each node to belong to each cluster; its $k^{\text{th}}$ column is denoted by $\mathbf{P}_{(:,k)}$.

● $\forall k, l \in \{0, \ldots, K-1\}$ where $K \geq 2$, the **probabilistic cut** from cluster $\mathcal{C}_k$ to $\mathcal{C}_l$ is defined as

$$W(\mathcal{C}_k, \mathcal{C}_l) = \sum_{i,j} \mathbf{A}_{i,j} \cdot \mathbf{P}_{i,k} \cdot \mathbf{P}_{j,l} = (\mathbf{P}_{(:,k)})^T \mathbf{A} \mathbf{P}_{(:,l)}.$$

● The **imbalance flow** between $\mathcal{C}_k$ and $\mathcal{C}_l$ is defined as $|W(\mathcal{C}_k, \mathcal{C}_l) - W(\mathcal{C}_l, \mathcal{C}_k)|$.

For interpretability and ease of comparison, we normalize the imbalance flows to obtain an imbalance score with values in $[0, 1]$ as follows (we defer additional details to App. B.2).

● The **probabilistic volume** for cluster $\mathcal{C}_k$ is defined as

$$VOL(\mathcal{C}_k) = VOL^{(\text{out})}(\mathcal{C}_k) + VOL^{(\text{in})}(\mathcal{C}_k)$$
$$= \sum_{i,j} (\mathbf{A}_{j,i} + \mathbf{A}_{i,j}) \cdot \mathbf{P}_{j,k}$$

Then $VOL(\mathcal{C}_k) \geq W(\mathcal{C}_k, \mathcal{C}_l)$ for all $l = 1, \ldots, K-1$ and

$$\min(VOL(\mathcal{C}_k), VOL(\mathcal{C}_l)) \geq |W(\mathcal{C}_k, \mathcal{C}_l) - W(\mathcal{C}_l, \mathcal{C}_k)|. \tag{1}$$

The imbalance term, which is used in most of our experiments, denoted $\text{CI}^{\text{vol\_sum}}$, is defined as

$$\text{CI}^{\text{vol\_sum}}(k, l) = 2 \frac{|W(\mathcal{C}_k, \mathcal{C}_l) - W(\mathcal{C}_l, \mathcal{C}_k)|}{VOL(\mathcal{C}_k) + VOL(\mathcal{C}_l)} \in [0, 1]. \tag{2}$$

In particular, for $K = n$, every node is a single cluster, and $\text{CI}^{\text{vol\_sum}}(k, l) = 1$, but then the partition is not informative. The aim is to find a partition that maximizes the imbalance flow under the constraint that the partition has at least two sets, to capture groups of nodes that could be viewed as representing clusters in the meta-graph. The normalization by the volumes penalizes partitions that put most nodes into a single cluster. The range $[0, 1]$ follows from Eq. (1). Other variants are discussed in App. B.3.

To obtain a **global probabilistic imbalance score**, based on $\text{CI}^{\text{vol\_sum}}$ from Eq. (2), we average over pairwise imbalance scores of different pairs of clusters. Since the scores discussed are symmetric and the cut difference before taking absolute value is skew-symmetric, we only need to consider the pairs in the set $\mathcal{T} = \{(\mathcal{C}_k, \mathcal{C}_l) : 0 \leq k < l \leq K - 1, k, l \in \mathbb{Z}\}$.

A naive approach, which we call the **"*naive*"** variant, considers all possible $\binom{K}{2}$ pairwise cut imbalance values. However, due to potentially high noise levels in certain data sets, one may only be interested in pairs that are not just noise but exhibit true signals. To this end, we introduce a **"*std*"** variant, which only considers pairwise cut imbalance values that are 3 standard deviations away from the observed purely noisy imbalance values; the standard deviation is calculated under the null hypothesis that the between-cluster relationship has no direction preference, i.e. $\mathbf{F}_{k,l} = \mathbf{F}_{l,k}$ (entries of the meta-graph adjacency matrix $\mathbf{F}$ to be introduced later in this section), as follows.

Suppose two clusters $\mathcal{C}_k$ and $\mathcal{C}_l$ have only noisy links between them, with no edge in the meta-graph $\mathbf{F}$, i.e. $\mathbf{F}_{kl} = 0$. Assume also that the underlying network is fixed in terms of the number of nodes and locations of edges; the only randomness stems from the direction of the edges. Then we can provide the following theoretical guarantee.

**Proposition 1.** *Suppose that $\mathcal{C}_k$ and $\mathcal{C}_l$ are two clusters of $n_k$ and $n_l$ nodes, respectively, with $m(k, l)$ edges between them, edge weights $w_{ij} = w_{ji} \in [0, 1]$ and edge direction drawn independently at random with equal probability $\frac{1}{2}$ for each direction. We assume that the edge weights satisfy $\max_e |w_e|(\sum_e w_e^2)^{-\frac{1}{2}} = o(m(k, l))$. Then $W(\mathcal{C}_k, \mathcal{C}_l) - W(\mathcal{C}_l, \mathcal{C}_k)$ is approximately normally distributed with mean 0 and variance $||w||^2$ as $m(k, l) \to \infty$.*

A consequence of Proposition 1, which is proved in App. B.1, is that under its assumptions, approximately 99.7 % of the observations fall within 3 standard deviations from 0. While Proposition 1 makes many assumptions and ignores reciprocal edges, the resulting threshold is still a useful guideline for restricting attention to pairwise imbalance values which are very likely to capture a true signal. In particular, we use it as motivation for our "std" variant to pick cluster pairs from $\mathcal{T}$ that satisfy $\left(W(\mathcal{C}_k, \mathcal{C}_l) - W(\mathcal{C}_l, \mathcal{C}_k)\right)^2 > 9\left(W(\mathcal{C}_k, \mathcal{C}_l) + W(\mathcal{C}_l, \mathcal{C}_k)\right).$

As we are mainly concerned about the top pairs (i.e., those exhibiting the largest imbalance flow), another option is the **"*sort*"** variant, which selects the largest $\beta$ pairwise cut imbalance values, where $\beta$ is half of the number of nonzero entries in the off-diagonal entries of the meta-graph adjacency matrix $\mathbf{F}$, if the meta-graph is known or can be approximated. For example, for a "cycle" meta-graph with three clusters and no ambient nodes, $\beta = 3$. When the meta-graph is a "path" with three clusters and ambient nodes, then $\beta = 1$. When considering the "sort" variant, with $\mathcal{T}(\beta) = \{(\mathcal{C}_k, \mathcal{C}_l) \in \mathcal{T} : \mathrm{CI}^{\mathrm{vol\_sum}}(k, l) \text{ is among the top } \beta \text{ values}\}$, where $1 \le \beta \le \binom{K}{2}$, we set

$$\mathcal{O}_{\mathrm{vol\_sum}}^{\mathrm{sort}} = \frac{1}{\beta} \sum_{(\mathcal{C}_k, \mathcal{C}_l) \in \mathcal{T}(\beta)} \mathrm{CI}^{\mathrm{vol\_sum}}(k, l), \quad \text{and} \quad \mathcal{L}_{\mathrm{vol\_sum}}^{\mathrm{sort}} = 1 - \mathcal{O}_{\mathrm{vol\_sum}}^{\mathrm{sort}}, \tag{3}$$

as the corresponding loss function. Definitions of meta-graph structures are discussed in Section 4.1. For the other variants, the corresponding scores and loss functions are defined analogously. We apply the "std" variant when we have no prior knowledge of the meta-graph structure during training, and the "sort" variant when we have information on the number of pairs to count.

When using the "std" variant for training, for the initial 50 epochs, we apply the "sort" variant with $\beta = 3$ for a reasonable starting clustering probability matrix for training, as otherwise during the initial training epochs possibly no pairs could be picked out. During the epochs actually utilizing this "std" variant, if no pairs could be picked out, we temporarily switch to the "naive" variant for that epoch.

Regarding complexity, the objective mainly contains matrix-vector multiplications and element-wise matrix divisions, which are at most quadratic in the number of nodes, but usually faster with our sparsity-aware implementation.

## 3.2 Instantiation of DIGRAC

To instantiate DI-GRAC, any aggregation scheme able to take directionality into account could be incorporated into our general framework, as long as it can output the node embedding matrix $\mathbf{Z}$. Here, by default, we adapt the Signed Mixed Path Aggregation (SIMPA) scheme from [46]. We remove the signed parts and devise a simple yet effective directed mixed path aggregation scheme, which we call Directed Mixed Path Aggregation (DIMPA), to obtain the probability

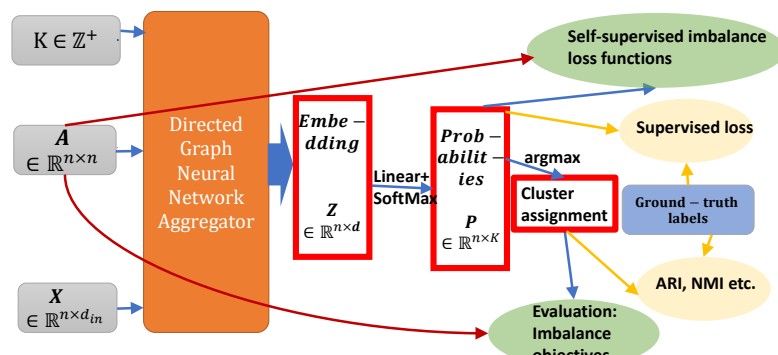

**Figure 2:** DIGRAC overview: from feature matrix $\mathbf{X}$, adjacency matrix $\mathbf{A}$ and number of clusters $K$, we first apply a directed GNN aggregator to obtain the node embedding matrix $\mathbf{Z}$, then apply a linear layer followed by a unit softmax function to get the probability matrix $\mathbf{P}$. Applying *argmax* on each row of $\mathbf{P}$ yields node cluster assignments. Green circles involve our proposed imbalance objective, while the yellow circles can only be used when ground-truth labels are provided.

assignment matrix $\mathbf{P}$ by applying a linear layer followed by a unit softmax function to the embedding

generated, and feed it to the loss function. Details of DIMPA are provided in App. A. A framework diagram is provided in Fig. 2, and an instantiation using DIMPA is visualized in Fig. 5.

## 4 Experiments

In our synthetic experiments, when by design ground truth is available, performance is assessed by the Adjusted Rand Index (ARI) [47]. Normalized Mutual Information (NMI) results give almost the same ranking for the best-performing methods as the ARI, with an average Kendall tau value of 83.8% and standard deviation 24.9%, for pairwise ranking comparison, on the methods compared in our experiments. We do not focus on NMI in the main text due to its shortcomings [48], see also App. C.9.

Clustering tasks will have different ground truths, depending on the pattern they are trying to detect. Many network clustering methods focus on detecting relatively dense clusters, and try to optimize classical network clustering measures, such as directed modularity or partition density. Ground truth for these clustering algorithms then relates to relatively densely connected subgroups in the data. DIGRAC is a novel method that addresses a novel task, namely that of detecting flow imbalances. To the best of our knowledge, real-world data sets with ground-truth flow imbalances are not available to date, and hence we introduce normalized imbalance scores to evaluate clustering performance based on flow imbalance. As ARI and NMI require ground-truth labels, they thus cannot be applied to the available real-world data sets. To address this shortcoming, for the real-world data sets, in Table 1, we include three performance measures which we introduce in the paper, and the appendix contains an additional 11 performance measures. Implementation details are provided in App. C. Codes and preprocessed data are available at `https://github.com/SherylHYX/DIGRAC_Directed_Clustering` and have been included in the open-source library *PyTorch Geometric Signed Directed* [49].

We compare DIGRAC against the most recent related methods from the literature for clustering digraphs. The **10** methods are • (1) InfoMap [14], • (2) Bibliometric and • (3) Degree-discounted introduced in [27], • (4) DI_SIM [28], • (5) Herm and • (6) Herm_sym introduced in [12], • (7) Mag-Net [33], • (8) DGCN [30], • (9) DiGCN [31], and • (10) DiGCL [34]. The abbreviations of these methods, when reported in the numerical experiments, are InfoMap, Bi_sym, DD_sym, DISG_LR, Herm, Herm_sym, MagNet, DGCN, DiGCN, DiGCL, respectively. DGCN is the least efficient method in terms of speed and space complexity, followed by DiGCN which involves the so-called *inception blocks*. We use the same hyperparameter settings stated in these papers. Methods (7), (8), (9), (10) are GNN methods which are trained with 80% nodes under label supervision, while all the other methods are trained without label supervision. DIGRAC further restricts itself to be trained on the subgraph induced by only the training nodes. All methods are designed for directed graphs, and all except Infomap require $K$ to be known. Runtime comparison is provided in App. C.2, illustrating that DIGRAC is among the fastest among competing GNNs. Implementation details for competitors are provided in App. C.7.

### 4.1 Data Sets

**Synthetic data: Directed Stochastic Block Models**  A standard directed stochastic block model (DSBM) is often used to represent a network cluster structure, see for example [1]. Its parameters are the number $K$ of clusters and the edge probabilities; given the cluster assignment of the nodes, the edge indicators are independent. The DSBMs used in our experiments also depend on a meta-graph adjacency matrix $\mathbf{F} = (\mathbf{F}_{k,l})_{k,l=0,\ldots,K-1}$ and a *filled* version of it, $\tilde{\mathbf{F}} = (\tilde{\mathbf{F}}_{k,l})_{k,l=0,\ldots,K-1}$, and on a noise level parameter $\eta \leq 0.5$. The meta-graph adjacency matrix $\mathbf{F}$ is generated from the given meta-graph structure, called $\mathcal{M}$. To include an ambient background, the filled meta-graph adjacency matrix $\tilde{\mathbf{F}}$ replaces every zero in $\mathbf{F}$ that is not part of the imbalance structure by 0.5. The filled meta-graph thus creates a number of *ambient nodes* which correspond to entries which are not part of $\mathcal{M}$ and thus are not part of a meaningful cluster; this set of *ambient nodes* is also called the *ambient cluster*. First, we provide examples of structures of $\mathbf{F}$ without any ambient nodes, where $\mathbb{1}$ denotes the indicator function.

•(1) "*cycle*": $\mathbf{F}_{k,l} = (1-\eta)\mathbb{1}(l = ((k+1) \mod K)) + \eta\mathbb{1}(l = ((k-1) \mod K)) + \frac{1}{2}\mathbb{1}(l = k)$.

•(2) "*path*": $\mathbf{F}_{k,l} = (1-\eta)\mathbb{1}(l = k+1) + \eta\mathbb{1}(l = k-1) + \frac{1}{2}\mathbb{1}(l = k)$.

•(3) "*complete*": assign diagonal entries $\frac{1}{2}$. For each pair $(k,l)$ with $k < l$, let $\mathbf{F}_{k,l}$ be $\eta$ and $1-\eta$

with equal probability, then assign $\mathbf{F}_{l,k} = 1 - \mathbf{F}_{k,l}$.

•(4) "*star*", following [50]: select the center node as $\omega = \lfloor \frac{K-1}{2} \rfloor$ and set $\mathbf{F}_{k,l} = (1 - \eta)\mathbb{1}(k = \omega, l \text{ odd}) + \eta\mathbb{1}(k = \omega, l \text{ even}) + (1 - \eta)\mathbb{1}(l = \omega, k \text{ odd}) + \eta\mathbb{1}(\tilde{l} = \omega, l \text{ even})$.

When ambient nodes are present, the construction involves two steps, with the first step the same as the above, but with the following changes: For "*cycle*" meta-graph structure, $\mathbf{F}_{k,l} = (1-\eta)\mathbb{1}(l = ((k+1) \mod (K-1))) + \eta\mathbb{1}(l = ((k-1) \mod (K-1))) + 0.5 \, \mathbb{1}(l = k)$. The second step is to assign $0$ ($0.5$, resp.) to the last row and the last column of $\mathbf{F}$ ($\tilde{\mathbf{F}}$, resp.). Figures 1(c-d) display a "*cycle*" meta-graph structure with ambient nodes (in cluster 4). The majority of edges flow in the form $0 \to 1 \to 2 \to 3 \to 0$, while few flow from the opposite direction. Fig. 1(d) illustrates the meta-graph adjacency matrix corresponding to this $\mathbf{F}$.

In our experiments, we choose the number of clusters, the (approximate) ratio, $\rho$, between the largest and the smallest cluster size, and the number, $n$, of nodes. To tackle the hardest clustering task and also focus on directionality, all pairs of nodes within a cluster and all pairs of nodes between clusters have the same edge probability, $p$. Note that for $\mathcal{M} =$"*cycle*", even the expected in-degree and out-degree of all nodes are identical. Our DSBM, which we denote by DSBM ($\mathcal{M}, \mathbb{1}(\text{ambient}), n, K, p, \rho, \eta$), is built similarly to [12] but with possibly unequal cluster sizes, with more details in App. C.3. For each node $v_i \in \mathcal{C}_k$, and each node $v_j \in \mathcal{C}_l$, independently sample an edge from node $v_i$ to node $v_j$ with probability $p \cdot \tilde{\mathbf{F}}_{k,l}$. The parameter settings in our experiments are $p \in \{0.001, 0.01, 0.02, 0.1\}$, $\rho \in \{1, 1.5\}$, $K \in \{3, 5, 10\}$, $\mathbb{1}(\text{ambient}) \in \{\text{T, F}\}$ (True and False), $n \in \{1000, 5000, 30000\}$, and we also vary the direction flip probability $\eta$ from 0 to 0.45, with a 0.05 step size.

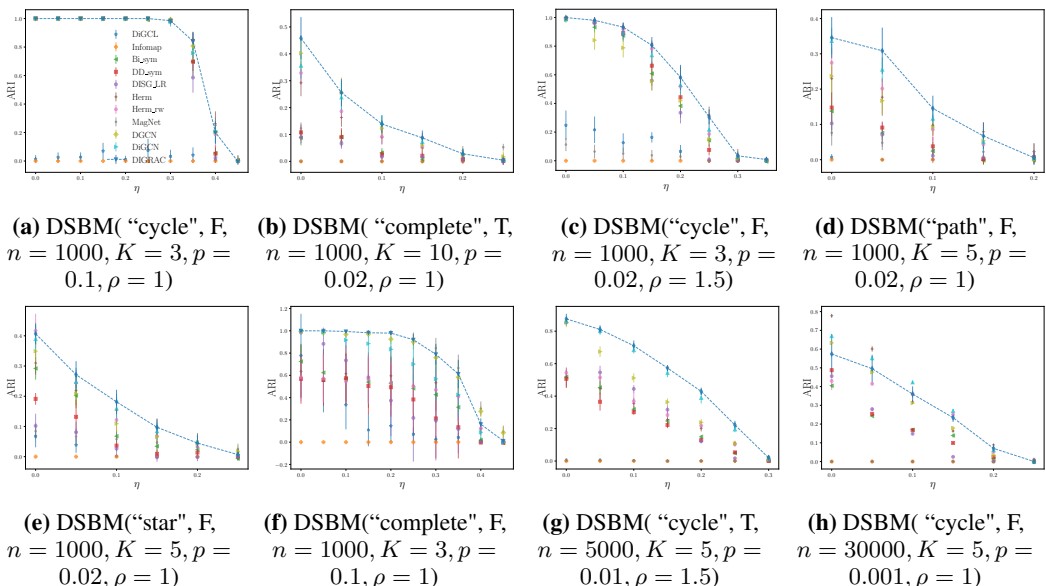

**(a)** DSBM( "cycle", F, $n = 1000, K = 3, p = 0.1, \rho = 1$)

**(b)** DSBM( "complete", T, $n = 1000, K = 10, p = 0.02, \rho = 1$)

**(c)** DSBM("cycle", F, $n = 1000, K = 3, p = 0.02, \rho = 1.5$)

**(d)** DSBM("path", F, $n = 1000, K = 5, p = 0.02, \rho = 1$)

**(e)** DSBM("star", F, $n = 1000, K = 5, p = 0.02, \rho = 1$)

**(f)** DSBM("complete", F, $n = 1000, K = 3, p = 0.1, \rho = 1$)

**(g)** DSBM( "cycle", T, $n = 5000, K = 5, p = 0.01, \rho = 1.5$)

**(h)** DSBM( "cycle", F, $n = 30000, K = 5, p = 0.001, \rho = 1$)

**Figure 3:** Test ARI comparison on synthetic data. Dashed lines highlight DIGRAC's performance. Error bars are given by one standard error.

**Real-world data** We perform experiments on five real-world digraph data sets with sizes ranging from 245 to over 2 million nodes: *Telegram* [3], *Blog* [51], *Migration* [4], *WikiTalk* [52], and *Lead-Lag* [20], with details in App. C.3. We set the number of clusters $K$ to be 4, 2, 10, 10, 10, respectively, and values of $\beta$ to be 5, 1, 9, 10, 3, respectively. Note that *Lead-Lag* comprises of 19 separate networks constructed from yearly financial time series, rendering a total of 23 real-world networks.

## 4.2 Experimental results

### 4.2.1 Training Set-Up

As training setup, we use 10% of all nodes from each cluster as test nodes, 10% as validation nodes to select the model, and the remaining 80% as training nodes. In each setting, unless otherwise stated, we carry out 10 experiments with different data splits. Error bars are given by one standard error.

**Table 1:** Performance comparison on real-world data sets. The best is marked in **bold red** and the second best is marked in underline blue. The objectives are defined in Section 3.1.

| Metric | Data set | InfoMap | Bi_sym | DD_sym | DISG_LR | Herm | Herm_rw | DIGRAC |
|---|---|---|---|---|---|---|---|---|
| $\mathcal{O}^{\text{sort}}_{\text{vol\_sum}}$ | *Telegram* | 0.04±0.00 | 0.21±0.0 | 0.21±0.0 | 0.21±0.01 | 0.2±0.01 | 0.14±0.0 | **0.32±0.01** |
| | *Blog* | 0.07±0.00 | 0.07±0.0 | 0.0±0.0 | 0.05±0.0 | 0.37±0.0 | 0.0±0.0 | **0.44±0.0** |
| | *Migration* | N/A | 0.03±0.00 | 0.01±0.00 | 0.02±0.00 | 0.04±0.00 | 0.02±0.00 | **0.05±0.00** |
| | *WikiTalk* | N/A | N/A | N/A | 0.18±0.03 | 0.15±0.02 | 0.0±0.0 | **0.24±0.05** |
| | *Lead-Lag* | N/A | 0.07±0.01 | 0.07±0.01 | 0.07±0.01 | 0.07±0.02 | 0.07±0.02 | **0.15±0.03** |
| $\mathcal{O}^{\text{std}}_{\text{vol\_sum}}$ | *Telegram* | 0.01±0.00 | 0.26±0.00 | 0.26±0.00 | 0.26±0.01 | 0.25±0.02 | **0.35±0.00** | 0.28±0.01 |
| | *Blog* | 0.00±0.00 | 0.07±0.00 | 0.00±0.00 | 0.05±0.00 | 0.37±0.00 | 0.00±0.00 | **0.44±0.00** |
| | *Migration* | N/A | 0.01±0.00 | 0.01±0.00 | 0.01±0.00 | 0.02±0.00 | 0.02±0.00 | **0.04±0.01** |
| | *WikiTalk* | N/A | N/A | N/A | **0.17±0.04** | 0.06±0.01 | 0.01±0.00 | 0.14±0.02 |
| | *Lead-Lag* | N/A | 0.04±0.01 | 0.04±0.01 | 0.04±0.01 | 0.04±0.01 | 0.04±0.01 | **0.12±0.03** |
| $\mathcal{O}^{\text{naive}}_{\text{vol\_sum}}$ | *Telegram* | 0.01±0.00 | 0.26±0.0 | 0.26±0.0 | 0.26±0.01 | 0.25±0.02 | 0.23±0.0 | **0.27±0.01** |
| | *Blog* | 0.00±0.00 | 0.07±0.0 | 0.0±0.0 | 0.05±0.0 | 0.37±0.0 | 0.0±0.0 | **0.44±0.0** |
| | *Migration* | N/A | 0.01±0.00 | 0.01±0.00 | 0.01±0.00 | 0.02±0.00 | 0.01±0.00 | **0.04±0.01** |
| | *WikiTalk* | N/A | N/A | N/A | 0.1±0.02 | 0.04±0.0 | 0.0±0.0 | **0.12±0.01** |
| | *Lead-Lag* | N/A | 0.30±0.06 | 0.28±0.06 | 0.27±0.06 | 0.29±0.05 | 0.29±0.05 | **0.32±0.11** |

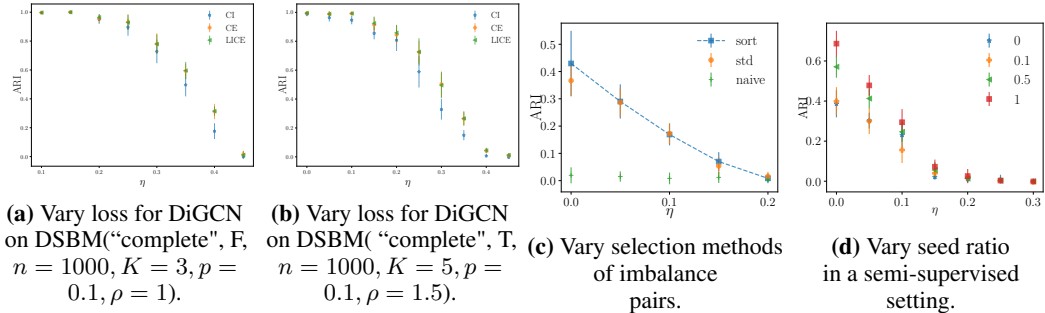

**(a)** Vary loss for DiGCN on DSBM("complete", F, $n = 1000, K = 3, p = 0.1, \rho = 1$).

**(b)** Vary loss for DiGCN on DSBM( "complete", T, $n = 1000, K = 5, p = 0.1, \rho = 1.5$).

**(c)** Vary selection methods of imbalance pairs.

**(d)** Vary seed ratio in a semi-supervised setting.

**Figure 4:** Ablation study. (c-d) are on DSBM("cycle", F, $n = 1000, K = 5, p = 0.02, \rho = 1$).

When no node attributes are given, the matrix $\mathbf{X}$ for DIGRAC is taken as the stacked eigenvectors corresponding to the largest $K$ eigenvalues of the random-walk symmetrized Hermitian matrix used in the comparison method Herm_rw. The imbalance loss function acts on the subgraph induced by the training nodes. To further clarify the training setup, DIGRAC uses 0% of the labels in training. As DIGRAC is a self-supervised method, in principle, we could use all nodes for training. However for a fair comparison with other GNN methods we use only 80% of the nodes for training. For supervised methods our split of 80% - 10% - 10% is a standard split. For the non-GNN methods, all nodes are used for training. The default loss function for DIGRAC is $\mathcal{L}^{\text{sort}}_{\text{vol\_sum}}$.

#### 4.2.2 Results on Synthetic Data

Fig. 3 compares the numerical performance of DIGRAC with other methods on synthetic data. For this Fig. we generate 5 DSBM networks under each parameter setting and use 10 different data splits for each network, then average over the 50 runs. Error bars are given by one standard error. App. C provides additional implementation details.

We conclude that DIGRAC compares favorably against state-of-the-art methods, on a wide range of network densities and noise levels, on different network sizes, and with different underlying meta-graph structures, with and without ambient nodes. Being a self-supervised method, DIGRAC even attains comparable or better performance than fully-supervised GNN competitors.

#### 4.2.3 Results on Real-World Data

For our real-world data sets, the node in- and out-degrees may not be identical across clusters. Moreover, as these data sets do not contain node attributes, DIGRAC considers the eigenvectors

corresponding to the largest $K$ eigenvalues of the Hermitian matrix from [12] to construct an input feature matrix. Table 1 reveals that DIGRAC provides competitive global imbalance scores in three objectives discussed and across all real-world data sets, and outperforms all other methods in 13 out of 15 instances, while attains the second-best performance for the remaining two instances. The N/A entries for *WikiTalk* are caused by memory error, and the N/A entries for InfoMap on *Migration* and *Lead-Lag* are due to its prediction of only one single cluster. For *Migration*, as detailed in Fig. 1(b) and App. D.4, DIGRAC is able to uncover nontrivial migration patterns, such as migration from California to Arizona, as discovered by [4]. *Lead-Lag* results in each year are averaged over ten runs, while the mean and standard deviation values are calculated with respect to the 19 years. The experiments indicate that edge directionality contains an important signal that DIGRAC is able to capture. As App. D.2 illustrates, DIGRAC is able to provide comparable or higher pairwise imbalance scores for the leading pairs. The fitted meta-graph plots in App. D.3 reveal that DIGRAC is able to recover a directed flow imbalance between clusters in all of the selected data sets. A comprehensive numerical comparison in App. D reveals similar conclusions.

### 4.3 Ablation Study

Figures 4(a-b) compare the performance of DiGCN replacing the loss function by $\mathcal{L}_{\text{vol\_sum}}^{\text{sort}}$ from Eq. (3), indicated by "CI" (self-supervised loss only), or "LICE" (sum of supervised and self-supervised loss), on two synthetic models. We find that replacing the supervised loss function with $\mathcal{L}_{\text{vol\_sum}}^{\text{sort}}$ leads to comparable results, and that adding $\mathcal{L}_{\text{vol\_sum}}^{\text{sort}}$ to the loss could be beneficial, indicating that the imbalance objectives are more general than only applicable to DIMPA. Fig. 4(c) compares the test ARI performance using three variants of loss functions on the same digraph. The current choice "*sort*" performs best among these variants, indicating a benefit in only considering top pairs of individual imbalance scores. The "std" variant is comparable with the "sort" variant, but the "sort" variant performs the best with prior knowledge on the network structure. More details on loss functions, comparison with other variants, and evaluation on additional metrics are discussed in App. B, with similar conclusions. As illustrated in Fig. 4(d), again on the same digraph, we also experiment on adding seeds, with the seed ratio defined as the ratio of the number of seed nodes to the number of training nodes. A supervised loss, following [46], is then applied to these seeds; App. C.5 contains additional details. In conclusion, seed nodes with a supervised loss function enhance performance, and we infer that our model can further boost its performance when additional label information is available.

## 5 Conclusion, Limitations and Outlook

DIGRAC provides an end-to-end pipeline to create node embeddings and perform directed clustering, with or without available additional node features or cluster labels. We illustrate DIGRAC on publicly available data without any personally identifiable information. DIGRAC could potentially have societal impact, for example, in detecting clusters of fake accounts in social networks. While we do not envision our work to have any negative societal impact, vigilance is of course required.

Current limitations that could be addressed by future work include detecting the number of clusters [10, 53], instead of specifying it a-priori, as this is typically not available in real-world applications. The relatively small sizes of the networks used in the paper (the largest has 2 million nodes) also opens future direction in adapting our pipeline to extremely large networks, possibly combined with sampling methods or mini-batch [54], rendering DIGRAC applicable to large scale industrial applications. We also intent to further explore the effect of normalization terms in our objectives, and to design more powerful objectives that could explicitly account for varying edge density.

Another future direction pertains to additional experiments in the semi-supervised setting, when there exist seed nodes with known cluster labels, or when additional information is available in the form of *must-link* and *cannot-link* constraints, popular in the *constrained clustering* literature [55, 56]. Further research directions will also address the performance in the sparse regime, where spectral methods are known to underperform, and various regularizations have been proven to be effective theoretically and empirically; e.g., see regularization in the sparse regime for the undirected settings [57–59].

## Author Contributions

Y.H.: Conceptualization, Data curation, Formal analysis, Funding acquisition, Investigation, Methodology, Project administration, Software, Visualization, Writing – original draft; G.R.: Conceptualization, Formal analysis, Funding acquisition, Methodology, Project administration, Supervision, Validation, Writing – review & editing; M.C.: Conceptualization, Data curation, Formal analysis, Funding acquisition, Methodology, Project administration, Resources, Supervision, Visualization, Writing – review & editing.

## Acknowledgements

Y.H. is supported by a Clarendon scholarship. G.R. is supported in part by EPSRC grants EP/T018445/1, EP/W037211/1 and EP/R018472/1. M.C. acknowledges support from the EPSRC grants EP/N510129/1 and EP/W037211/1 at The Alan Turing Institute.

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

# Contents

## A    Directed Mixed Path Aggregation (DIMPA)

To instantiate DIGRAC, we can employ any digraph aggregator that could generate the probability matrix $\mathbf{P}$. In this paper, we devise a simple yet effective directed mixed path aggregation scheme, to obtain the probability assignment matrix $\mathbf{P}$ and feed it to the loss function, as a special case of the successful *SSSNET* method introduced by [46]. Thus, in order to build node embeddings, we capture local network information by taking a weighted average of information from neighbors within $h$ hops. To this end, we row-normalize the adjacency matrix, $\mathbf{A}$, to obtain $\overline{\mathbf{A}}^s$. Similar to the regularization discussed in [60], we add a weighted self-loop to each node and normalize by setting $\overline{\mathbf{A}}^s = (\tilde{\mathbf{D}}^s)^{-1}\tilde{\mathbf{A}}^s$, where $\tilde{\mathbf{A}}^s = \mathbf{A} + \tau\mathbf{I}$, with $\tilde{\mathbf{D}}^s$ the diagonal matrix with entries $\tilde{\mathbf{D}}^s(i,i) = \sum_j \tilde{\mathbf{A}}^s(i,j)$, and $\tau$ is a small value; we take $\tau = 0.5$; see Section C.4 for details.

The $h$-hop **source** matrix is given by $(\overline{\mathbf{A}}^s)^h$. We denote the set of *up-to-$h$-hop* source neighborhood matrices as $\mathcal{A}^{s,h} = \{\mathbf{I}, \overline{\mathbf{A}}^s, \ldots, (\overline{\mathbf{A}}^s)^h\}$. Similarly, for aggregating information when each node is viewed as a **target** node of a link, we carry out the same procedure for $\mathbf{A}^T$ which is the transpose of $\mathbf{A}$. We denote the set of up-to-$h$-hop target neighborhood matrices as $\mathcal{A}^{t,h} = \{\mathbf{I}, \overline{\mathbf{A}}^t, \ldots, (\overline{\mathbf{A}}^t)^h\}$, where $\overline{\mathbf{A}}^t$ is the row-normalized target adjacency matrix calculated from $\mathbf{A}^T$. As convention, the superscript $s$ stands for *source* and the superscript $t$ stands for *target*.

Next, we define two feature mapping functions for source and target embeddings, respectively. Assume that for each node in $\mathcal{V}$, a vector of features is available, and summarize these features in the input feature matrix $\mathbf{X}$. The source embedding is given by

$$\mathbf{Z}^s = \left( \sum_{\mathbf{M} \in \mathcal{A}^{s,h}} \omega_{\mathbf{M}}^s \cdot \mathbf{M} \right) \cdot \mathbf{H}^s \in \mathbb{R}^{n \times d}, \tag{4}$$

where for each $\mathbf{M}$, $\omega_{\mathbf{M}}^s$ is a learnable scalar, $d$ is the dimension of this embedding, and $\mathbf{H}^s = \mathbf{MLP}^{(s,l)}(\mathbf{X})$. Here, the hyperparameter $l$ controls the number of layers in the multilayer perceptron (MLP) with ReLU activation; we fix $l = 2$ throughout. Each layer of the MLP has the same number $d$ of hidden units. The target embedding $\mathbf{Z}^t$ is defined similarly, with $s$ replaced by $t$ in Eq. (4). Different parameters for the MLPs for different embeddings are possible. After these two decoupled aggregations, we concatenate the embeddings to obtain the final node embedding as a $n \times (2d)$ matrix $\mathbf{Z} = \text{CONCAT}(\mathbf{Z}^s, \mathbf{Z}^t)$. The embedding vector $\mathbf{z}_i$ for a node $v_i$ is the $i^{\text{th}}$ row of $\mathbf{Z}$, $\mathbf{z}_i := (\mathbf{Z})_{(i,:)} \in \mathbb{R}^{2d}$.

After obtaining the embedding matrix $\mathbf{Z}$, we apply a linear layer (an affine transformation) to $\mathbf{Z}$, so that the resulting matrix has $K$ columns. Next, we apply the unit *softmax* function to the rows and obtain the assignment probability matrix $\mathbf{P}$. Fig. 5 gives an overview of this implementation.

To avoid computationally expensive and space unfriendly matrix operations, as described in Eq. 4, DIGRAC uses an efficient sparsity-aware implementation, described in Algorithm 1, without explicitly calculating the sets of powers $\mathcal{A}^{s,h}$ and $\mathcal{A}^{t,h}$. We omit the subscript $\mathcal{V}$ for ease of notation. The

**Figure 5:** DIGRAC with DIMPA as aggregator overview: from feature matrix $\mathbf{X}$ and adjacency matrix $\mathbf{A}$, we first compute the row-normalized adjacency matrices $\overline{\mathbf{A}}^s$ and $\overline{\mathbf{A}}^t$. Then, we apply two separate MLPs on $\mathbf{X}$, to obtain hidden representations $\mathbf{H}^s$ and $\mathbf{H}^t$. Next, we compute their decoupled embeddings using Eq. (4), and its equivalent for target embeddings. The concatenated decoupled embeddings are the final embeddings. For node clustering tasks, we add a linear layer followed by a unit *softmax* to obtain the probability matrix $\mathbf{P}$. Applying *argmax* on each row of $\mathbf{P}$ yields node cluster assignments.

algorithm is efficient in the sense that it takes sparse matrices as input, and never explicitly computes a multiplication of two $n \times n$ matrices. Therefore, for input feature dimension $d_{\text{in}}$ and hidden dimension $d$, if $d' = \max(d_{\text{in}}, d) \ll n$, time and space complexity of DIMPA, and implicitly DIGRAC, is $\mathcal{O}(|\mathcal{E}|d'h^2 + 2nd'K)$ and $\mathcal{O}(2|\mathcal{E}| + 4nd' + nK)$, respectively [61, 62].

While it is a current shortcoming of DIGRAC that it does not scale well to very large networks, this limitation is shared by all the GNN competitors compared against in the paper, and some of the spectral methods. DIGRAC scales well in the sense that when the underlying network is sparse, the sparsity is preserved throughout the pipeline. In contrast, Bi_sym and DD_sym [27] construct derived dense matrices for manipulation, rendering the methods no longer scalable. These methods resulted in N/A values in Table 1 in the main text. For large-scale networks, DIMPA is amenable to a minibatch version using neighborhood sampling, similar to the minibatch forward propagation algorithm in [54, 63]. We are also aware of a framework [64] for scaling up graph neural networks automatically, where theoretical guarantees are provided, and ideas there will be exploited in future. We expect that the theoretical guarantees could be adapted to our situation.

---

**Algorithm 1:** Weighted Multi-Hop Neighbor Aggregation (DIMPA).

---

**Input** : (Sparse) row-normalized adjacency matrices $\overline{\mathbf{A}}^s, \overline{\mathbf{A}}^t$; initial hidden representations $\mathbf{H}^s, \mathbf{H}^t$; hop $h (h \geq 2)$; lists of scalar weights $\Omega^s = (\omega_{\mathbf{M}}^s, \mathbf{M} \in \mathcal{A}^{s,h}), \Omega^t = (\omega_{\mathbf{M}}^t, \mathbf{M} \in \mathcal{A}^{t,h})$.

**Output**: Vector representations $\mathbf{z}_i$ for all $v_i \in \mathcal{V}$ given by $\mathbf{Z}$.

$\quad \tilde{\mathbf{X}}^s \leftarrow \overline{\mathbf{A}}^s \mathbf{H}^s; \qquad\qquad\qquad\qquad\quad \tilde{\mathbf{X}}^t \leftarrow \overline{\mathbf{A}}^t \mathbf{H}^t;$

$\quad \mathbf{Z}^s \leftarrow \Omega^s[0] \cdot \mathbf{H}^s + \Omega^s[1] \cdot \tilde{\mathbf{X}}^s; \qquad\quad \mathbf{Z}^t \leftarrow \Omega^t[0] \cdot \mathbf{H}^t + \Omega^t[1] \cdot \tilde{\mathbf{X}}^t;$

**for** $i \leftarrow 2$ **to** $h$ **do**

$\quad\Big| \quad \tilde{\mathbf{X}}^s \leftarrow \overline{\mathbf{A}}^s \tilde{\mathbf{X}}^s; \qquad\qquad\qquad\qquad \tilde{\mathbf{X}}^t \leftarrow \overline{\mathbf{A}}^t \tilde{\mathbf{X}}^t;$

$\quad\Big| \quad \mathbf{Z}^s \leftarrow \mathbf{Z}^s + \Omega^s[i] \cdot \tilde{\mathbf{X}}^s; \qquad\qquad \mathbf{Z}^t \leftarrow \mathbf{Z}^t + \Omega^t[i] \cdot \tilde{\mathbf{X}}^t;$

**end**

$\mathbf{Z} = \text{CONCAT}(\mathbf{Z}^s, \mathbf{Z}^t);$

---

# B  Loss and Objectives

## B.1  Proof of Proposition 1

Moreover, we clarify that we make an assumption on the limiting behavior of the weights, namely that

$$\frac{\max_e |w_e|}{\sqrt{\sum_e w_e^2}} = o(m(k,l))$$

where $m(k,l)$ is the number of edges. This is a natural assumption: In the case that all weights are equal in absolute value, this assumption is satisfied as then $\frac{\max_e |w_e|}{\sqrt{\sum_e w_e^2}} = \frac{1}{\sqrt{m(k,l)}}$. The assumption is

generally satisfied when there is not too much variability in the weights. If for example all but one weight pair was equal to 0, then the assumption would be violated, and also a normal approximation would not hold as there would only be two non-zero observations.

**Proposition 2.** *Suppose that $\mathcal{C}_k$ and $\mathcal{C}_l$ are two clusters of $n_k$ and $n_l$ nodes, respectively, with $m(k, l)$ edges between them, with symmetric edge weights $w_{ij} = w_{ji} \in [0, 1]$ and with edge direction drawn independently at random with equal probability $\frac{1}{2}$ for each direction. We assume that the edge weights satisfy $\frac{\max_e |w_e|}{\sqrt{\sum_e w_e^2}} = o(m(k, l))$. Then $W(\mathcal{C}_k, \mathcal{C}_l) - W(\mathcal{C}_l, \mathcal{C}_k)$ is approximately normally distributed with mean 0 and variance $||w||^2$ as $m(k, l) \to \infty$.*

**Proof.** For each edge between the two clusters $\mathcal{C}_k$ and $\mathcal{C}_l$, the edge direction is random, i.e. the edge is from $\mathcal{C}_k$ to $\mathcal{C}_l$ with probability 0.5, and $\mathcal{C}_l$ to $\mathcal{C}_k$ with probability 0.5 also. Let $\mathcal{E}^{k,l}$ denote the set of $m(k, l) > 0$ edges between $\mathcal{C}_k$ and $\mathcal{C}_l$. For every edge $e \in \mathcal{E}^{k,l}$, the edge direction is encoded by a Rademacher random variable $X_e$ with $X_e = 1$ if the edge is from $\mathcal{C}_k$ to $\mathcal{C}_l$, and $X_e = -1$ otherwise. Then $(X_e + 1)/2 \sim Ber(0.5)$ is a Bernoulli(0.5) random variable with mean $2 \times 0.5 - 1 = 0$ and variance $2^2 \times 0.5 \times (1 - 0.5) = 1$. We have the representation

$$W(\mathcal{C}_k, \mathcal{C}_l) - W(\mathcal{C}_l, \mathcal{C}_k) = \sum_{e \in \mathcal{E}^{k,l}} X_e w_e$$

as the sum of $m(k, l)$ independent bounded random variables with finite third moments. Moreover, $W(\mathcal{C}_k, \mathcal{C}_l) - W(\mathcal{C}_l, \mathcal{C}_k)$ has mean 0 and variance $||w||^2$. The assertion now follows from a version of the Central Limit Theorem, Theorem 3.4 in [65]; we repeat the relevant part here:

**Theorem 1** (Extract from Theorem 3.4 in [65])**.** *Let $\xi_1, \ldots, \xi_n$ be independent random variables with zero means satisfying $\sum_{i=1}^n Var(\xi_i) = 1$ and assume that there is a $\delta > 0$ such that $|\xi_i| \leq \delta$ for $1 \leq i \leq n$. Let $\Phi$ denote the cumulative distribution function of the standard normal distribution. Then*

$$\sup_{z in \mathbb{R}} \left| \mathbb{P} \left( \sum_{i=1}^n \xi_i \leq z \right) - \Phi(z) \right| \leq 3.3\delta.$$

We apply this theorem with $n$ replaced by $m(k, l)$, the number of edges, and take $\xi_e = \frac{X_e w_e}{\sqrt{\sum_e w_e^2}}$.

Then $\xi_e$ has mean zero and, using an enumeration of the edges, $\sum_{e=1}^{m(k,l)} Var(\xi_e) = 1$. Moreover, $|\xi_e| \leq \frac{\max_e |w_e|}{\sqrt{\sum_e w_e^2}} =: \delta$ holds for all $e \in \{1, \ldots, m(k, l)\}$ and hence the theorem applies for the limit $m(k, l) \to \infty$. The stated result follows from using that if $Z/\sigma$ has the standard normal distribution then $Z$ has the mean zero normal distribution with variance $\sigma^2$.

$\square$

## B.2 Additional Details on Probabilistic Cut and Volume

Recall that the **probabilistic cut** from cluster $\mathcal{C}_k$ to $\mathcal{C}_l$ is defined as

$$W(\mathcal{C}_k, \mathcal{C}_l) = \sum_{i,j \in \{1, \ldots, n\}} \mathbf{A}_{i,j} \cdot \mathbf{P}_{i,k} \cdot \mathbf{P}_{j,l} = (\mathbf{P}_{(:,k)})^T \mathbf{A} \mathbf{P}_{(:,l)},$$

where $\mathbf{P}_{(:,k)}, \mathbf{P}_{(:,l)}$ denote the $k^{\text{th}}$ and $l^{\text{th}}$ columns of the assignment probability matrix $\mathbf{P}$, respectively. The **imbalance flow** between clusters $\mathcal{C}_k$ and $\mathcal{C}_l$ is defined as

$$|W(\mathcal{C}_k, \mathcal{C}_l) - W(\mathcal{C}_l, \mathcal{C}_k)|,$$

for $k, l \in \{0, \ldots, K - 1\}$. The loss functions proposed in the main paper can be understood in terms of a probabilistic notion of degrees, as follows. We define the probabilistic out-degree of node $v_i$ with respect to cluster $k$ by $\tilde{d}_{i,k}^{(\text{out})} = \sum_{j=1}^n \mathbf{A}_{i,j} \cdot \mathbf{P}_{j,k} = (\mathbf{A}\mathbf{P}_{(:,k)})_i$, where subscript $i$ refers to the $i^{\text{th}}$ entry of the vector $\mathbf{A}\mathbf{P}_{(:,k)}$. Similarly, we define the probabilistic in-degree of node $v_i$ with respect to cluster $k$ by $\tilde{d}_{i,k}^{(\text{in})} = (\mathbf{A}^T \mathbf{P}_{(:,k)})_i$, where $\mathbf{A}^T$ is the transpose of $\mathbf{A}$. The **probabilistic degree** of node $v_i$ with respect to cluster $k$ is $\tilde{d}_{i,k} = \tilde{d}_{i,k}^{(\text{in})} + \tilde{d}_{i,k}^{(\text{out})} = ((\mathbf{A}^T + \mathbf{A})\mathbf{P}_{(:,k)})_i = \sum_{j=1}^n (\mathbf{A}_{i,j} + \mathbf{A}_{j,i}) \cdot \mathbf{P}_{j,k}$.

For comparisons and ease of interpretation, it is advantageous to normalize the imbalance flow between clusters; for this purpose, we introduce the probabilistic volume of a cluster, as follows.

The *probabilistic out-volume* for cluster $\mathcal{C}_k$ is defined as $VOL^{\text{(out)}}(\mathcal{C}_k) = \sum_{i,j} \mathbf{A}_{j,i} \cdot \mathbf{P}_{j,k}$, and the *probabilistic in-volume* for cluster $\mathcal{C}_k$ is defined as $VOL^{\text{(in)}}(\mathcal{C}_k)(\mathbf{A}^T \mathbf{P}_{(:,k)})_i$, where $\mathbf{A}^T$ is the transpose of $\mathbf{A}$. These volumes can be viewed as sum of probabilistic out-degrees and in-degrees, respectively; for example, $VOL^{\text{(in)}}(\mathcal{C}_k) = \sum_{i=1}^n \tilde{d}_{i,k}^{\text{(in)}}$. Then, it holds true that

$$VOL^{\text{(out)}}(\mathcal{C}_k) = \sum_{i,j} \mathbf{A}_{i,j} \cdot \mathbf{P}_{i,k} \geq \sum_{i,j} \mathbf{A}_{i,j} \cdot \mathbf{P}_{i,k} \cdot \mathbf{P}_{j,l} = W(\mathcal{C}_k, \mathcal{C}_l), \tag{5}$$

since entries in $\mathbf{P}$ are probabilities, which are in $[0, 1]$, and all entries of $\mathbf{A}$ are nonnegative. Similarly, $VOL^{\text{(in)}}(\mathcal{C}_k) \geq W(\mathcal{C}_l, \mathcal{C}_k)$.

The **probabilistic volume** for cluster $\mathcal{C}_k$ is defined as

$$VOL(\mathcal{C}_k) = VOL^{\text{(out)}}(\mathcal{C}_k) + VOL^{\text{(in)}}(\mathcal{C}_k) = \sum_{i,j}(\mathbf{A}_{i,j} + \mathbf{A}_{j,i}) \cdot \mathbf{P}_{j,k}.$$

Then, it holds true that $VOL(\mathcal{C}_k) \geq W(\mathcal{C}_k, \mathcal{C}_l)$ for all $l \in \{0, \ldots, K-1\}$ and

$$\min(VOL(\mathcal{C}_k), VOL(\mathcal{C}_l)) \geq \max(W(\mathcal{C}_k, \mathcal{C}_l), W(\mathcal{C}_l, \mathcal{C}_k)) \geq |W(\mathcal{C}_k, \mathcal{C}_l) - W(\mathcal{C}_l, \mathcal{C}_k)|. \tag{6}$$

When there exists a strong imbalance, then $|W(\mathcal{C}_k, \mathcal{C}_l) - W(\mathcal{C}_l, \mathcal{C}_k)| \approx \max(W(\mathcal{C}_k, \mathcal{C}_l), W(\mathcal{C}_l, \mathcal{C}_k))$. As an extreme case, if $\mathbf{P}_{j,l} = 1$ for all nonnegative terms in the summations in Eq. (5), and $VOL^{\text{(in)}}(\mathcal{C}_k) = 0$, then $|W(\mathcal{C}_k, \mathcal{C}_l) - W(\mathcal{C}_l, \mathcal{C}_k)| = VOL(\mathcal{C}_k)$.

## B.3 Variants of Normalization

Recall that the imbalance term involved in most of our experiments, named $\text{CI}^{\text{vol\_sum}}$, is defined as

$$\text{CI}^{\text{vol\_sum}}(k, l) = 2\frac{|W(\mathcal{C}_k, \mathcal{C}_l) - W(\mathcal{C}_l, \mathcal{C}_k)|}{VOL(\mathcal{C}_k) + VOL(\mathcal{C}_l)} \in [0, 1]. \tag{7}$$

An alternative, which does not take volumes into account, is given by

$$\text{CI}^{\text{plain}}(k, l) = \left|\frac{W(\mathcal{C}_k, \mathcal{C}_l) - W(\mathcal{C}_l, \mathcal{C}_k)}{W(\mathcal{C}_k, \mathcal{C}_l) + W(\mathcal{C}_l, \mathcal{C}_k)}\right| = 2\left|\frac{W(\mathcal{C}_k, \mathcal{C}_l)}{W(\mathcal{C}_k, \mathcal{C}_l) + W(\mathcal{C}_l, \mathcal{C}_k)} - \frac{1}{2}\right| \in [0, 1]. \tag{8}$$

We call this cut flow imbalance $\text{CI}^{\text{plain}}$ as it does not penalize extremely unbalanced cluster sizes.

To achieve balanced cluster sizes and still constrain each imbalance term to be in $[0, 1]$, one solution is to multiply the imbalance flow value by the minimum of $VOL(\mathcal{C}_k)$ and $VOL(\mathcal{C}_l)$, and then divide by $\max_{(k',l') \in \mathcal{T}}(\min(VOL(\mathcal{C}_{k'}), VOL(\mathcal{C}_{l'})))$, where $\mathcal{T} = \{(\mathcal{C}_k, \mathcal{C}_l) : 0 \leq k < l \leq K-1, k, l \in \mathbb{Z}\}$. The reason for using $\mathcal{T}$ is that $\text{CI}^{\text{plain}}(k, l)$ is symmetric with respect to $k$ and $l$, and $\text{CI}^{\text{plain}}(k, l) = 0$ whenever $k = l$. Note that the maximum of the minimum here equals the second largest volume among clusters. We then obtain $\text{CI}^{\text{vol\_min}}$ as

$$\text{CI}^{\text{vol\_min}}(k, l) = \text{CI}^{\text{plain}}(k, l) \times \frac{\min(VOL(\mathcal{C}_k), VOL(\mathcal{C}_l))}{\max_{(k',l') \in \mathcal{T}}(\min(VOL(\mathcal{C}_{k'}), VOL(\mathcal{C}_{l'})))}. \tag{9}$$

Another potential choice, denoted $\text{CI}^{\text{vol\_max}}$, whose normalization follows from the same reasoning as $\text{CI}^{\text{vol\_sum}}$, is given by

$$\text{CI}^{\text{vol\_max}}(k, l) = \frac{|W(\mathcal{C}_k, \mathcal{C}_l) - W(\mathcal{C}_l, \mathcal{C}_k)|}{\max(VOL(\mathcal{C}_k), VOL(\mathcal{C}_l))} \in [0, 1]. \tag{10}$$

Note that the current $\text{CI}^{\text{vol\_sum}}(k, l)$ term can be reformulated as

$$\text{CI}^{\text{vol\_sum}}(k, l) = 2\frac{|W(\mathcal{C}_k, \mathcal{C}_l) - W(\mathcal{C}_l, \mathcal{C}_k)|}{VOL(\mathcal{C}_k) + VOL(\mathcal{C}_l)} = 2\frac{W(\mathcal{C}_k, \mathcal{C}_l) + W(\mathcal{C}_l, \mathcal{C}_k)}{VOL(\mathcal{C}_k) + VOL(\mathcal{C}_l)} \times \text{CI}^{\text{plain}}(k, l), \tag{11}$$

with the first term in the decomposition corresponding to the relative ratio of inter- and intra-cluster edge density. For our synthetic data, this term is constant as we have constant edge density across the graph. However, for certain real-world data sets, one could also maximize this first term by increasing the inter-cluster density while decreasing the intra-cluster density, which seems to be a side effect. However, in our experiments, we also evaluate our results with different metrics, including objectives without any normalization, and conclude that this side effect does not create any issues in our data sets.

**Table 2:** Naming conventions for objectives and loss functions

| Selection variant / $CI$ | $CI^{\text{vol\_sum}}$ | | $CI^{\text{vol\_min}}$ | | $CI^{\text{vol\_max}}$ | | $CI^{\text{plain}}$ | |
|---|---|---|---|---|---|---|---|---|
| sort | $\mathcal{O}^{\text{sort}}_{\text{vol\_sum}},$ | $\mathcal{L}^{\text{sort}}_{\text{vol\_sum}}$ | $\mathcal{O}^{\text{sort}}_{\text{vol\_min}},$ | $\mathcal{L}^{\text{sort}}_{\text{vol\_min}}$ | $\mathcal{O}^{\text{sort}}_{\text{vol\_max}},$ | $\mathcal{L}^{\text{sort}}_{\text{vol\_max}}$ | $\mathcal{O}^{\text{sort}}_{\text{plain}},$ | $\mathcal{L}^{\text{sort}}_{\text{plain}}$ |
| std | $\mathcal{O}^{\text{std}}_{\text{vol\_sum}},$ | $\mathcal{L}^{\text{std}}_{\text{vol\_sum}}$ | $\mathcal{O}^{\text{std}}_{\text{vol\_min}},$ | $\mathcal{L}^{\text{std}}_{\text{vol\_min}}$ | $\mathcal{O}^{\text{std}}_{\text{vol\_max}},$ | $\mathcal{L}^{\text{std}}_{\text{vol\_max}}$ | $\mathcal{O}^{\text{std}}_{\text{plain}},$ | $\mathcal{L}^{\text{std}}_{\text{plain}}$ |
| naive | $\mathcal{O}^{\text{naive}}_{\text{vol\_sum}},$ | $\mathcal{L}^{\text{naive}}_{\text{vol\_sum}}$ | $\mathcal{O}^{\text{naive}}_{\text{vol\_min}},$ | $\mathcal{L}^{\text{naive}}_{\text{vol\_min}}$ | $\mathcal{O}^{\text{naive}}_{\text{vol\_max}},$ | $\mathcal{L}^{\text{naive}}_{\text{vol\_max}}$ | $\mathcal{O}^{\text{naive}}_{\text{plain}},$ | $\mathcal{L}^{\text{naive}}_{\text{plain}}$ |

## B.4 Selection of the Loss Function

Table 2 provides naming conventions of all the twelve pairs of variants of objectives and loss functions used in this paper. We select the loss functions for DIGRAC based on two representative models, and compare the performance of different loss functions. We use DIMPA (introduced in A) as an instantiation of DIGRAC's aggregator, for which $d = 32$, hidden units, $h = 2$ hops, and no seed nodes. Figures 6(a) and 7 compare twelve choices of loss combinations on a DSBM with $n = 1000$ nodes, $K = 5$ blocks, $\rho = 1, p = 0.02$ without ambient nodes, with a complete meta-graph structure. The subscript indicates the choice of pairwise imbalance, and the superscript indicates the variant for selecting pairs. Figures 6(b) and 8 are based on a DSBM with $n = 1000$ nodes, $K = 5$ blocks, $\rho = 1, p = 0.02$ without ambient nodes, with a cycle meta-graph structure. For these figures, dash lines highlight the "*sort*" variant as well as the "*std*" variant based on $CI^{\text{vol\_sum}}$, which have been introduced in the main text.

We also plot the imbalance evolution curves for the above two synthetic models when $\eta = 0.05$, for all the loss variants, in Figure 9.

These figures indicate that the "*sort*" variant generally provides the best test ARI performance and the best overall global imbalance scores, among which using normalizations $CI^{\text{vol\_sum}}$ and $CI^{\text{vol\_max}}$ perform the best. The "*std*" variant is comparable with the "*sort*" variant in many instances, but is less stable in performance. We observe, however, from Figure 9, that the "std" variants normally converge much faster. Taking the above into account, if we have prior knowledge of the network structure, or when we could conduct some prior analysis on the value $\beta$ to take, the "*sort*" variant should be the variant of choice. Further, from Figure 9, we observe that normalization in the loss function helps avoid the degenerate situation that the loss does not decrease. Such degeneracy can occur in the "plain" variants, raising issues about the practical usefulness of these variants. We observe that $\mathcal{L}^{\text{sort}}_{\text{vol\_min}}$ appears to behave worse than $\mathcal{L}^{\text{sort}}_{\text{vol\_sum}}$ and $\mathcal{L}^{\text{sort}}_{\text{vol\_max}}$, even when using the "sort" variant to select pairwise imbalance scores. One possible explanation is that $\mathcal{L}^{\text{sort}}_{\text{vol\_min}}$ does not penalize extreme volume sizes, and that it takes minimum as well as maximum which, as functions of the data, are not as smooth as taking a summation. Throughout our experiments in the main text, we hence use the loss function $\mathcal{L}^{\text{sort}}_{\text{vol\_sum}}$.

## C  Implementation Details

### C.1  Code

To fully reproduce our results, code and preprocessed data are available at `https://github.com/SherylHYX/DIGRAC_Directed_Clustering`.

### C.2  Hardware

Experiments were conducted on a compute node with 8 Nvidia RTX 8000, 48 Intel Xeon Silver 4116 CPUs and 1000GB RAM, a compute node with 4 NVIDIA GeForce RTX 2080, 32 Intel Xeon E5-2690 v3 CPUs and 64GB RAM, a compute node with 2 NVIDIA Tesla K80, 16 Intel Xeon E5-2690 CPUs and 252GB RAM, and an Intel 2.90GHz i7-10700 processor with 8 cores and 16 threads.

With this setup, all experiments for spectral methods, MagNet, DiGCL, and DIGRAC can be completed within two days, including repeated experiments, to obtain averages over multiple runs. DGCN, DiGCN, and MagNet have much longer run time (especially DGCN, which is space-consuming, and we cannot run many experiments in parallel), with a total of three days for them to finish. The slow speed stems from the competitor methods; some of the other GNN methods take a long time to

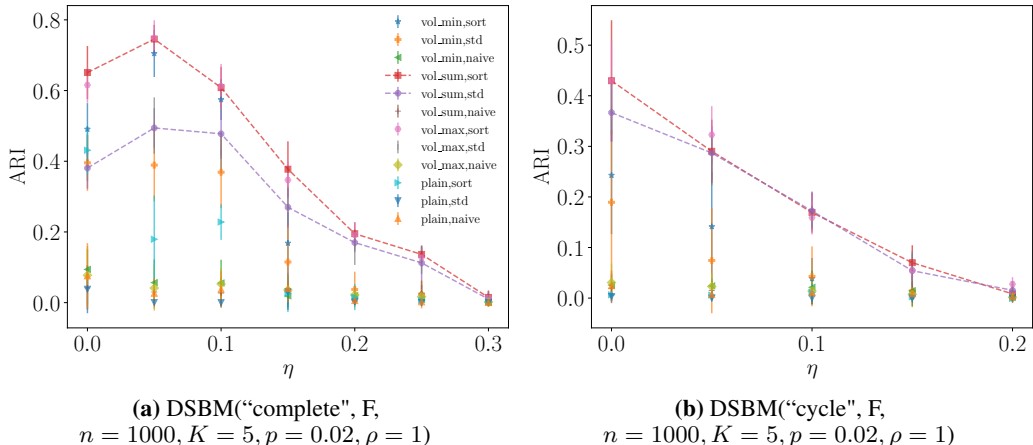

**(a)** DSBM("complete", F,
$n = 1000, K = 5, p = 0.02, \rho = 1$)

**(b)** DSBM("cycle", F,
$n = 1000, K = 5, p = 0.02, \rho = 1$)

**Figure 6:** ARI comparison of loss functions on DSBM with 1000 nodes, 5 blocks, $\rho = 1, p = 0.02$ without ambient nodes, of cycle (left) and complete (right) meta-graph structures, respectively. The first component of the legend is the choice of pairwise imbalance, and the second component is the variant of selecting pairs. The naming conventions for the abbreviations in the legend are provided in Table 2.

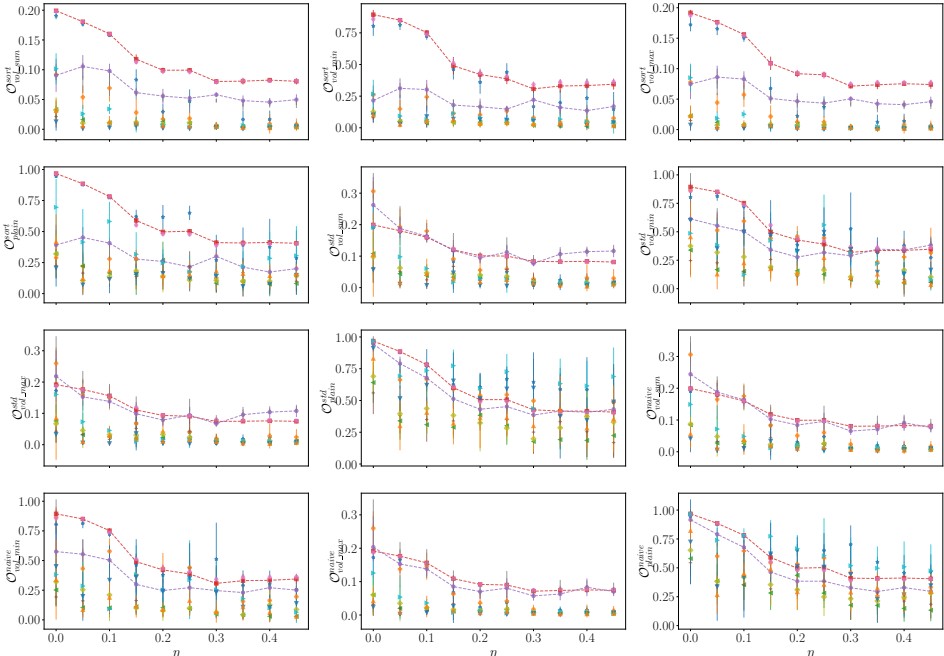

**Figure 7:** Imbalance scores comparison of loss functions on DSBM with 1000 nodes, 5 blocks, $\rho = 1, p = 0.02$ without ambient nodes, of the **complete meta-graph** structure. The legend is the same as Fig. 6(a).

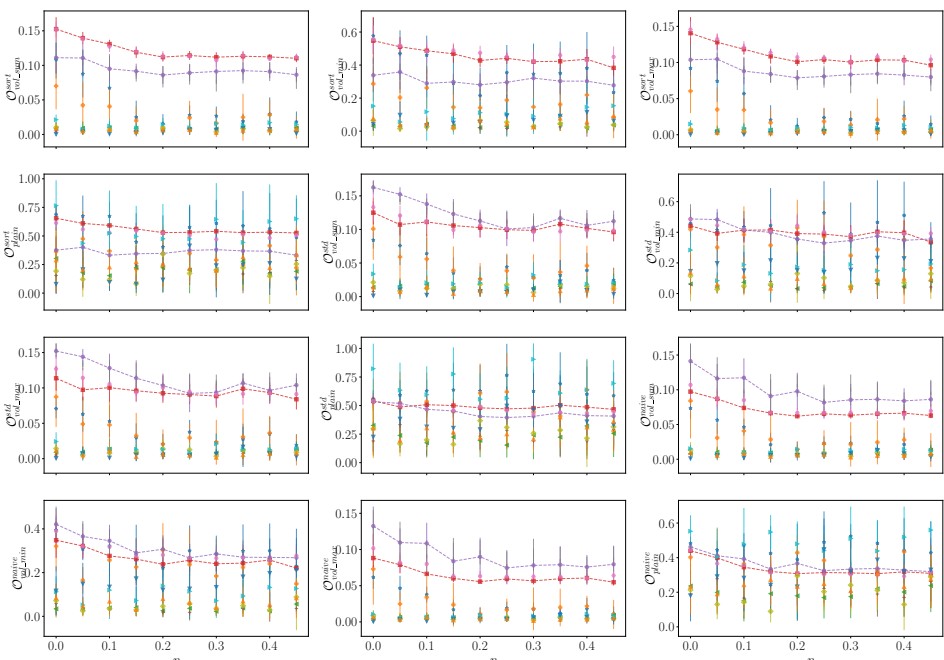

**Figure 8:** Imbalance scores comparison of loss functions on DSBM with 1000 nodes, 5 blocks, $\rho = 1, p = 0.02$ without ambient nodes, of the **cyclic meta-graph** structure. The legend is the same as Fig. 6(a).

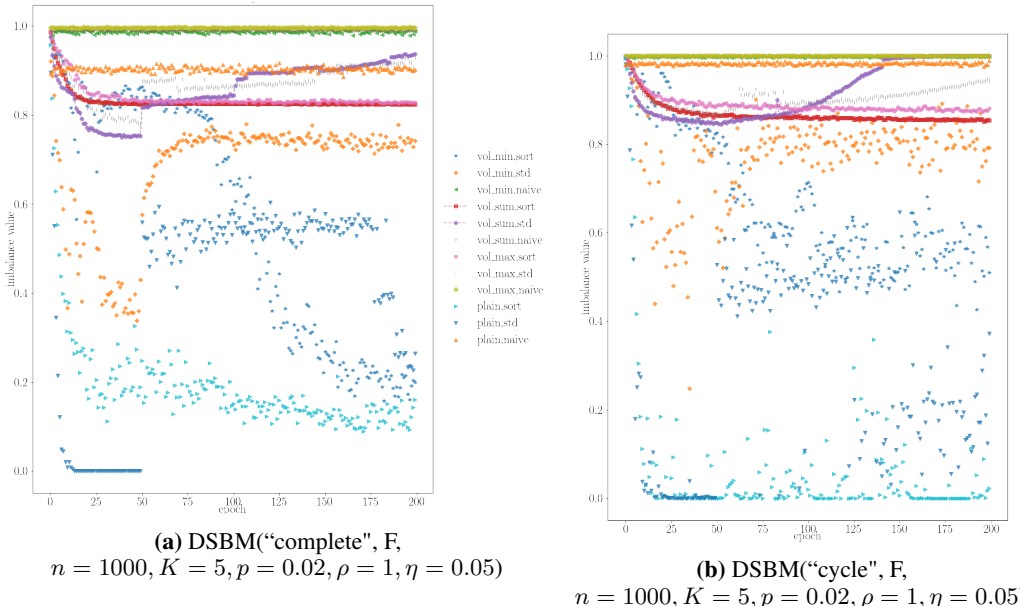

**(a)** DSBM("complete", F, $n = 1000, K = 5, p = 0.02, \rho = 1, \eta = 0.05$)

**(b)** DSBM("cycle", F, $n = 1000, K = 5, p = 0.02, \rho = 1, \eta = 0.05$)

**Figure 9:** Imbalance loss evolution comparison of loss functions on DSBM with 1000 nodes, 5 blocks, $\rho = 1, p = 0.02, \eta = 0.05$ without ambient nodes, of cycle (left) and complete (right) meta-graph structures, respectively. The first component of the legend is the choice of pairwise imbalance, and the second component is the variant of selecting pairs. The naming conventions for the abbreviations in the legend are provided in Table 2.

run. Table 1 in the main text shows N/A values for Bi_sym and for DD_sym exactly for this reason. Empirically, DIGRAC is among the fastest among all GNN methods to which it is compared. In detail, Table 3 reports the average runtime for all GNN methods on a variety of DSBM models, and illustrates that DIGRAC indeed takes the least or second least computational time per epoch. The results are averaged over 10 runs for the first 200 epochs. DiGCL is also efficient in running time, but with worse performance than DIGRAC even as a supervised method, see the enlarged synthetic results in Sec. C.8 (Figure 13). The total number of epochs required until the validation loss does not decrease for 200 epochs (or the maximum number of 1000 epochs is reached) varies for different data sets.

**Table 3:** GNN average runtime (seconds per epoch) comparison. The results are averaged over 10 runs for the first 200 epochs. The fastest is highlighted in **bold red** while the second fastest is marked with underline blue.

| Runtime (second per epoch on average)/GNN method | DiGCL | DGCN | DiGCN | MagNet | DIGRAC |
|---|---|---|---|---|---|
| DSBM( "complete", T, $n = 1000, K = 5, p = 0.1, \rho = 1.5, \eta = 0.1$) | **0.107** | 0.606 | 0.469 | 0.369 | 0.308 |
| DSBM( "path", F, $n = 1000, K = 5, p = 0.02, \rho = 1, \eta = 0.15$) | **0.061** | 0.227 | 0.212 | 0.238 | 0.201 |
| DSBM( "star", F, $n = 1000, K = 5, p = 0.02, \rho = 1, \eta = 0.3$) | **0.095** | 0.305 | 0.294 | 0.324 | 0.292 |
| DSBM( "star", F, $n = 5000, K = 5, p = 0.02, \rho = 1, \eta = 0.4$) | 0.222 | 0.966 | 0.276 | 0.116 | **0.101** |
| DSBM( "cycle", F, $n = 5000, K = 5, p = 0.01, \rho = 1.5, \eta = 0$) | 0.177 | 0.330 | 0.099 | 0.095 | **0.089** |
| DSBM( "cycle", F, $n = 30000, K = 5, p = 0.001, \rho = 1, \eta = 0$) | **0.070** | 0.868 | 0.208 | 0.183 | 0.156 |

## C.3 Data

### C.3.1 Data Splits and Preprocessing

The results comparing DIGRAC with other methods on synthetic data are averaged over 50 runs, five synthetic networks under the same setting, each with 10 different data splits. For synthetic data, 10% of all nodes are selected as test nodes for each cluster (the actual number is the ceiling of the total number of nodes times 0.1, to avoid falling below 10% of test nodes), 10% are selected as validation nodes (for model selection and early-stopping; again, we consider the ceiling for the actual number), while the remaining roughly 80% are selected as training nodes (the actual number can never be higher than 80% due to using the ceiling for both the test and validation splits). To further clarify the training setup, we use 0% of the labels in training. As DIGRAC is a self-supervised method, in principle, we could use all nodes for training. However, for a fair comparison with other GNN methods, we use only 80% of the nodes for training. For supervised methods our split of 80% - 10% - 10% is a standard split. For the non-GNN methods, all nodes are used for training.

For both synthetic and real-world data sets, we extract the largest weakly connected component for experiments, as our framework could be applied to different weakly connected components, if the digraph is disconnected. Isolated nodes do not include any imbalance information. As customary in community detection, they are often omitted in real networks. When "ground-truth" is given, test results are averaged over 10 different data splits on one network. When no labels are available, results are averaged over 10 different data splits.

Averaged results are reported with error bars representing one standard deviation in the figures, and plus/minus one standard deviation in the tables.

### C.3.2 Synthetic Data

Our synthetic data, DSBM, which we denote by DSBM $(\mathcal{M}, \mathbb{1}(\text{ambient}), n, K, p, \rho, \eta)$, is built similarly to [12] but with possibly unequal cluster sizes: ●(1) Assign cluster sizes $n_0 \leq n_1 \leq \cdots \leq n_{K-1}$ with size ratio $\rho \geq 1$ , as follows. If $\rho = 1$ then the first $K - 1$ clusters have the same size $\lfloor n/K \rfloor$ and the last cluster has size $n - (K - 1)\lfloor n/K \rfloor$. If $\rho > 1$, we set $\rho_0 = \rho^{\frac{1}{K-1}}$. Solving $\sum_{i=0}^{K-1} \rho_0^i n_0 = n$ and taking integer value gives $n_0 = \lfloor n(1 - \rho_0)/(1 - \rho_0^K) \rfloor$ . Further, set $n_i = \lfloor \rho_0 n_{i-1} \rfloor$, for $i = 1, \cdots, K - 2$ if $K \geq 3$, and $n_{K-1} = n - \sum_{i=0}^{K-2} n_i$. Then the ratio of the size of the largest to the smallest cluster is approximately $\rho_0^{K-1} = \rho$. ●(2) Assign each node randomly to one of $K$ clusters, so that each cluster has the allocated size. ●(3) For node $v_i, v_j \in \mathcal{C}_k$, independently sample an edge from node $v_i$ to node $v_j$ with probability $p \cdot \tilde{\mathbf{F}}_{k,k}$. ●(4) For each pair

of different clusters $\mathcal{C}_k, \mathcal{C}_l$ with $k \neq l$, for each node $v_i \in \mathcal{C}_k$, and each node $v_j \in \mathcal{C}_l$, independently sample an edge from node $v_i$ to node $v_j$ with probability $p \cdot \tilde{\mathbf{F}}_{k,l}$.

### C.3.3 Real-World Data

For real-world data sets, we choose the number $K$ of clusters in the meta-graph and the number $\beta$ of edges between clusters in the meta-graph as follows. As they are needed as input for DIGRAC, we resort to Herm_rw [12] as an initial view of the network clustering. When a suitable meta-graph is suggested in a previous publication, then we use that choice. Otherwise, the number $K$ of clusters is determined using the clustering from Herm_rw. First, we pick a range of $K$, and for each $K$, we calculate the global imbalance scores and plot the predicted meta-graph flow matrix $\mathbf{F}'$ based on the clustering from Herm_rw. Its entries are defined as

$$\mathbf{F}'(k,l) = \mathbb{1}(W(\mathcal{C}_k, \mathcal{C}_l) + W(\mathcal{C}_l, \mathcal{C}_k) > 0) \times \frac{W(\mathcal{C}_k, \mathcal{C}_l)}{W(\mathcal{C}_k, \mathcal{C}_l) + W(\mathcal{C}_l, \mathcal{C}_k)}. \tag{12}$$

These entries can be viewed as predicted probabilities of edge directions. Then, we choose $K$ from this range so that the predicted meta-graph flow matrix has the highest imbalance scores and strong imbalance in the predicted meta-graph flow matrix.

The choice of $\beta$, which we assume should be equal to the number of edges in the meta-graph, is as follows. We plot the ranked pairs of CI$^{\text{plain}}$ values from Herm_rw and select the $\beta$ which is at least as large as $K - 2$, to allow the meta-graph to be connected, and which corresponds to a large drop in the plot.

Here we provide a brief description for each of the data sets; Table 4 gives the number, $n$, of nodes, the number, $|\mathcal{E}|$, of directed edges, the number $|\mathcal{E}^r|$, of reciprocal edges (self-loops are counted once and for $u \neq v$, a reciprocal edge $u \rightarrow v, v \rightarrow u$ is counted twice) as well as their percentage among all edges, for the real-world networks, illustrating the variability in network size and density (defined as $|\mathcal{E}|/[n(n-1)]$).

•*Telegram* [3] is a pairwise influence network between $n = 245$ Telegram channels with $|\mathcal{E}| = 8,912$ directed edges. It is found in [3] that this network reveals a core-periphery structure in the sense of [5]. A directed core-periphery structure arises when there is a densely connected group of edges – a core – and sparsely connected groups of peripheral nodes with edges leading into the core, as well as sparsely connected groups of peripheral nodes with edges coming out of the core. Following [3] we assume $K = 4$ clusters, and the core-periphery structures gives $\beta = 5$.

•*Blog* [51] records $|\mathcal{E}| = 19,024$ directed edges between $n = 1,212$ political blogs from the 2004 US presidential election. In [51] it is found that there is an underlying structure with $K = 2$ clusters corresponding to the Republican and Democratic parties. Hence we choose $K = 2$ and $\beta = 1$.

•*Migration* [4] reports the number of people that migrated between pairs of counties in the US during 1995-2000. It involves $n = 3,075$ countries and $|\mathcal{E}| = 721,432$ directed edges after obtaining the largest weakly connected component. We choose $K = 10$ and $\beta = 9$, following [12]. Since the original digraph has extremely large entries, to cope with these outliers, we preprocess the input network by

$$\mathbf{A}_{i,j} = \frac{\mathbf{A}_{i,j}}{\mathbf{A}_{i,j} + \mathbf{A}_{j,i}} \mathbb{1}(\mathbf{A}_{i,j} > 0), \forall i, j \in \{1, \cdots, n\}, \tag{13}$$

which follows the preprocessing of [12]. The results for not doing this preprocessing is provided in Table 12.

•*WikiTalk* [52] contains all users and discussion from the inception of Wikipedia until Jan. 2008. The $n = 2,388,953$ nodes in the network represent Wikipedia users and a directed edge from node $v_i$ to node $v_j$ denotes that user $i$ edited at least once a talk page of user $j$. There are $|\mathcal{E}| = 5,018,445$ edges. We choose $K = 10$ clusters among candidates $\{2, 3, 5, 6, 8, 10\}$, and $\beta = 10$.

•*Lead-Lag* [20] contains yearly lead-lag matrices from 269 stocks from 2001 to 2019. We choose $K = 10$ clusters based on the GICS industry sectors [66], and choose $\beta = 3$ to emphasize the top three pairs of imbalance values. The lead-lag matrices are built from time series of daily price log returns, as detailed in [20]. The lead-lag metric for entry $(i, j)$ in the network encodes a measure of the extent to which stock $i$ leads stock $j$, and is obtained by applying a functional that computes the signed normalized area under the curve (auc) of the standard cross-correlation function (ccf). The resulting matrix is skew-symmetric, and entry $(i, j)$ quantifies the extent to which stock $i$ leads or lags stocks $j$, thus leading to a directed network interpretation. Starting from the skew-symmetric matrix, we further convert negative entries to zero, so that the resulting digraph can be directly fed

into other methods; note that this step does not throw away any information, and is pursued only to render the representation of the digraph consistent with the format expected by all methods compared, including DIGRAC. Note that the statistics given in Table 4 are averaged over the 19 years.

**Table 4:** Summary statistics for the real-world networks.

| data set | $n$ | $|\mathcal{E}|$ | density | weighted | $|\mathcal{E}^r|$ | $\frac{|\mathcal{E}^r|}{|\mathcal{E}|}(\%)$ |
|---|---|---|---|---|---|---|
| *Telegram* | 245 | 8,912 | $1.28 \cdot 10^{-2}$ | True | 1,572 | 17.64 |
| *Blog* | 1,222 | 19,024 | $1.49 \cdot 10^{-1}$ | True | 4,617 | 24.27 |
| *Migration* | 3,075 | 721,432 | $7.63 \cdot 10^{-2}$ | True | 351,100 | 48.67 |
| *WikiTalk* | 2,388,953 | 5,018,445 | $8.79 \cdot 10^{-7}$ | False | 723,526 | 14.42 |
| *Lead-Lag* | 269 | 29,159 | $4.04 \cdot 10^{-1}$ | True | 0.00 | 0.00 |

As input features, after obtaining eigenvectors from Hermitian matrices constructed as in [12], we standardize each column vector so that it has mean zero and variance one. We use these features for all GNN methods except MagNet, since MagNet has its own way of generating random features of dimension one.

### C.4 Hyperparameter Selection for DIMPA

We conduct hyperparmeter selection via a greedy search, for DIGRAC implemented with DIMPA as its aggregator. To explain the details, consider for example the following synthetic data setting: DSBM with 1000 nodes, 5 clusters, $\rho = 1$, and $p = 0.02$, without ambient nodes under different hyperparameter settings. By default, we use the loss function $\mathcal{L}_{\text{vol\_sum}}^{\text{sort}}$, $d = 32$ hidden units, hop $h = 2$, and no seed nodes. Instead of a grid search, we tune hyperparameters according to what performs the best in the default setting of the respective GNN method. The procedure starts with a random setting. For the next iteration, the hyperparameters are set to the current best setting (based on the last iteration), independently. For example, if we start with $a = 1, b = 2, c = 3$, and we find that under this default setting, the best $a$ (when fixing $b = 2, c = 3$) is 2 and the best $b$ (when fixing $a = 1, c = 3$) is 3, and the best $c$ is 3 (when fixing $a = 1, b = 2$), then for the next iteration, we set $a = 2, b = 3, c = 3$. If two settings give similar results, we choose the simpler setting, for example, the smaller hop size. When we reach a local optimum, we stop searching. Indeed, just a few iterations (less than five) were required for us to find the current setting, as DIGRAC tends to be robust to most hyperparameters.

Fig. 10, 11 and 12 are plots corresponding to the same setting but for three different meta-graph structures, namely the complete meta-graph structure, the cycle structure but with ambient nodes, and the complete structure with ambient nodes, respectively.

In theory, more hidden units give better expressive power. To reduce complexity, we use 32 hidden units throughout, which seems to have desirable performance. We observe that for low-noise regimes, more hidden units actually hurt performance. We can draw a similar conclusion about the hyperparameter selection. In terms of $\tau$, DIGRAC seems to be robust to different choices. Therefore, we use $\tau = 0.5$ throughout.

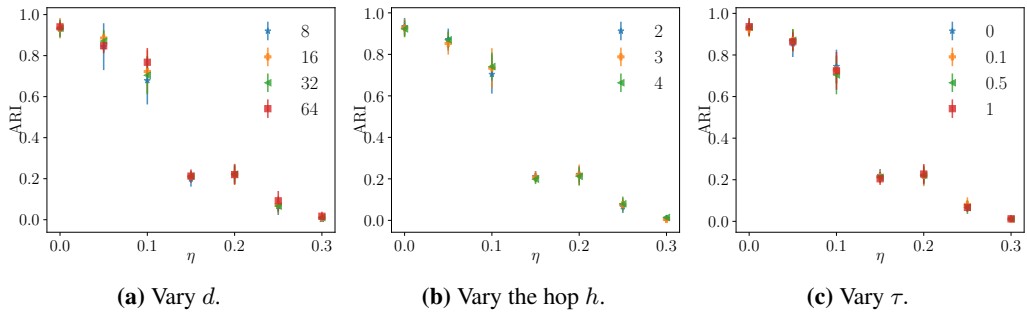

(a) Vary $d$.      (b) Vary the hop $h$.      (c) Vary $\tau$.

**Figure 10:** Hyperparameter analysis on different hyperparameter settings on the **complete** DSBM with 1000 nodes, 5 clusters, $\rho = 1$, and $p = 0.02$ **without** ambient nodes.

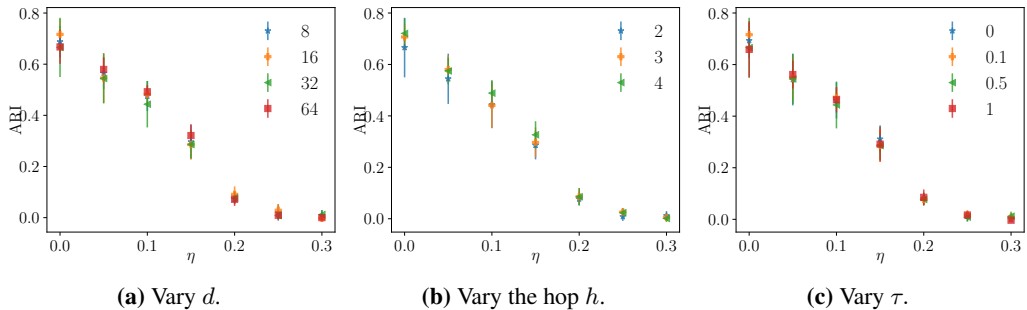

**(a)** Vary $d$.      **(b)** Vary the hop $h$.      **(c)** Vary $\tau$.

**Figure 11:** Hyperparameter analysis on different hyperparameter settings on the **complete** DSBM with 1000 nodes, 5 clusters, $\rho = 1$, and $p = 0.02$ **with** ambient nodes.

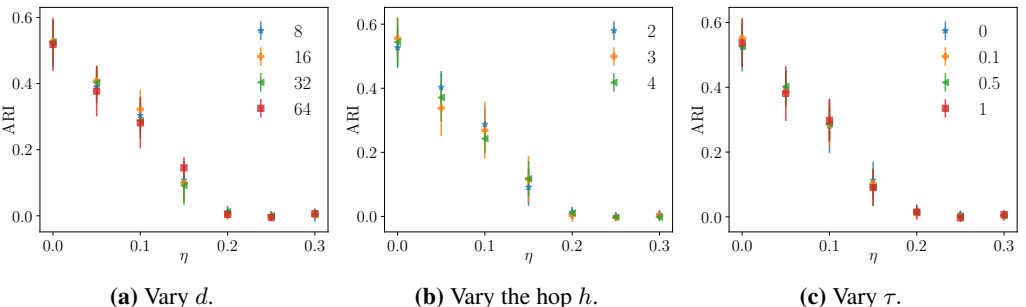

**(a)** Vary $d$.      **(b)** Vary the hop $h$.      **(c)** Vary $\tau$.

**Figure 12:** Hyperparameter analysis on different hyperparameter settings on the **cycle** DSBM with 1000 nodes, 5 clusters, $\rho = 1$, and $p = 0.02$ **with** ambient nodes.

### C.5 Use of Seed Nodes in a Semi-Supervised Manner

#### C.5.1 Supervised Loss

For seed nodes in $\mathcal{V}^{\text{seed}}$, similar to the loss function in [46], we use as a supervised loss function the sum of a cross-entropy loss and a triplet loss. The cross-entropy loss is given by

$$\mathcal{L}_{\text{CE}} = -\frac{1}{|\mathcal{V}^{\text{seed}}|} \sum_{v_i \in \mathcal{V}^{\text{seed}}} \sum_{k=1}^{K} \mathbb{1}(v_i \in \mathcal{C}_k) \log\left((\mathbf{p}_i)_k\right), \tag{14}$$

where $\mathbb{1}$ is the indicator function, $\mathcal{C}_k$ denotes the $k^{th}$ cluster, and $(\mathbf{p}_i)_k$ denotes the $k^{th}$ entry of probability vector $(\mathbf{p}_i)$. With the function $L : \mathbb{R}^2 \to \mathbb{R}$ given by $L(x, y) = [x - y]_+$ (where the subscript $+$ indicates taking the maximum of the expression value and 0), the triplet loss is defined as

$$\mathcal{L}_{\text{triplet}} = \frac{1}{|\mathcal{S}|} \sum_{(v_i, v_j, v_k) \in \mathcal{S}} L(\text{CS}(\mathbf{z}_i, \mathbf{z}_j), \text{CS}(\mathbf{z}_i, \mathbf{z}_k), \tag{15}$$

where $\mathcal{S} \subseteq \mathcal{V}^{\text{seed}} \times \mathcal{V}^{\text{seed}} \times \mathcal{V}^{\text{seed}}$ is a set of node triplets: $v_i$ is an anchor seed node, and $v_j$ is a seed node from the same cluster as the anchor, while $v_k$ is from a different cluster; and $\text{CS}(\mathbf{z}_i, \mathbf{z}_j)$ is the cosine similarity of the embeddings of nodes $v_i$ and $v_j$. We choose cosine similarity so as to avoid sensitivity to the magnitude of the embeddings. The triplet loss is designed so that, given two seed nodes from the same cluster and one seed node from a different cluster, the respective embeddings of the pairs from different clusters should be farther away than the embedding of the pair within the same cluster.

We then consider the weighted sum $\mathcal{L}_{\text{CE}} + \gamma_t \mathcal{L}_{\text{triplet}}$ as the supervised part of the loss function for DIGRAC, for some parameter $\gamma_t > 0$. The parameter $\gamma_t$ arises as follows. The cosine similarity between two randomly picked vectors in $d$ dimensions is bounded by $\sqrt{\ln(d)/d}$ with high probability. In our experiments $d = 32$, and $\sqrt{\ln(2d)/(2d)} \approx 0.25$. In contrast, for fairly uniform clustering, the cross-entropy loss grows like $\log n$, which in our experiments ranges between 3 and 17. Thus some balancing of the contribution is required. Following [46], we choose $\gamma_t = 0.1$ in our experiments.

### C.5.2 Overall Objective Function

By combining Eq. (14), Eq. (15), and Eq. (3)), our objective function for semi-supervised training with known seed nodes minimizes

$$\mathcal{L} = \mathcal{L}^{\text{sort}}_{\text{vol\_sum}} + \gamma_s(\mathcal{L}_{\text{CE}} + \gamma_t\mathcal{L}_{\text{triplet}}), \tag{16}$$

where $\gamma_s, \gamma_t > 0$ are weights for the supervised part of the loss and triplet loss within the supervised part, respectively. We set $\gamma_s = 50$ as we want our model to perform well on seed nodes. The weights could be tuned depending on how important each term is perceived to be.

### C.6 Training

For all synthetic data, we train DIGRAC with a maximum of 1000 epochs, and stop training when no gain in validation performance is achieved for 200 epochs (early-stopping). For real-world data, no "ground-truth" labels are available; we use all nodes to train and stop training when the training loss does not decrease for 200 epochs, or when we reach the maximum number of epochs, 1000.

When using the "std" variant for training, for the initial 50 epochs, we apply the "sort" variant with $\beta = 3$ for a reasonable starting clustering probability matrix for training, as otherwise during the initial training epochs possibly no pairs could be picked out. During the epochs actually utilizing this "std" variant, if no pairs could be picked out, we temporarily switch to the "naive" variant to count all the pairs for that epoch.

For the two-layer MLP, we do not have a bias term for each layer, and we use Rectified Linear Unit (ReLU) followed by a dropout layer with 0.5 dropout probability between the two layers, following [46]. We use Adam [67] as the optimizer and $\ell_2$ regularization with weight decay $5 \cdot 10^{-4}$ to avoid overfitting. We use as learning rate 0.01 throughout.

### C.7 Implementation Details for the Comparison Methods

In our experiments, we compare DIGRAC against five spectral methods, InfoMap, and four GNN-based supervised methods on synthetic data, and spectral methods and InfoMap on real data. The reason we are not able to compare DIGRAC with the other GNNs (namely, DGCN, DiGCN, MagNet, and DiGCL) on these data sets is due to the fact that these data sets do not have labels, which are required by the other GNN methods. We use the same hyperparameter settings stated in these papers. Data splits for all models are the same; the comparison GNNs are trained with 80% nodes under label supervision.

For MagNet, we use $q = 0.25$ for the phase matrix as in [33], because it is mentioned that $q = 0.25$ lays the most emphasis on directionality, which is our main focus in this paper. Code for MagNet is from `https://github.com/matthew-hirn/magnet`. For DiGCN, we use the code from `https://github.com/flyingtango/DiGCN/blob/main/code/digcn_ib.py` with option "adj_type" equals "ib". As a recommended option in [31], we use three layers for DiGCN. All other settings are the same as in the original paper [31]. Code for DiGCL is from `https://github.com/flyingtango/DiGCL`, where we adopt the settings for Cora_ML for hyperparameters.

### C.8 Enlarged Synthetic Result Figures

Figure 13 enlarges the results in the main text on synthetic data, with the same conclusions to be drawn.

### C.9 NMI Results Example and Reasons against Using NMI

As NMI is an often used measure for assessing similarities between partitions, Fig. 14 provides NMI results on some synthetic models mentioned in the main text. The results are qualitatively similar to the ARI results in Fig. 3.

We do not use NMI in the main text to evaluate results as NMI is known to suffer from finite size effects [48, 68]. In particular NMI prefers a larger number of partitions. Moreover it has been observed that the NMI between two independent partitions can be much larger than zero. This feature makes NMI more difficult to interpret than for example ARI.

# D  Additional Results on Real-World Data

## D.1  Extended Result Tables

Tables 5, 6, 7 and 8 provide a detailed comparison of DIGRAC with spectral methods and InfoMap. Since no labeling information is available and all of the other competing GNN methods require labels, we do not compare DIGRAC with them on these real data sets.

In Tables 5, 6, 7 and 8, we report 12 combinations of global imbalance scores by data set. The naming convention of these imbalance scores is provided in Table 2. To assess how balanced our recovered clusters are in terms of sizes, we also report the size ratio, which is defined as the size of the largest predicted cluster to the smallest one, and the standard deviation of sizes, size std, in order to show how varied the sizes of predicted clusters are. For a relatively balanced clustering, we expect the latter two terms to be small.

**Table 5:** Performance comparison on *Telegram*. The best is marked in **bold red** and the second best is marked in underline blue.

| Metric/Method | InfoMap | Bi_sym | DD_sym | DISG_LR | Herm | Herm_rw | DIGRAC |
|---|---|---|---|---|---|---|---|
| $\mathcal{O}^{\text{sort}}_{\text{vol\_sum}}$ | 0.04±0.00 | 0.21±0.00 | 0.21±0.00 | 0.21±0.01 | 0.20±0.01 | 0.14±0.00 | **0.32±0.01** |
| $\mathcal{O}^{\text{sort}}_{\text{vol\_min}}$ | 0.47±0.00 | 0.67±0.00 | 0.61±0.00 | 0.66±0.02 | 0.66±0.02 | 0.19±0.00 | **0.79±0.06** |
| $\mathcal{O}^{\text{sort}}_{\text{vol\_max}}$ | 0.03±0.00 | 0.20±0.00 | 0.20±0.00 | 0.20±0.01 | 0.19±0.01 | 0.12±0.00 | **0.29±0.01** |
| $\mathcal{O}^{\text{sort}}_{\text{plain}}$ | **1.00±0.00** | 0.80±0.00 | 0.75±0.00 | 0.78±0.03 | 0.76±0.04 | 0.59±0.00 | 0.96±0.01 |
| $\mathcal{O}^{\text{std}}_{\text{vol\_sum}}$ | 0.01±0.00 | 0.26±0.00 | 0.26±0.00 | 0.26±0.01 | 0.25±0.02 | **0.35±0.00** | 0.28±0.01 |
| $\mathcal{O}^{\text{std}}_{\text{vol\_min}}$ | 0.16±0.00 | **0.84±0.00** | 0.76±0.00 | 0.82±0.03 | 0.82±0.03 | 0.49±0.00 | 0.73±0.03 |
| $\mathcal{O}^{\text{std}}_{\text{vol\_max}}$ | 0.01±0.00 | 0.25±0.00 | 0.25±0.00 | 0.25±0.01 | 0.24±0.02 | **0.29±0.00** | 0.25±0.01 |
| $\mathcal{O}^{\text{std}}_{\text{plain}}$ | 0.68±0.00 | **1.00±0.00** | 0.94±0.00 | 0.98±0.04 | 0.95±0.04 | 0.99±0.00 | 0.90±0.05 |
| $\mathcal{O}^{\text{naive}}_{\text{vol\_sum}}$ | 0.01±0.00 | 0.26±0.00 | 0.26±0.00 | 0.26±0.01 | 0.25±0.02 | 0.23±0.00 | **0.27±0.01** |
| $\mathcal{O}^{\text{naive}}_{\text{vol\_min}}$ | 0.11±0.00 | **0.84±0.00** | 0.76±0.00 | 0.82±0.03 | 0.82±0.03 | 0.32±0.00 | 0.72±0.04 |
| $\mathcal{O}^{\text{naive}}_{\text{vol\_max}}$ | 0.00±0.00 | 0.25±0.00 | 0.25±0.00 | 0.25±0.01 | 0.24±0.02 | 0.20±0.00 | 0.24±0.01 |
| $\mathcal{O}^{\text{naive}}_{\text{plain}}$ | 0.63±0.00 | **1.00±0.00** | 0.94±0.00 | 0.98±0.04 | 0.95±0.04 | 0.99±0.00 | 0.89±0.06 |
| size ratio | 24.750 | 242.000 | 242.000 | 242.000 | 242.00 | 53 | **3.090** |
| size std | 35.57 | 104.360 | 104.360 | 104.360 | 104.360 | 63.460 | **26.39** |

**Table 6:** Performance comparison on *Blog*. The best is marked in **bold red** and the second best is marked in underline blue.

| Metric/Method | InfoMap | Bi_sym | DD_sym | DISG_LR | Herm | Herm_rw | DIGRAC |
|---|---|---|---|---|---|---|---|
| $\mathcal{O}^{\text{sort}}_{\text{vol\_sum}}$ | 0.07±0.00 | 0.07±0.00 | 0.00±0.00 | 0.05±0.00 | 0.37±0.00 | 0.00±0.00 | **0.44±0.00** |
| $\mathcal{O}^{\text{sort}}_{\text{vol\_min}}$ | 0.02±0.00 | 0.33±0.00 | 0.05±0.00 | 0.31±0.00 | 0.78±0.01 | **0.89±0.00** | 0.76±0.00 |
| $\mathcal{O}^{\text{sort}}_{\text{vol\_max}}$ | 0.05±0.00 | 0.05±0.00 | 0.00±0.00 | 0.04±0.00 | 0.26±0.00 | 0.00±0.00 | **0.40±0.00** |
| $\mathcal{O}^{\text{sort}}_{\text{plain}}$ | **1.00±0.00** | 0.33±0.00 | 0.05±0.00 | 0.31±0.00 | 0.78±0.01 | 0.89±0.00 | 0.76±0.00 |
| $\mathcal{O}^{\text{std}}_{\text{vol\_sum}}$ | 0.00±0.00 | 0.07±0.00 | 0.00±0.00 | 0.05±0.00 | 0.37±0.00 | 0.00±0.00 | **0.44±0.00** |
| $\mathcal{O}^{\text{std}}_{\text{vol\_min}}$ | 0.00±0.00 | 0.33±0.00 | 0.05±0.00 | 0.31±0.00 | 0.78±0.01 | **0.89±0.00** | 0.76±0.00 |
| $\mathcal{O}^{\text{std}}_{\text{vol\_max}}$ | 0.00±0.00 | 0.05±0.00 | 0.00±0.00 | 0.04±0.00 | 0.26±0.00 | 0.00±0.00 | **0.40±0.00** |
| $\mathcal{O}^{\text{std}}_{\text{plain}}$ | 0.73±0.00 | 0.33±0.00 | 0.05±0.00 | 0.31±0.00 | 0.78±0.01 | **0.89±0.00** | 0.76±0.00 |
| $\mathcal{O}^{\text{naive}}_{\text{vol\_sum}}$ | 0.00±0.00 | 0.07±0.00 | 0.00±0.00 | 0.05±0.00 | 0.37±0.00 | 0.00±0.00 | **0.44±0.00** |
| $\mathcal{O}^{\text{naive}}_{\text{vol\_min}}$ | 0.00±0.00 | 0.33±0.00 | 0.05±0.00 | 0.31±0.00 | 0.78±0.01 | **0.89±0.00** | 0.76±0.00 |
| $\mathcal{O}^{\text{naive}}_{\text{vol\_max}}$ | 0.00±0.00 | 0.05±0.00 | 0.00±0.00 | 0.04±0.00 | 0.26±0.00 | 0.00±0.00 | **0.40±0.00** |
| $\mathcal{O}^{\text{naive}}_{\text{plain}}$ | 0.76±0.00 | 0.33±0.00 | 0.05±0.00 | 0.31±0.00 | 0.78±0.01 | **0.89±0.00** | 0.76±0.00 |
| size ratio | **1.270** | 8.700 | 2.450 | 6.100 | 11.93 | 44.26 | 1.860 |
| size std | **64.50** | 485 | 256.200 | 439 | 516.500 | 584 | 183.20 |

Tables 5, 6, 7, 8, 9, 10 and 11 reveal that DIGRAC provides competitive global imbalance scores in all of the 12 objectives introduced, and across all the real data sets, usually outperforming all the other methods. Among the tables, Table 11 provides results in terms of the distance to the best yearly performance, averaged across the 19 years; DIGRAC usually outperforms all the other methods across all the years. Note that Bi_sym and DD_sym are not able to generate results for *WikiTalk*, as

**Table 7:** Performance comparison on *Migration*. The best is marked in **bold red** and the second best is marked in underline blue. InfoMap results are omitted here as it predicts a single huge cluster and could not generate imbalance results.

| Metric/Method | Bi_sym | DD_sym | DISG_LR | Herm | Herm_rw | DIGRAC |
|---|---|---|---|---|---|---|
| $\mathcal{O}^{sort}_{vol\_sum}$ | 0.03±0.00 | 0.01±0.00 | 0.02±0.00 | 0.04±0.00 | 0.02±0.00 | **0.05±0.00** |
| $\mathcal{O}^{sort}_{vol\_min}$ | **0.19±0.00** | 0.08±0.00 | 0.08±0.00 | 0.15±0.02 | 0.05±0.00 | 0.18±0.03 |
| $\mathcal{O}^{sort}_{vol\_max}$ | 0.03±0.00 | 0.01±0.00 | 0.01±0.00 | 0.03±0.00 | 0.02±0.00 | **0.04±0.00** |
| $\mathcal{O}^{sort}_{plain}$ | 0.24±0.00 | 0.20±0.00 | 0.17±0.00 | 0.40±0.01 | **0.49±0.06** | 0.29±0.04 |
| $\mathcal{O}^{std}_{vol\_sum}$ | 0.01±0.00 | 0.01±0.00 | 0.01±0.00 | 0.02±0.00 | 0.02±0.00 | **0.04±0.01** |
| $\mathcal{O}^{std}_{vol\_min}$ | 0.10±0.00 | 0.05±0.00 | 0.05±0.00 | 0.08±0.01 | 0.04±0.00 | **0.16±0.03** |
| $\mathcal{O}^{std}_{vol\_max}$ | 0.01±0.00 | 0.01±0.00 | 0.01±0.00 | 0.01±0.00 | 0.01±0.00 | **0.03±0.01** |
| $\mathcal{O}^{std}_{plain}$ | 0.13±0.00 | 0.12±0.00 | 0.11±0.00 | 0.20±0.01 | 0.20±0.01 | **0.26±0.01** |
| $\mathcal{O}^{naive}_{vol\_sum}$ | 0.01±0.00 | 0.01±0.00 | 0.01±0.00 | 0.02±0.00 | 0.01±0.00 | **0.04±0.01** |
| $\mathcal{O}^{naive}_{vol\_min}$ | 0.09±0.00 | 0.04±0.00 | 0.04±0.00 | 0.08±0.01 | 0.01±0.00 | **0.16±0.03** |
| $\mathcal{O}^{naive}_{vol\_max}$ | 0.01±0.00 | 0.00±0.00 | 0.01±0.00 | 0.01±0.00 | 0.00±0.00 | **0.03±0.01** |
| $\mathcal{O}^{naive}_{plain}$ | 0.12±0.00 | 0.10±0.00 | 0.08±0.00 | 0.19±0.00 | 0.19±0.03 | **0.26±0.01** |
| size ratio | 7.780 | 6.070 | **4.360** | 36.05 | 1035.90 | 4.420 |
| size std | 135.210 | 132.76 | **103.43** | 335.790 | 353.060 | 264.500 |

**Table 8:** Performance comparison on *WikiTalk*. The best is marked in **bold red** and the second best is marked in underline blue. InfoMap results are omitted here as its large number of predicted clusters leads to memory error in imbalance calculation.

| Metric/Method | DISG_LR | Herm | Herm_rw | DIGRAC |
|---|---|---|---|---|
| $\mathcal{O}^{sort}_{vol\_sum}$ | 0.18±0.03 | 0.15±0.02 | 0.00±0.00 | **0.24±0.05** |
| $\mathcal{O}^{sort}_{vol\_min}$ | 0.10±0.03 | 0.22±0.05 | 0.26±0.00 | **0.28±0.13** |
| $\mathcal{O}^{sort}_{vol\_max}$ | 0.16±0.03 | 0.09±0.01 | 0.00±0.00 | **0.19±0.04** |
| $\mathcal{O}^{sort}_{plain}$ | 0.87±0.08 | 0.99±0.01 | 0.98±0.00 | **1.00±0.00** |
| $\mathcal{O}^{std}_{vol\_sum}$ | **0.17±0.04** | 0.06±0.01 | 0.01±0.00 | 0.14±0.02 |
| $\mathcal{O}^{std}_{vol\_min}$ | 0.09±0.02 | 0.09±0.02 | **0.27±0.00** | 0.18±0.08 |
| $\mathcal{O}^{std}_{vol\_max}$ | **0.15±0.04** | 0.04±0.00 | 0.00±0.00 | 0.11±0.02 |
| $\mathcal{O}^{std}_{plain}$ | 0.72±0.03 | 0.70±0.05 | **0.98±0.00** | 0.84±0.06 |
| $\mathcal{O}^{naive}_{vol\_sum}$ | 0.10±0.02 | 0.04±0.00 | 0.00±0.00 | **0.12±0.01** |
| $\mathcal{O}^{naive}_{vol\_min}$ | 0.06±0.03 | 0.07±0.02 | **0.26±0.00** | 0.15±0.07 |
| $\mathcal{O}^{naive}_{vol\_max}$ | **0.09±0.02** | 0.03±0.00 | 0.00±0.00 | **0.09±0.01** |
| $\mathcal{O}^{naive}_{plain}$ | 0.64±0.04 | 0.61±0.04 | **0.98±0.00** | 0.76±0.06 |
| size ratio | 1190162.25 | 2217434.50 | **250.48** | 71765.14 |
| size std | 713813.72 | 660060.33 | 657941.88 | **643220.37** |

large $n \times n$ matrix multiplication with its transpose causes memory issue, when $n = 2,388,953$. Small values of the size ratio and size standard deviation suggest that the normalization in the loss function penalizes tiny clusters, and that DIGRAC tends to predict balanced cluster sizes.

**Table 9:** Performance comparison on *Lead-Lag* for year 2015. The best is marked in **bold red** and the second best is marked in underline blue. InfoMap results are omitted here as it usually predicts a single huge cluster and could not generate imbalance results.

| Metric/Method | Bi_sym | DD_sym | DISG_LR | Herm | Herm_rw | DIGRAC |
|---|---|---|---|---|---|---|
| $\mathcal{O}^{sort}_{vol\_sum}$ | 0.07±0.00 | 0.07±0.00 | 0.06±0.00 | 0.07±0.00 | 0.06±0.01 | **0.15±0.00** |
| $\mathcal{O}^{sort}_{vol\_min}$ | **0.53±0.06** | 0.50±0.02 | 0.45±0.07 | 0.50±0.03 | 0.46±0.06 | 0.50±0.02 |
| $\mathcal{O}^{sort}_{vol\_max}$ | 0.07±0.00 | 0.06±0.00 | 0.06±0.00 | 0.06±0.00 | 0.06±0.00 | **0.15±0.01** |
| $\mathcal{O}^{sort}_{plain}$ | 0.65±0.03 | **0.67±0.03** | 0.59±0.03 | 0.65±0.03 | 0.65±0.02 | 0.55±0.07 |
| $\mathcal{O}^{std}_{vol\_sum}$ | 0.04±0.00 | 0.04±0.00 | 0.04±0.00 | 0.04±0.00 | 0.04±0.00 | **0.11±0.02** |
| $\mathcal{O}^{std}_{vol\_min}$ | 0.27±0.03 | 0.27±0.02 | 0.24±0.02 | 0.27±0.02 | 0.26±0.04 | **0.35±0.04** |
| $\mathcal{O}^{std}_{vol\_max}$ | 0.03±0.00 | 0.03±0.00 | 0.03±0.00 | 0.03±0.00 | 0.03±0.00 | **0.10±0.02** |
| $\mathcal{O}^{std}_{plain}$ | 0.39±0.02 | 0.39±0.01 | 0.37±0.02 | 0.39±0.02 | **0.40±0.02** | 0.38±0.04 |
| $\mathcal{O}^{naive}_{vol\_sum}$ | 0.03±0.00 | 0.03±0.00 | 0.03±0.00 | 0.03±0.00 | 0.03±0.00 | **0.08±0.03** |
| $\mathcal{O}^{naive}_{vol\_min}$ | 0.20±0.02 | 0.20±0.02 | 0.17±0.03 | 0.20±0.02 | 0.20±0.03 | **0.25±0.08** |
| $\mathcal{O}^{naive}_{vol\_max}$ | 0.02±0.00 | 0.03±0.00 | 0.02±0.00 | 0.03±0.00 | 0.03±0.00 | **0.08±0.03** |
| $\mathcal{O}^{naive}_{plain}$ | 0.29±0.01 | 0.29±0.01 | 0.26±0.02 | 0.30±0.01 | 0.30±0.01 | **0.31±0.05** |
| size ratio | 3.070 | 3.110 | 3.060 | **2.89** | 2.95 | 15.640 |
| size std | 8.390 | 7.94 | 8.680 | **7.28** | 8.050 | 18.680 |

**Table 10:** Performance comparison on *Lead-Lag*. Results in each year is averaged over ten runs. Mean and standard deviation (after ±) are calculated over the 19 years. The best is marked in **bold red** and the second best is marked in underline blue. InfoMap results are omitted here as it usually predicts a single huge cluster and could not generate imbalance results.

| Metric/Method | Bi_sym | DD_sym | DISG_LR | Herm | Herm_rw | DIGRAC |
|---|---|---|---|---|---|---|
| $\mathcal{O}^{sort}_{vol\_sum}$ | 0.07±0.01 | 0.07±0.01 | 0.07±0.01 | 0.07±0.02 | 0.07±0.02 | **0.15±0.03** |
| $\mathcal{O}^{sort}_{vol\_min}$ | **0.51±0.10** | 0.48±0.09 | 0.47±0.10 | **0.51±0.11** | 0.50±0.10 | 0.47±0.09 |
| $\mathcal{O}^{sort}_{vol\_max}$ | 0.07±0.01 | 0.06±0.01 | 0.06±0.01 | 0.07±0.01 | 0.07±0.01 | **0.14±0.03** |
| $\mathcal{O}^{sort}_{plain}$ | **0.66±0.09** | 0.64±0.08 | 0.63±0.08 | **0.66±0.09** | 0.65±0.09 | 0.53±0.09 |
| $\mathcal{O}^{std}_{vol\_sum}$ | 0.04±0.01 | 0.04±0.01 | 0.04±0.01 | 0.04±0.01 | 0.04±0.01 | **0.12±0.03** |
| $\mathcal{O}^{std}_{vol\_min}$ | 0.27±0.04 | 0.27±0.04 | 0.25±0.04 | 0.27±0.03 | 0.27±0.03 | **0.38±0.07** |
| $\mathcal{O}^{std}_{vol\_max}$ | 0.04±0.00 | 0.03±0.00 | 0.03±0.00 | 0.03±0.00 | 0.03±0.00 | **0.11±0.02** |
| $\mathcal{O}^{std}_{plain}$ | 0.40±0.05 | 0.39±0.05 | 0.38±0.05 | 0.40±0.05 | 0.40±0.05 | **0.44±0.07** |
| $\mathcal{O}^{naive}_{vol\_sum}$ | 0.03±0.01 | 0.03±0.01 | 0.03±0.01 | 0.03±0.01 | 0.03±0.01 | **0.08±0.04** |
| $\mathcal{O}^{naive}_{vol\_min}$ | 0.20±0.05 | 0.19±0.05 | 0.18±0.05 | 0.19±0.04 | 0.19±0.04 | **0.26±0.10** |
| $\mathcal{O}^{naive}_{vol\_max}$ | 0.03±0.01 | 0.02±0.01 | 0.02±0.01 | 0.02±0.00 | 0.02±0.00 | **0.08±0.03** |
| $\mathcal{O}^{naive}_{plain}$ | 0.30±0.06 | 0.28±0.06 | 0.27±0.06 | 0.29±0.05 | 0.29±0.05 | **0.32±0.11** |
| size ratio | 3.67 | **3.34** | 3.900 | 4.110 | 3.880 | 8.070 |
| size std | 9.31 | **9.14** | 10.090 | 10.490 | 10.360 | 17.060 |

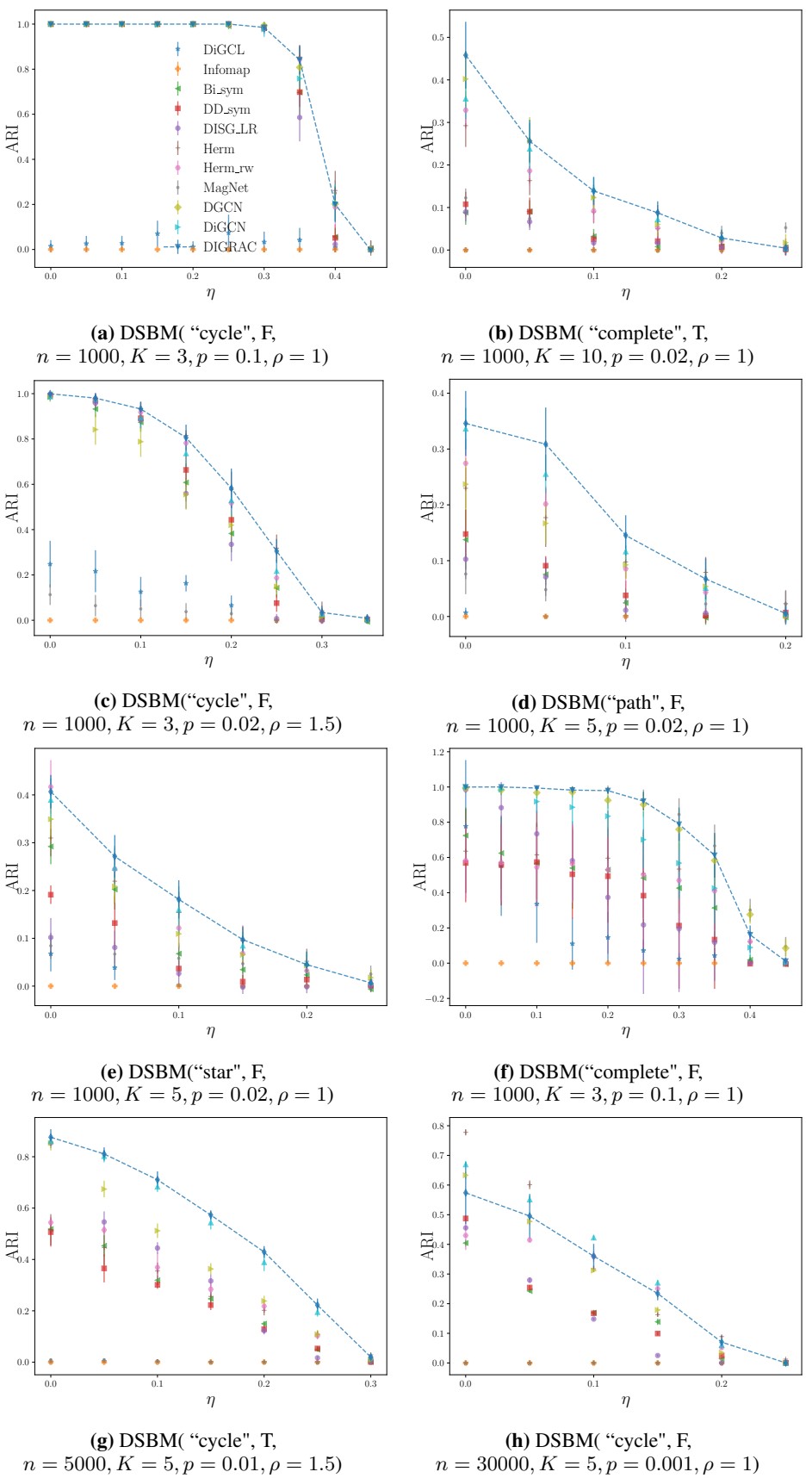

**(a)** DSBM( "cycle", F, $n = 1000, K = 3, p = 0.1, \rho = 1$)

**(b)** DSBM( "complete", T, $n = 1000, K = 10, p = 0.02, \rho = 1$)

**(c)** DSBM("cycle", F, $n = 1000, K = 3, p = 0.02, \rho = 1.5$)

**(d)** DSBM("path", F, $n = 1000, K = 5, p = 0.02, \rho = 1$)

**(e)** DSBM("star", F, $n = 1000, K = 5, p = 0.02, \rho = 1$)

**(f)** DSBM("complete", F, $n = 1000, K = 3, p = 0.1, \rho = 1$)

**(g)** DSBM( "cycle", T, $n = 5000, K = 5, p = 0.01, \rho = 1.5$)

**(h)** DSBM( "cycle", F, $n = 30000, K = 5, p = 0.001, \rho = 1$)

**Figure 13:** Test ARI comparison on synthetic data. Dashed lines highlight DIGRAC's performance. Error bars are given by one standard error.

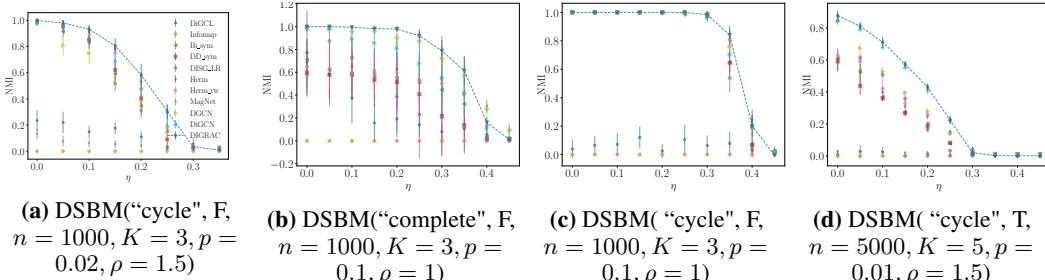

**(a)** DSBM("cycle", F, $n = 1000, K = 3, p = 0.02, \rho = 1.5$)

**(b)** DSBM("complete", F, $n = 1000, K = 3, p = 0.1, \rho = 1$)

**(c)** DSBM( "cycle", F, $n = 1000, K = 3, p = 0.1, \rho = 1$)

**(d)** DSBM( "cycle", T, $n = 5000, K = 5, p = 0.01, \rho = 1.5$)

**Figure 14:** Test NMI comparison on some synthetic data. Dashed lines highlight DIGRAC's performance. Error bars are given by one standard error.

**Table 11:** Performance comparison on *Lead-Lag*, where we evaluate the performance distance to the best one in each year. Results in each year is averaged over ten runs. Mean and standard deviation (after $\pm$) are calculated over the 19 years. The best is marked in **bold red** and the second best is marked in underline blue. InfoMap results are omitted here as it usually predicts a single huge cluster and could not generate imbalance results.

| Metric/Method | Bi_sym | DD_sym | DISG_LR | Herm | Herm_rw | DIGRAC |
|---|---|---|---|---|---|---|
| $\mathcal{O}_{\text{vol\_sum}}^{\text{sort}}$ | 0.07±0.02 | 0.08±0.02 | 0.08±0.02 | 0.07±0.02 | 0.07±0.02 | **0.00±0.00** |
| $\mathcal{O}_{\text{vol\_min}}^{\text{sort}}$ | **0.01±0.01** | 0.05±0.03 | 0.06±0.03 | 0.02±0.02 | 0.02±0.02 | 0.06±0.04 |
| $\mathcal{O}_{\text{vol\_max}}^{\text{sort}}$ | 0.07±0.02 | 0.07±0.02 | 0.07±0.02 | 0.07±0.02 | 0.07±0.02 | **0.00±0.00** |
| $\mathcal{O}_{\text{plain}}^{\text{sort}}$ | **0.01±0.02** | 0.03±0.03 | 0.05±0.03 | **0.01±0.02** | 0.02±0.02 | 0.14±0.03 |
| $\mathcal{O}_{\text{vol\_sum}}^{\text{std}}$ | 0.08±0.02 | 0.08±0.02 | 0.08±0.02 | 0.08±0.02 | 0.08±0.02 | **0.00±0.00** |
| $\mathcal{O}_{\text{vol\_min}}^{\text{std}}$ | 0.10±0.05 | 0.11±0.04 | 0.13±0.05 | 0.11±0.05 | 0.11±0.05 | **0.00±0.00** |
| $\mathcal{O}_{\text{vol\_max}}^{\text{std}}$ | 0.07±0.02 | 0.08±0.02 | 0.08±0.02 | 0.08±0.02 | 0.08±0.02 | **0.00±0.00** |
| $\mathcal{O}_{\text{plain}}^{\text{std}}$ | 0.04±0.03 | 0.05±0.04 | 0.06±0.04 | 0.04±0.04 | 0.04±0.03 | **0.00±0.00** |
| $\mathcal{O}_{\text{vol\_sum}}^{\text{naive}}$ | 0.05±0.03 | 0.06±0.03 | 0.06±0.03 | 0.05±0.03 | 0.05±0.03 | **0.00±0.00** |
| $\mathcal{O}_{\text{vol\_min}}^{\text{naive}}$ | 0.06±0.07 | 0.07±0.06 | 0.08±0.07 | 0.07±0.08 | 0.07±0.08 | **0.00±0.00** |
| $\mathcal{O}_{\text{vol\_max}}^{\text{naive}}$ | 0.05±0.03 | 0.05±0.03 | 0.05±0.03 | 0.05±0.03 | 0.05±0.03 | **0.00±0.00** |
| $\mathcal{O}_{\text{plain}}^{\text{naive}}$ | 0.03±0.06 | 0.05±0.05 | 0.06±0.06 | 0.04±0.06 | 0.04±0.06 | **0.01±0.02** |
| size ratio | 1.04 | **0.71** | 1.270 | 1.480 | 1.250 | 5.440 |
| size std | 0.58 | **0.41** | 1.360 | 1.770 | 1.630 | 8.340 |

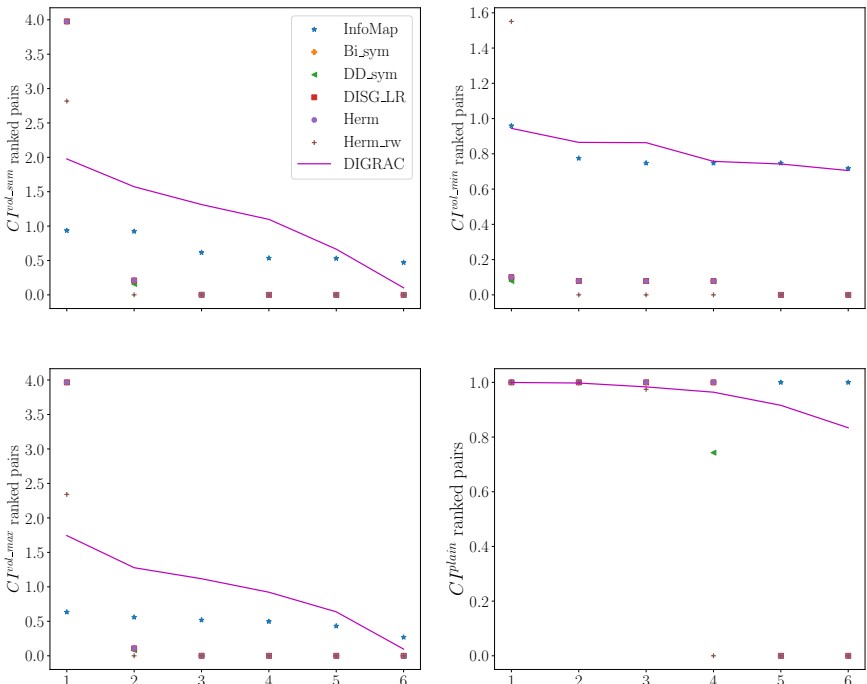

**Figure 15:** Ranked pairs of pairwise imbalance recovered by comparing methods for different choices of normalization on the *Telegram* data set. Lines are used to highlight DIGRAC's performance.

## D.2   Ranked Pairwise Imbalance Scores

We also plot the ranked pairwise imbalance scores for all data sets except *Blog*, which has only one possible pairwise imbalance score. For *Lead-Lag*, we only plot the year 2015 as an example; the plots for the other years are similar. Figures 15, 16, 17 and 18 illustrate that DIGRAC is able to provide comparable or higher pairwise imbalance scores for the leading pairs, especially on CI$^{\text{vol\_min}}$ pairs. We also observe that except for CI$^{\text{plain}}$, DIGRAC has a less rapid drop in pairwise imbalance scores after the first leading pair compared to Herm and Herm_rw, which can have a few pairs with higher imbalance scores than DIGRAC.

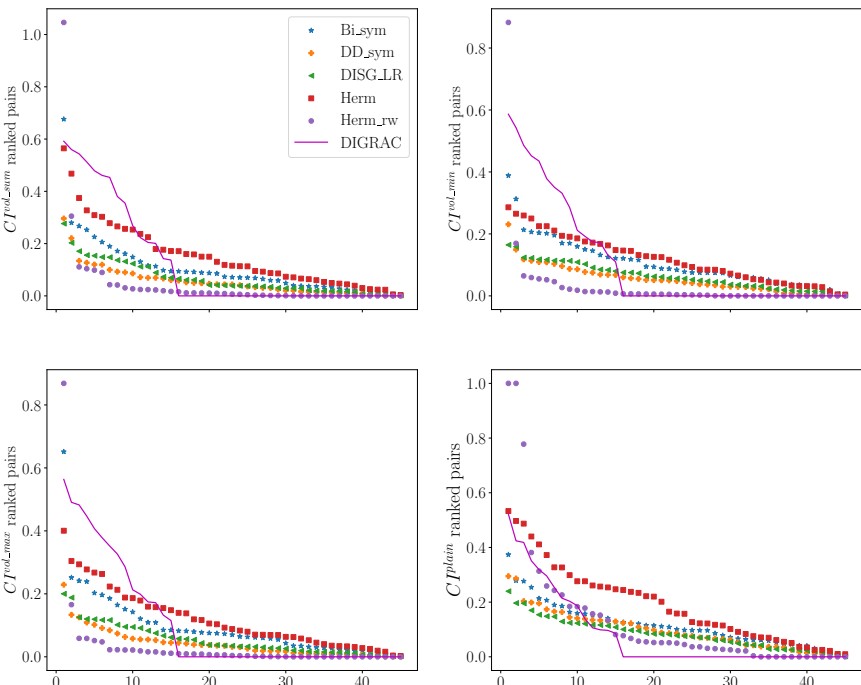

**Figure 16:** Ranked pairs of pairwise imbalance recovered by comparing methods for different choices of normalization on the *Migration* data set. Lines are used to highlight DIGRAC's performance. InfoMap results are omitted as it predicts one single huge cluster and could not produce imbalance results.

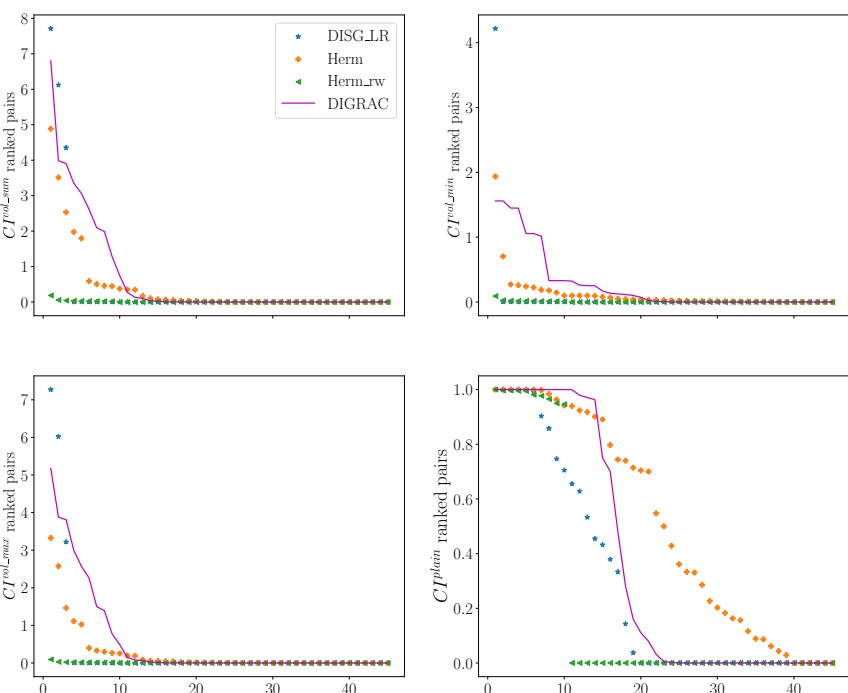

**Figure 17:** Ranked pairs of pairwise imbalance recovered by comparing methods for different choices of normalization on *WikiTalk* data set. Lines are used to highlight DIGRAC's performance. InfoMap results are omitted here because it triggers memory error due to the large number of predicted clusters.

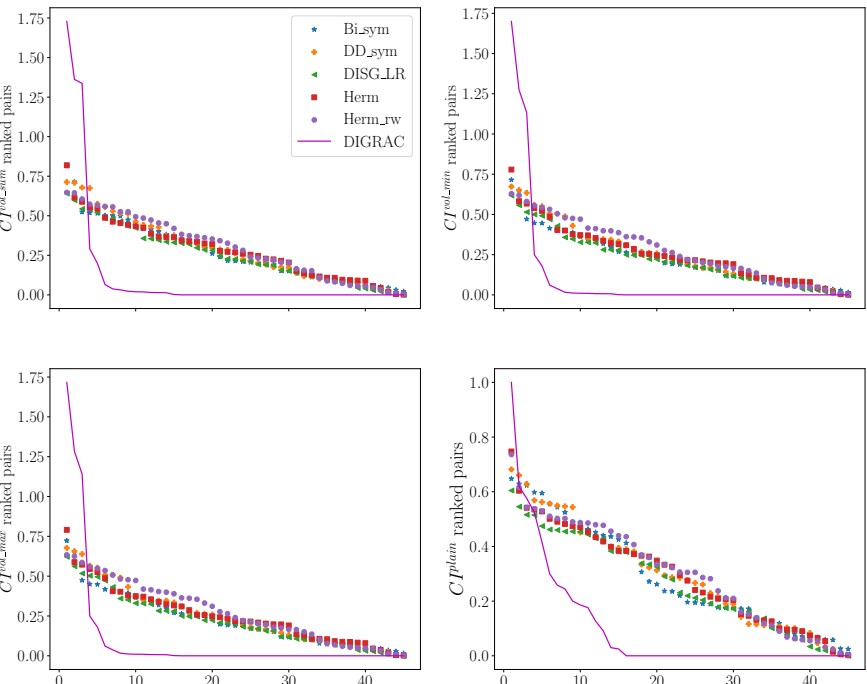

**Figure 18:** Ranked pairs of pairwise imbalance recovered by comparing methods for different choices of normalization on *Lead-Lag* data set. Lines are used to highlight DIGRAC's performance. InfoMap results are omitted here because it only predicts a single cluster.

### D.3 Predicted Meta-Graph Flow Matrix Plots

For each data set, we plot the predicted meta-graph flow matrix $\mathbf{F}'$ defined in Eq. (12).

From Fig. 19, we conclude that DIGRAC is able to recover a directed flow imbalance between clusters in all of the selected data sets. Fig. 19a shows a clear cut imbalance between two clusters, possibly corresponding to the Republican and Democratic parties. Fig. 19b plots imbalance flows in the real data set *Telegram*, where cluster 3 is a core-transient cluster, cluster 0 is a core-sink cluster, cluster 2 is a periphery-upstream cluster, while cluster 1 is a periphery-downstream cluster [3, 5]. For *WikiTalk*, illustrated in Fig. 19d, the lower-triangular part entries are typically source nodes for edges, while the upper-triangular part are target nodes. For *Lead-Lag*, taking the year 2015 as an example, DIGRAC is also able to recover high imbalance in the data.

We also note that DIGRAC would not necessarily predict the same number of clusters as assumed, so that we do not need to specify the exact number of clusters before training DIGRAC; specifying the maximum number of possible clusters suffices.

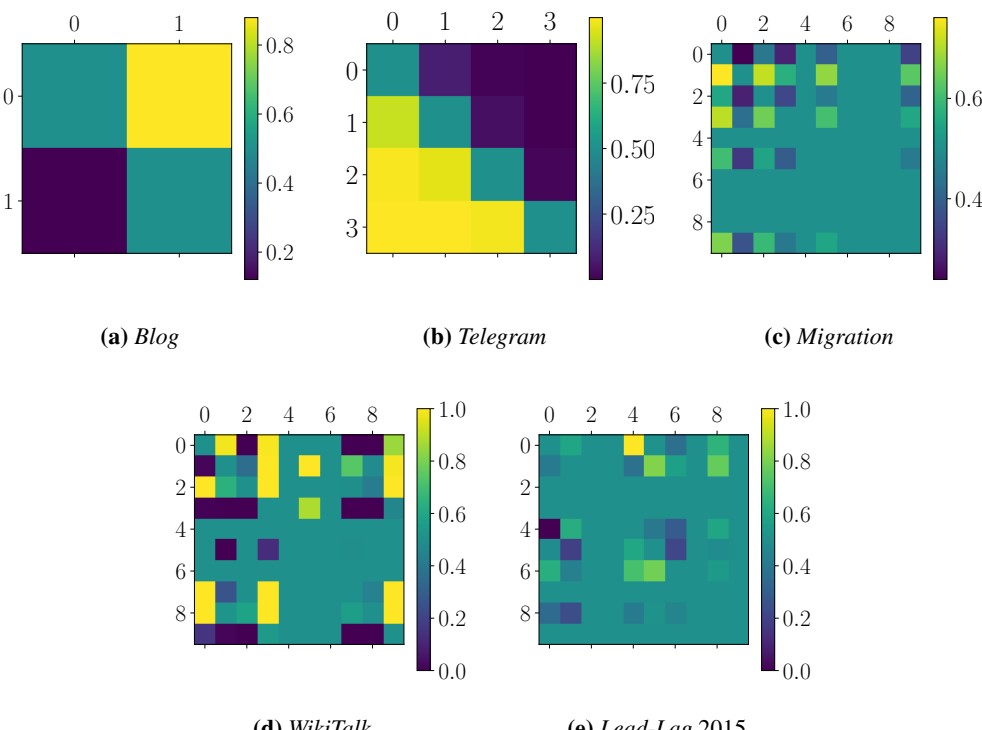

**Figure 19:** Predicted meta-graph flow matrix from DIGRAC of five real-world data sets.

## D.4 Migration Plots

We compare DIGRAC to five spectral methods for recovering clusters for the US migration data set, and plot the recovered clusters on a map, in Fig. 20.

The visualization in Fig. (a-c) shows that clusters align particularly well with the political and administrative boundaries of the US states, as previously observed in [69]. This outcome is not deemed too insightful, as it trivially reveals the fact there there is significant intra-state and inter-state migration, and does not uncover any of the information on latent migration patterns between far-away states, and more generally, between regions which are not necessarily geographically cohesive. DIGRAC outcomes, however, reveal nontrivial migration patterns, for example migration from New York to Florida, and from California to Arizona, which is consistent with the patterns discovered by [4]. Fig. 21 details on the top pair migration patterns uncovered by DIGRAC.

## D.5 Coping with Outliers

As mentioned in Section C.3, the preprocessing step to use ratio of migration instead of absolute migration numbers is a way to cope with outliers (here, extremely large entries in the original digraph) in *Migration*. To validate the effectiveness of this approach to cope with outliers, Table 12 provides imbalance results for *Migration* when we do not transform the nonzero entries into ratios. Comparing with Table 7, we witness an overall decrease in the performance. In this case InfoMap no longer predicts a single huge cluster. However, its predicted number of clusters is about 44, which is too large. This also implies that InfoMap is very sensitive to the magnitude of digraph entries, while DIGRAC is not. Indeed, InfoMap gives 43 (too many) clusters for *Blog*, 19 (too many) for *Telegram*, 1 (too small) for *Migration*, and 17498 (far too many) for *WikiTalk*.

We compare DIGRAC to five spectral methods as well as InfoMap for recovering clusters for the US migration data set without the preprocessing step discussed earlier, and plot the recovered clusters on a map in Fig. 22. Note that all methods, except DIGRAC, recover either clusters which are trivially small in size or contain one very large dominant cluster (as in (a), (b), (e) and to some extent, also

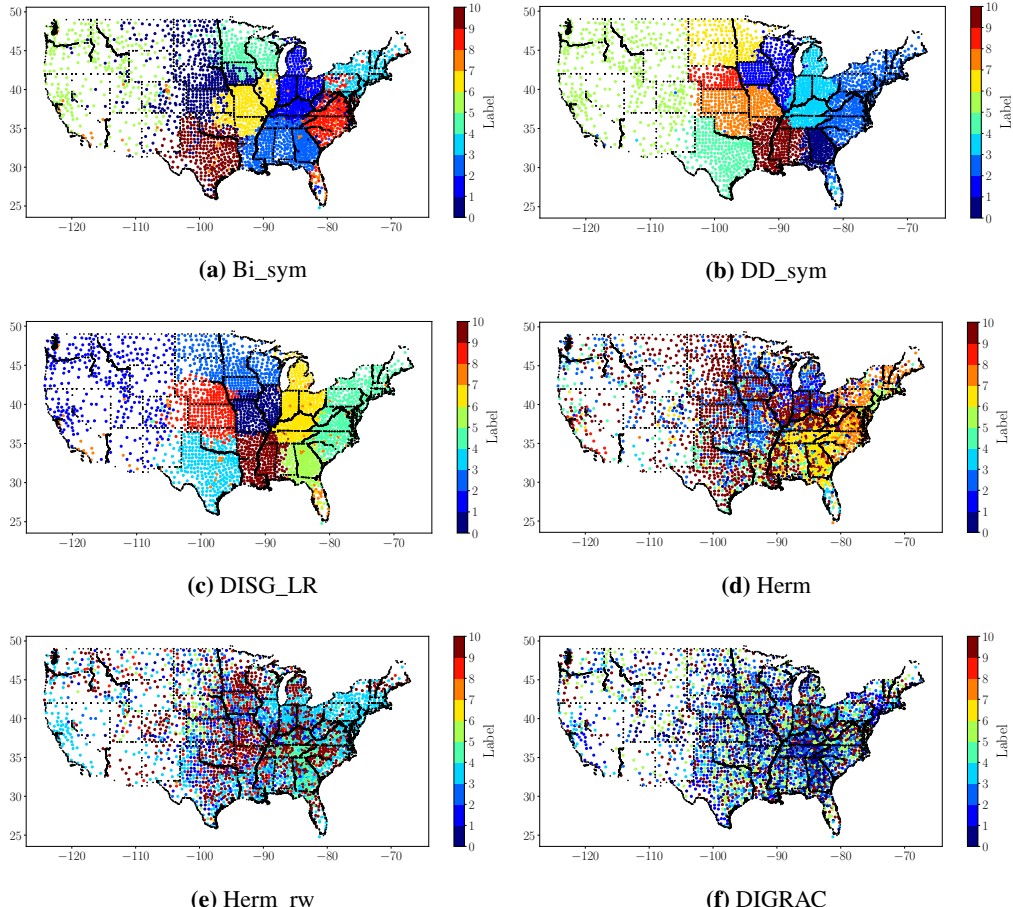

**Figure 20:** US migration predicted clusters, along with the geographic locations of the counties as well as state boundaries (in black). InfoMap results are omitted here because it only produces one huge cluster. The input data is normalized, following Eq. (13).

(f)). The DISG_LR clustering and InfoMap clustering provide clear geographic boundaries, but were not able to recover the imbalance among clusters. Other spectral methods generally have a dominant cluster containing most of the nodes, whereas DIGRAC has more balanced cluster sizes.

When employing methods that symmetrize the adjacency matrix (as in (a) and (b)), the migration flows between counties in different states will be lost in the process. Furthermore, the visualization in Fig. (c) shows that clusters align particularly well with the political and administrative boundaries of the US states, as previously observed in [69]. The same is for Fig. (d). This outcome is not deemed very insightful, as it trivially reveals the fact that there is significant intra-state and inter-state migration, and does not uncover any of the information on latent migration patterns between far-away states, and more generally, between regions which are not necessarily geographically cohesive.

Fig. 21 further plots the top three pairs of clusters based on four different imbalance scores given by DIGRAC. As shown in the figure, DIGRAC uncovers the migration trend from coastal to interior, across states. This trend of the directed flow agrees with that discussed in [4], with many people migrating from New York and California to the interior states.

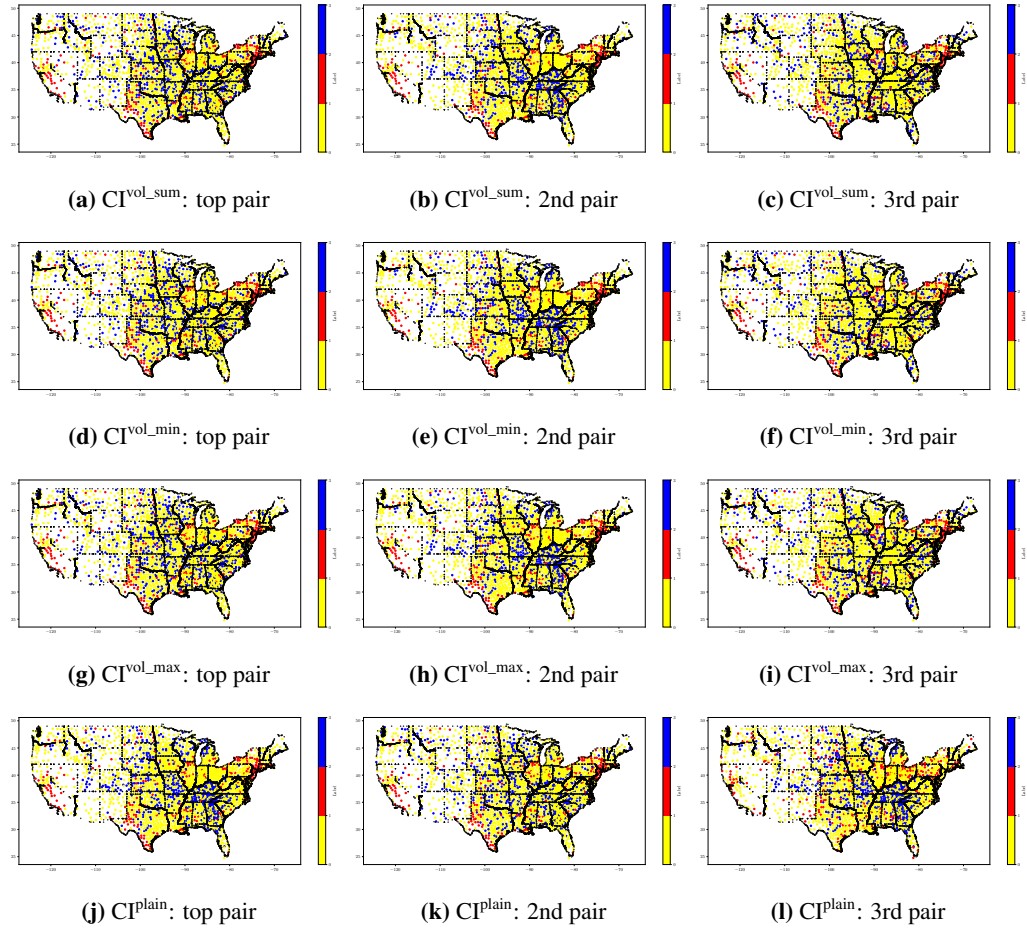

**(a)** CI$^{\text{vol\_sum}}$: top pair  **(b)** CI$^{\text{vol\_sum}}$: 2nd pair  **(c)** CI$^{\text{vol\_sum}}$: 3rd pair

**(d)** CI$^{\text{vol\_min}}$: top pair  **(e)** CI$^{\text{vol\_min}}$: 2nd pair  **(f)** CI$^{\text{vol\_min}}$: 3rd pair

**(g)** CI$^{\text{vol\_max}}$: top pair  **(h)** CI$^{\text{vol\_max}}$: 2nd pair  **(i)** CI$^{\text{vol\_max}}$: 3rd pair

**(j)** CI$^{\text{plain}}$: top pair  **(k)** CI$^{\text{plain}}$: 2nd pair  **(l)** CI$^{\text{plain}}$: 3rd pair

**Figure 21:** US migration predicted cluster pairs with top imbalance, along with the geographic locations of the counties as well as state boundaries (in black). Red (label 1) is the sending cluster while blue (label 2) is the receiving cluster. Yellow (label 0) denotes all the other locations being considered. Subcaptions show the imbalance score and the rank based on that score.

## E  Discussion of Related Methods that are not Compared against in the Main Text

To further emphasize the importance of directionality, our synthetic data sets have no difference in density between clusters; their sole signal is in the directionality of the edges. If all edge directions were to be removed, then no algorithm should be available to detect the clusters. To further support our claim why some methods mentioned in Section 2 in the main text are not appropriate for comparison, we have applied the default setting versions of the Louvain method [39], the Leiden algorithm [40] and OSLOM [38], to our synthetic data sets, and find that they do not detect the structure in the data, with ARI and NMI values very close to zero, and very low imbalance values. In particular, Louvain and Leiden tend to give a larger number of clusters than the ground truth which is designed to have small cluster sizes. OSLOM outputs clusters with extreme sizes, either a huge cluster containing (almost always) all the nodes, or every node forming a cluster by itself. To further demonstrate that comparing DIGRAC with density-based methods is unfair, We report the test ARI results for Infomap, Louvain and Leiden in Figure 23.We can see that Infomap, Louvain and Leiden normally produces nero-zero ARI values, which are much worse than the results from DIGRAC given in Figure 3.

On the real-world data sets, these methods often give numbers of clusters that do not match our expectations. (*Blog* has two underlying parties, *Telegram* has a four-cluster core-periphery structure). Louvain clusters nodes from *Blog* into 8-13 clusters (too many), *Telegram* into 4-5 clusters

**Table 12:** Performance comparison on *Migration* (without preprocessing). The best is marked in **bold red** and the second best is marked in underline blue.

| Metric/Method | InfoMap | Bi_sym | DD_sym | DISG_LR | Herm | Herm_rw | DIGRAC |
|---|---|---|---|---|---|---|---|
| $\mathcal{O}^{\text{sort}}_{\text{vol\_sum}}$ | 0.02±0.00 | 0.03±0.00 | 0.01±0.00 | 0.01±0.00 | **0.07±0.00** | 0.01±0.00 | 0.04±0.00 |
| $\mathcal{O}^{\text{sort}}_{\text{vol\_min}}$ | **0.24±0.00** | 0.20±0.01 | 0.12±0.02 | 0.14±0.00 | 0.21±0.01 | 0.05±0.02 | 0.18±0.02 |
| $\mathcal{O}^{\text{sort}}_{\text{vol\_max}}$ | 0.02±0.00 | 0.03±0.00 | 0.01±0.00 | 0.01±0.00 | **0.06±0.00** | 0.00±0.00 | 0.04±0.00 |
| $\mathcal{O}^{\text{sort}}_{\text{plain}}$ | 0.61±0.00 | 0.46±0.00 | 0.29±0.02 | 0.26±0.00 | **0.62±0.02** | 0.40±0.00 | 0.32±0.11 |
| $\mathcal{O}^{\text{std}}_{\text{vol\_sum}}$ | 0.00±0.00 | 0.01±0.00 | 0.00±0.00 | 0.00±0.00 | 0.02±0.00 | 0.00±0.00 | **0.03±0.01** |
| $\mathcal{O}^{\text{std}}_{\text{vol\_min}}$ | 0.03±0.00 | 0.09±0.00 | 0.04±0.01 | 0.05±0.00 | 0.08±0.01 | 0.02±0.01 | **0.11±0.03** |
| $\mathcal{O}^{\text{std}}_{\text{vol\_max}}$ | 0.00±0.00 | 0.01±0.00 | 0.00±0.00 | 0.00±0.00 | 0.01±0.00 | 0.00±0.00 | **0.02±0.01** |
| $\mathcal{O}^{\text{std}}_{\text{plain}}$ | 0.19±0.00 | 0.23±0.00 | 0.14±0.01 | 0.12±0.00 | **0.32±0.01** | 0.25±0.01 | 0.21±0.03 |
| $\mathcal{O}^{\text{naive}}_{\text{vol\_sum}}$ | 0.00±0.00 | 0.01±0.00 | 0.00±0.00 | 0.00±0.00 | 0.02±0.00 | 0.00±0.00 | **0.03±0.01** |
| $\mathcal{O}^{\text{naive}}_{\text{vol\_min}}$ | 0.02±0.00 | 0.08±0.00 | 0.04±0.01 | 0.05±0.00 | 0.08±0.01 | 0.02±0.01 | **0.11±0.04** |
| $\mathcal{O}^{\text{naive}}_{\text{vol\_max}}$ | 0.00±0.00 | 0.01±0.00 | 0.00±0.00 | 0.00±0.00 | 0.01±0.00 | 0.00±0.00 | **0.02±0.01** |
| $\mathcal{O}^{\text{naive}}_{\text{plain}}$ | 0.16±0.00 | 0.22±0.00 | 0.13±0.01 | 0.11±0.00 | **0.31±0.01** | 0.22±0.00 | 0.21±0.03 |
| size ratio | 8.500 | 3043.80 | 722.620 | 25.780 | **3059.20** | 415.880 | 203.230 |
| size std | **58.96** | 912.100 | 861.280 | 409.900 | 917.230 | 844.750 | 342.38 |

(acceptable), *Migration* into 5-7 clusters (acceptable), *WikiTalk* into 150-219 clusters (too many), and *Lead-Lag* into 10-55 clusters (acceptable or a bit too many). Leiden gives 12 (too many) clusters for *Blog*, 4-5 for *Telegram*, 5-6 for *Migration*, 170-248 (too many) for *WikiTalk*, and 10-55 clusters (acceptable or a bit too many) for *Lead-Lag*. OSLOM gives 6 clusters for *Blog* (too many), 16 for *Telegram* (too many), and 46 for *Migration* (too many). It could not generate results for *WikiTalk* after running for 12 hours, and hence we omit its discussion here. On *Lead-Lag*, OSLOM places every node in a single cluster for most of the years, and clusters the rest of the years into either a huge single cluster or two clusters.

None of the methods outperform DIGRAC on our chosen performance measures from Table 1 , except on the *Lead-Lag* data set (See Tables 14, 15, 16 and 17 for the other results). With regards to the 12 imbalance measures from Appendix Table 6, leaving out OSLOM as before, Louvain and Leiden perform poorly on all of the real data sets, except on *Lead-Lag*. Indeed, for *Lead-Lag*, the number of clusters we use for DIGRAC is ten according to the GICS sector memberships. However, if we use the sector memberships as labels, the imbalance values are poor, which implies that ten may not be a desirable choice of the number of clusters. Further, DIGRAC usually clusters the nodes into smaller number of clusters, while Louvain and Leiden usually cluster the nodes into a larger number of clusters (usually around 30, and sometimes above 50 clusters).

Finally, we provide more examples/explanations on why these density-based methods or even other methods that are based on random-walk should fail. We would mainly like to point out a family of illustrative examples demonstrating the subtle nuance concerning edge density.

Consider a meta graph with $K = 3$ nodes (clusters) A,B,C with directed edges AB, BC, CA, hence a directed cycle (our "cycle" DSBM models). Each pair of nodes $(v_i, v_j)$ in the graph of size n is connected by an edge independently with probability $p$ (which can even be equal to 1, in the case of a complete graph), hence the graph has the same density throughout. Now suppose we consider a pair of nodes $(v_i, v_j)$ such that $v_i$ belongs to cluster A, and $v_j$ to cluster B. Since this edge is part of the metagraph, with probability 1-eta, it is directed from $v_i$ to $v_j$, and with probability eta, $v_j$ sends an edge to $v_i$ (here, eta is the noise level parameter). Similar arguments can be made when $v_i$ (resp $v_j$) belongs to cluster B (resp C); and when $v_i$ (resp $v_j$) belongs to cluster C (resp A). See Figure 24 for an illustration. We also see that when the network is complete (see Figure 23 (g) and Table 9), InfoMap [14] fails empirically as it produces a single huge cluster. As a method based on random walks, this failure might occur as the chain could hardly be trapped inside a cluster as in the usual setting.

In such synthetic DSBM models with a "cycle" meta-graph structure, it can be shown that all nodes have the same in-degree and out-degree in expectation. Therefore, any density-based methods or modularity-based methods should fail. As the simplest possible example, one could just consider $K = 3$ clusters as above, without any noise (thus $\eta = 0$). InfoMap [14] tries to minimize the

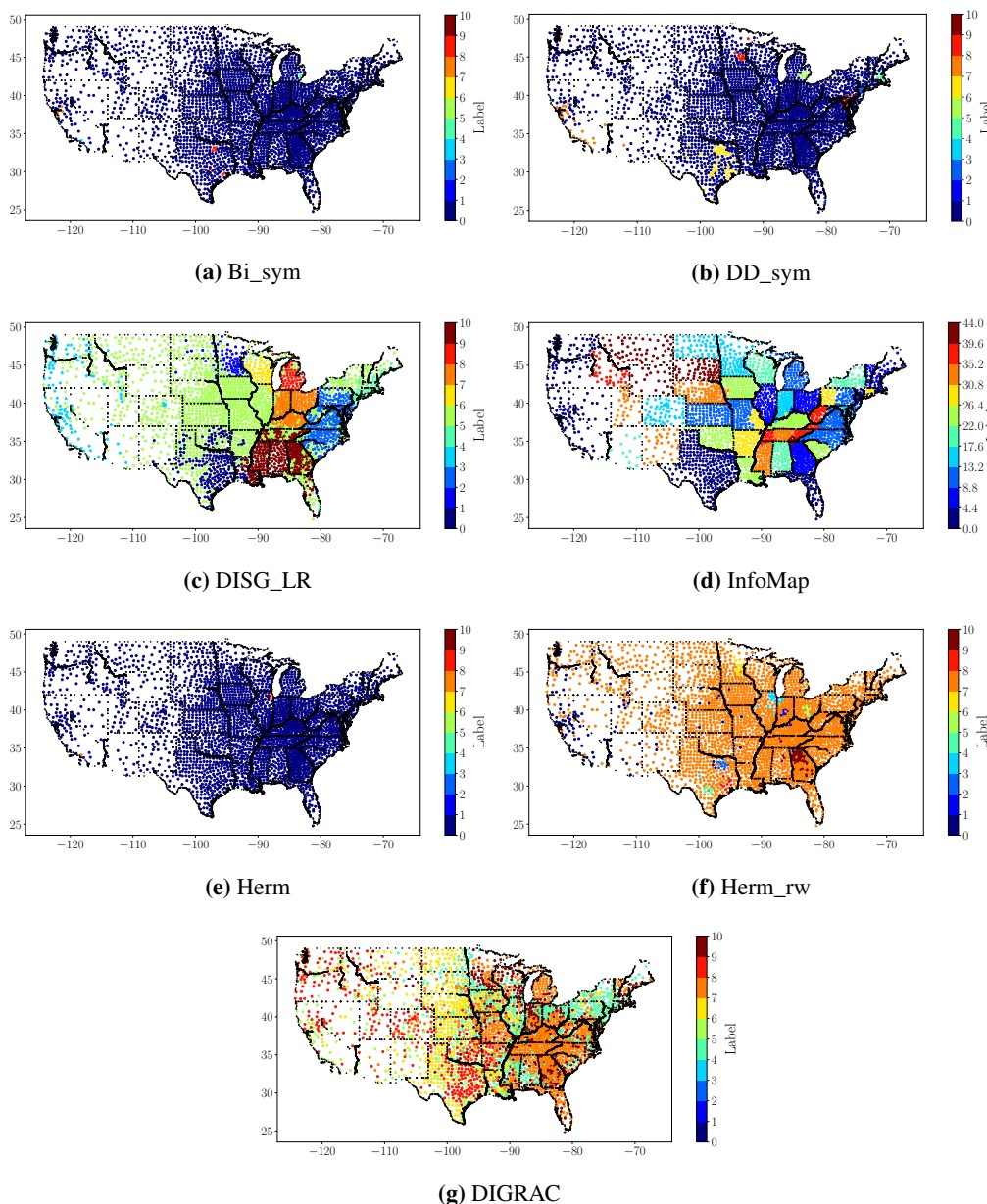

**Figure 22:** US migration predicted clusters, along with the geographic locations of the counties as well as state boundaries (in black). The input digraph has extremely large entries; unlike in Fig. 20, we do not employ here the normalization given by Eq. (13). Altogether, this demonstrates the robustness of DIGRAC to outliers in the data, which is not a characteristic of other state-of-the-art methods such as Herm and Herm_rw.

description length, but as no description length difference occurs in the ground-truth clustering structure for such "cycle" DSBMs, if we consider a brute-force optimization of the map equation. Indeed, for any method that is based on a random walk, the probability of the random walker going from one cluster to another is the same as staying within the cluster. Therefore, we could hardly optimize anything if we base our clustering structure on a random walker's visit frequencies/path lengths. Similarly, the Markov clustering algorithm [70] is based on the intuition that higher-length paths would be relatively more likely to stay within clusters – an assumption that is not warranted when there is no density difference. [15] and [16] are two interesting Markov aggregation algorithms

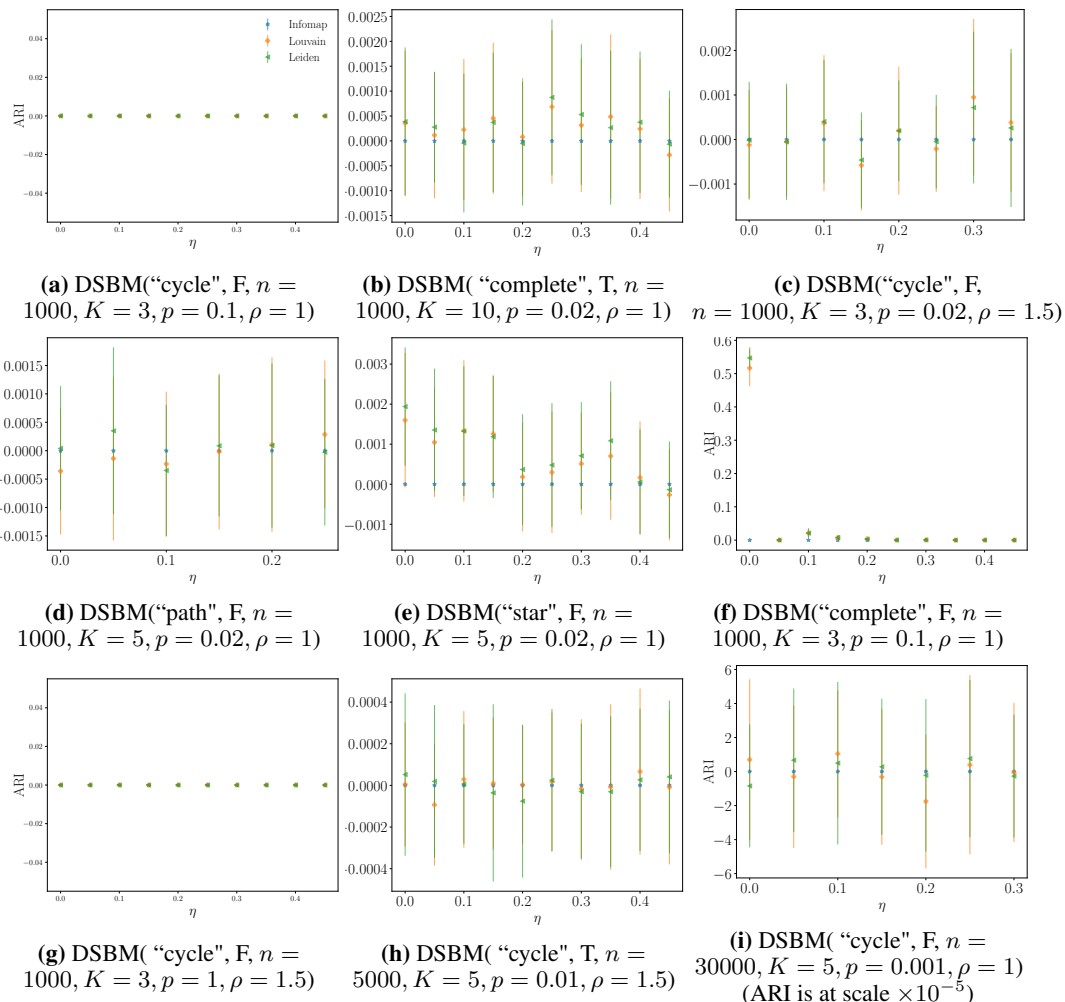

**(a)** DSBM("cycle", F, $n = 1000, K = 3, p = 0.1, \rho = 1$)

**(b)** DSBM( "complete", T, $n = 1000, K = 10, p = 0.02, \rho = 1$)

**(c)** DSBM("cycle", F, $n = 1000, K = 3, p = 0.02, \rho = 1.5$)

**(d)** DSBM("path", F, $n = 1000, K = 5, p = 0.02, \rho = 1$)

**(e)** DSBM("star", F, $n = 1000, K = 5, p = 0.02, \rho = 1$)

**(f)** DSBM("complete", F, $n = 1000, K = 3, p = 0.1, \rho = 1$)

**(g)** DSBM( "cycle", F, $n = 1000, K = 3, p = 1, \rho = 1.5$)

**(h)** DSBM( "cycle", T, $n = 5000, K = 5, p = 0.01, \rho = 1.5$)

**(i)** DSBM( "cycle", F, $n = 30000, K = 5, p = 0.001, \rho = 1$) (ARI is at scale $\times 10^{-5}$)

**Figure 23:** Test ARI comparison on synthetic data for Infomap, Louvain and Leiden. Error bars are given by one standard error.

based on information theory and automatic control ideas that might be able to cover the above example and may inspire some further comparison, but we omit comparison to them for now as we already have more than ten comparison methods and that InfoMap shares similar ideas to these two papers. As another example, as shown in [19], using belief propagation, in our model community structure should not be detectible (the right-hand side of (20) in [19] is zero for our "cycle" DSBMs). Therefore, at least methods that rely on belief propagation will fail on our benchmark models.

**Table 13:** Performance comparison on *Lead-Lag*, including Louvain and Leiden. Results in each year is averaged over ten runs. Mean and standard deviation (after $\pm$) are calculated over the 19 years. The best is marked in **bold red** and the second best is marked in underline blue. InfoMap results are omitted here as it usually predicts a single huge cluster and could not generate imbalance results. Louvain and Leiden yield essentially identical results and often attain the highest objectives, while DIGRAC almost always places either first or second across all methods considered.

| Metric/Method | Louvain/Leiden | Bi_sym | DD_sym | DISG_LR | Herm | Herm_rw | DIGRAC |
|---|---|---|---|---|---|---|---|
| $\mathcal{O}^{sort}_{vol\_sum}$ | 0.08±0.02 | 0.07±0.01 | 0.07±0.01 | 0.07±0.01 | 0.07±0.02 | 0.07±0.02 | **0.15±0.03** |
| $\mathcal{O}^{sort}_{vol\_min}$ | 0.15±0.04 | **0.51±0.10** | 0.48±0.09 | 0.47±0.10 | **0.51±0.11** | 0.50±0.10 | 0.47±0.09 |
| $\mathcal{O}^{sort}_{vol\_max}$ | 0.08±0.02 | 0.07±0.01 | 0.06±0.01 | 0.06±0.01 | 0.07±0.01 | 0.07±0.01 | **0.14±0.03** |
| $\mathcal{O}^{sort}_{plain}$ | 0.15±0.04 | **0.66±0.09** | 0.64±0.08 | 0.63±0.08 | **0.66±0.09** | 0.65±0.09 | 0.53±0.09 |
| $\mathcal{O}^{std}_{vol\_sum}$ | **0.23±0.06** | 0.04±0.01 | 0.04±0.01 | 0.04±0.01 | 0.04±0.01 | 0.04±0.01 | 0.12±0.03 |
| $\mathcal{O}^{std}_{vol\_min}$ | **0.46±0.11** | 0.27±0.04 | 0.27±0.04 | 0.25±0.04 | 0.27±0.03 | 0.27±0.03 | 0.38±0.07 |
| $\mathcal{O}^{std}_{vol\_max}$ | **0.23±0.05** | 0.04±0.00 | 0.03±0.00 | 0.03±0.00 | 0.03±0.00 | 0.03±0.00 | 0.11±0.02 |
| $\mathcal{O}^{std}_{plain}$ | **0.46±0.11** | 0.40±0.05 | 0.39±0.05 | 0.38±0.05 | 0.40±0.05 | 0.40±0.05 | 0.44±0.07 |
| $\mathcal{O}^{naive}_{vol\_sum}$ | **0.23±0.06** | 0.03±0.01 | 0.03±0.01 | 0.03±0.01 | 0.03±0.01 | 0.03±0.01 | 0.08±0.04 |
| $\mathcal{O}^{naive}_{vol\_min}$ | **0.46±0.11** | 0.20±0.05 | 0.19±0.05 | 0.18±0.05 | 0.19±0.04 | 0.19±0.04 | 0.26±0.10 |
| $\mathcal{O}^{naive}_{vol\_max}$ | **0.23±0.05** | 0.03±0.01 | 0.02±0.01 | 0.02±0.01 | 0.02±0.00 | 0.02±0.00 | 0.08±0.03 |
| $\mathcal{O}^{naive}_{plain}$ | **0.46±0.11** | 0.30±0.06 | 0.28±0.06 | 0.27±0.06 | 0.29±0.05 | 0.29±0.05 | 0.32±0.11 |
| size ratio | 124.530 | 3.67 | **3.34** | 3.900 | 4.110 | 3.880 | 8.070 |
| size std | 47.960 | 9.31 | **9.14** | 10.090 | 10.490 | 10.360 | 17.060 |

**Table 14:** Performance comparison on *Telegram*, including Louvain and Leiden. The best is marked in **bold red** and the second best is marked in underline blue.

| Metric/Method | Louvain/Leiden | InfoMap | Bi_sym | DD_sym | DISG_LR | Herm | Herm_rw | DIGRAC |
|---|---|---|---|---|---|---|---|---|
| $\mathcal{O}^{sort}_{vol\_sum}$ | 0.08±0.01 | 0.04±0.00 | 0.21±0.00 | 0.21±0.00 | 0.21±0.01 | 0.20±0.01 | 0.14±0.00 | **0.32±0.01** |
| $\mathcal{O}^{sort}_{vol\_min}$ | 0.39±0.07 | 0.47±0.00 | 0.67±0.00 | 0.61±0.00 | 0.66±0.02 | 0.66±0.02 | 0.19±0.00 | **0.79±0.06** |
| $\mathcal{O}^{sort}_{vol\_max}$ | 0.06±0.01 | 0.03±0.00 | 0.20±0.00 | 0.20±0.00 | 0.20±0.01 | 0.19±0.01 | 0.12±0.00 | **0.29±0.01** |
| $\mathcal{O}^{sort}_{plain}$ | 0.71±0.05 | **1.00±0.00** | 0.80±0.00 | 0.75±0.00 | 0.78±0.03 | 0.76±0.04 | 0.59±0.00 | 0.96±0.01 |
| $\mathcal{O}^{std}_{vol\_sum}$ | 0.07±0.01 | 0.01±0.00 | 0.26±0.00 | 0.26±0.00 | 0.26±0.01 | 0.25±0.02 | **0.35±0.00** | 0.28±0.01 |
| $\mathcal{O}^{std}_{vol\_min}$ | 0.33±0.08 | 0.16±0.00 | **0.84±0.00** | 0.76±0.00 | 0.82±0.03 | 0.82±0.03 | 0.49±0.00 | 0.73±0.03 |
| $\mathcal{O}^{std}_{vol\_max}$ | 0.05±0.01 | 0.01±0.00 | 0.25±0.00 | 0.25±0.00 | 0.25±0.01 | 0.24±0.02 | **0.29±0.00** | 0.25±0.01 |
| $\mathcal{O}^{std}_{plain}$ | 0.59±0.05 | 0.68±0.00 | **1.00±0.00** | 0.94±0.00 | 0.98±0.04 | 0.95±0.04 | 0.99±0.00 | 0.90±0.05 |
| $\mathcal{O}^{naive}_{vol\_sum}$ | 0.06±0.02 | 0.01±0.00 | 0.26±0.00 | 0.26±0.00 | 0.26±0.01 | 0.25±0.02 | 0.23±0.00 | **0.27±0.01** |
| $\mathcal{O}^{naive}_{vol\_min}$ | 0.28±0.11 | 0.11±0.00 | **0.84±0.00** | 0.76±0.00 | 0.82±0.03 | 0.82±0.03 | 0.32±0.00 | 0.72±0.04 |
| $\mathcal{O}^{naive}_{vol\_max}$ | 0.04±0.01 | 0.00±0.00 | **0.25±0.00** | **0.25±0.00** | **0.25±0.01** | 0.24±0.02 | 0.20±0.00 | 0.24±0.01 |
| $\mathcal{O}^{naive}_{plain}$ | 0.56±0.01 | 0.63±0.00 | **1.00±0.00** | 0.94±0.00 | 0.98±0.04 | 0.95±0.04 | 0.99±0.00 | 0.89±0.06 |

**Table 15:** Performance comparison on *Blog*, including Louvain and Leiden. The best is marked in **bold red** and the second best is marked in underline blue.

| Metric/Method | Louvain/Leiden | InfoMap | Bi_sym | DD_sym | DISG_LR | Herm | Herm_rw | DIGRAC |
|---|---|---|---|---|---|---|---|---|
| $\mathcal{O}^{sort}_{vol\_sum}$ | 0.00±0.00 | 0.07±0.00 | 0.07±0.00 | 0.00±0.00 | 0.05±0.00 | 0.37±0.00 | 0.00±0.00 | **0.44±0.00** |
| $\mathcal{O}^{sort}_{vol\_min}$ | 0.01±0.01 | 0.02±0.00 | 0.33±0.00 | 0.05±0.00 | 0.31±0.00 | 0.78±0.01 | **0.89±0.00** | 0.76±0.00 |
| $\mathcal{O}^{sort}_{vol\_max}$ | 0.01±0.01 | 0.05±0.00 | 0.05±0.00 | 0.00±0.00 | 0.04±0.00 | 0.26±0.00 | 0.00±0.00 | **0.40±0.00** |
| $\mathcal{O}^{sort}_{plain}$ | **1.00±0.00** | **1.00±0.00** | 0.33±0.00 | 0.05±0.00 | 0.31±0.00 | 0.78±0.01 | 0.89±0.00 | 0.76±0.00 |
| $\mathcal{O}^{std}_{vol\_sum}$ | 0.00±0.00 | 0.00±0.00 | 0.07±0.00 | 0.00±0.00 | 0.05±0.00 | 0.37±0.00 | 0.00±0.00 | **0.44±0.00** |
| $\mathcal{O}^{std}_{vol\_min}$ | 0.00±0.00 | 0.00±0.00 | 0.33±0.00 | 0.05±0.00 | 0.31±0.00 | 0.78±0.01 | **0.89±0.00** | 0.76±0.00 |
| $\mathcal{O}^{std}_{vol\_max}$ | 0.00±0.00 | 0.00±0.00 | 0.05±0.00 | 0.00±0.00 | 0.04±0.00 | 0.26±0.00 | 0.00±0.00 | **0.40±0.00** |
| $\mathcal{O}^{std}_{plain}$ | 0.56±0.13 | 0.73±0.00 | 0.33±0.00 | 0.05±0.00 | 0.31±0.00 | 0.78±0.01 | **0.89±0.00** | 0.76±0.00 |
| $\mathcal{O}^{naive}_{vol\_sum}$ | 0.00±0.00 | 0.00±0.00 | 0.07±0.00 | 0.00±0.00 | 0.05±0.00 | 0.37±0.00 | 0.00±0.00 | **0.44±0.00** |
| $\mathcal{O}^{naive}_{vol\_min}$ | 0.00±0.00 | 0.00±0.00 | 0.33±0.00 | 0.05±0.00 | 0.31±0.00 | 0.78±0.01 | **0.89±0.00** | 0.76±0.00 |
| $\mathcal{O}^{naive}_{vol\_max}$ | 0.00±0.00 | 0.00±0.00 | 0.05±0.00 | 0.00±0.00 | 0.04±0.00 | 0.26±0.00 | 0.00±0.00 | **0.40±0.00** |
| $\mathcal{O}^{naive}_{plain}$ | 0.76±0.00 | 0.76±0.00 | 0.33±0.00 | 0.05±0.00 | 0.31±0.00 | 0.78±0.01 | **0.89±0.00** | 0.76±0.00 |

**Table 16:** Performance comparison on *Migration*, including Louvain and Leiden. The best is marked in **bold red** and the second best is marked in underline blue. InfoMap results are omitted here as it predicts a single huge cluster and could not generate imbalance results.

| Metric/Method | Louvain/Leiden | Bi_sym | DD_sym | DISG_LR | Herm | Herm_rw | DIGRAC |
|---|---|---|---|---|---|---|---|
| $\mathcal{O}^{\text{sort}}_{\text{vol\_sum}}$ | 0.01±0.00 | 0.03±0.00 | 0.01±0.00 | 0.02±0.00 | 0.04±0.00 | 0.02±0.00 | **0.05±0.00** |
| $\mathcal{O}^{\text{sort}}_{\text{vol\_min}}$ | 0.05±0.01 | **0.19±0.00** | 0.08±0.00 | 0.08±0.00 | 0.15±0.02 | 0.05±0.00 | 0.18±0.03 |
| $\mathcal{O}^{\text{sort}}_{\text{vol\_max}}$ | 0.01±0.00 | 0.03±0.00 | 0.01±0.00 | 0.01±0.00 | 0.03±0.00 | 0.02±0.00 | **0.04±0.00** |
| $\mathcal{O}^{\text{sort}}_{\text{plain}}$ | 0.09±0.02 | 0.24±0.00 | 0.20±0.00 | 0.17±0.00 | 0.40±0.01 | **0.49±0.06** | 0.29±0.04 |
| $\mathcal{O}^{\text{std}}_{\text{vol\_sum}}$ | 0.00±0.00 | 0.01±0.00 | 0.01±0.00 | 0.01±0.00 | 0.02±0.00 | 0.02±0.00 | **0.04±0.01** |
| $\mathcal{O}^{\text{std}}_{\text{vol\_min}}$ | 0.04±0.01 | 0.10±0.00 | 0.05±0.00 | 0.05±0.00 | 0.08±0.01 | 0.04±0.00 | **0.16±0.03** |
| $\mathcal{O}^{\text{std}}_{\text{vol\_max}}$ | 0.00±0.00 | 0.01±0.00 | 0.01±0.00 | 0.01±0.00 | 0.01±0.00 | 0.01±0.00 | **0.03±0.01** |
| $\mathcal{O}^{\text{std}}_{\text{plain}}$ | 0.07±0.01 | 0.13±0.00 | 0.12±0.00 | 0.11±0.00 | 0.20±0.01 | 0.20±0.01 | **0.26±0.01** |
| $\mathcal{O}^{\text{naive}}_{\text{vol\_sum}}$ | 0.00±0.00 | 0.01±0.00 | 0.01±0.00 | 0.01±0.00 | 0.02±0.00 | 0.01±0.00 | **0.04±0.01** |
| $\mathcal{O}^{\text{naive}}_{\text{vol\_min}}$ | 0.04±0.01 | 0.09±0.00 | 0.04±0.00 | 0.04±0.00 | 0.08±0.01 | 0.01±0.00 | **0.16±0.03** |
| $\mathcal{O}^{\text{naive}}_{\text{vol\_max}}$ | 0.00±0.00 | 0.01±0.00 | 0.00±0.00 | 0.01±0.00 | 0.01±0.00 | 0.00±0.00 | **0.03±0.01** |
| $\mathcal{O}^{\text{naive}}_{\text{plain}}$ | 0.07±0.00 | 0.12±0.00 | 0.10±0.00 | 0.08±0.00 | 0.19±0.00 | 0.19±0.03 | **0.26±0.01** |

**Table 17:** Performance comparison on *WikiTalk*, including Louvain and Leiden. The best is marked in **bold red** and the second best is marked in underline blue. InfoMap results are omitted here as its large number of predicted clusters leads to memory error in imbalance calculation.

| Metric/Method | Louvain/Leiden | DISG_LR | Herm | Herm_rw | DIGRAC |
|---|---|---|---|---|---|
| $\mathcal{O}^{\text{sort}}_{\text{vol\_sum}}$ | 0.01±0.00 | 0.18±0.03 | 0.15±0.02 | 0.00±0.00 | **0.24±0.05** |
| $\mathcal{O}^{\text{sort}}_{\text{vol\_min}}$ | 0.15±0.00 | 0.10±0.03 | 0.22±0.05 | 0.26±0.00 | **0.28±0.13** |
| $\mathcal{O}^{\text{sort}}_{\text{vol\_max}}$ | 0.01±0.00 | 0.16±0.03 | 0.09±0.01 | 0.00±0.00 | **0.19±0.04** |
| $\mathcal{O}^{\text{sort}}_{\text{plain}}$ | **1.00±0.00** | 0.87±0.08 | 0.99±0.01 | 0.98±0.00 | **1.00±0.00** |
| $\mathcal{O}^{\text{std}}_{\text{vol\_sum}}$ | 0.00±0.00 | **0.17±0.04** | 0.06±0.01 | 0.01±0.00 | 0.14±0.02 |
| $\mathcal{O}^{\text{std}}_{\text{vol\_min}}$ | 0.01±0.00 | 0.09±0.02 | 0.09±0.02 | **0.27±0.00** | 0.18±0.08 |
| $\mathcal{O}^{\text{std}}_{\text{vol\_max}}$ | 0.00±0.00 | **0.15±0.04** | 0.04±0.00 | 0.00±0.00 | 0.11±0.02 |
| $\mathcal{O}^{\text{std}}_{\text{plain}}$ | 0.42±0.00 | 0.72±0.03 | 0.70±0.05 | **0.98±0.00** | 0.84±0.06 |
| $\mathcal{O}^{\text{naive}}_{\text{vol\_sum}}$ | 0.00±0.00 | 0.10±0.02 | 0.04±0.00 | 0.00±0.00 | **0.12±0.01** |
| $\mathcal{O}^{\text{naive}}_{\text{vol\_min}}$ | 0.01±0.00 | 0.06±0.03 | 0.07±0.02 | **0.26±0.00** | 0.15±0.07 |
| $\mathcal{O}^{\text{naive}}_{\text{vol\_max}}$ | 0.00±0.00 | **0.09±0.02** | 0.03±0.00 | 0.00±0.00 | **0.09±0.01** |
| $\mathcal{O}^{\text{naive}}_{\text{plain}}$ | 0.43±0.00 | 0.64±0.04 | 0.61±0.04 | **0.98±0.00** | 0.76±0.06 |

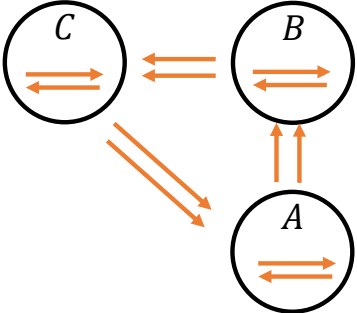

**Figure 24:** An example of a "cycle" meta-graph.

