# OpenReview forum: "DIGRAC:  Digraph Clustering Based on  Flow Imbalance"
_logconference.io/LOG/2022/Conference — LoG 2022 Poster_

### Official Review · Reviewer_a8JJ · 2022-10-12

**Overall Score:** 6
**Confidence:** 5

**Review:**

The authors propose a self-supervised approach for community detection in directed graphs that is based on neural graph embeddings. More specifically, the graph embedding is jointly learned with soft cluster/community assignments, with the aim to maximize the "flow imbalance" between different clusters.

The paper treats the interesting and relevant topic of community detection/graph clustering and proposes not only a GNN-based approach to it, but also a novel cost function that is inspired by random walk-based approaches. The experimental evidence suggests that DIGRAC can outperform the state-of-the-art on synthetic and real-world datasets. Another strength of the paper is the ablation study, which shows that flow imbalance is a valid self-supervision objective also for GNN architecture different from the one used in the paper. The discussion of the related work is extensive and exemplary.

Coming to the weaknesses, I had the feeling that the paper is not clear in all its statements and that, thus, a thorough evaluation of the proposed approach is difficult for the reader. Further, while the discussion of the related work is extensive, I had the feeling that a large part of flow-based community detection methods have been neglected. Finally, the experimental setup is not fully clear, which makes the results a bit less convincing that they could be. I hope that the following questions and comments may help:

### Related Work on Flow-Based Community Detection
The authors argue that their proposed flow imbalance measure is a novelty. While indeed I have not seen the exact formulations that were proposed by the authors, flow- or random walk-based community detection has been around for some time (incl. approaches such as Infomap, Walktrap, Synwalk, etc.). As far as I understand, all random walk-based approaches to community detection are capable for clustering directed graphs. Regarding the "meta-graph structures, which are otherwise not detectable by existing methods", I would like to point out that in addition to the mentioned random walk-based approaches, also approaches to Markov chain aggregation provide such meta-graph structures. I suggest to include the corresponding related work and, preferably, also compare to it (e.g., including Walktrap).

In line 90, it is stated that Infomap "still relies on some amount of density-based signal". Please explain why this is the case.

In the paragraph starting at line 118, the authors cover methods that are not used in the comparisons. Some of the arguments are questionable, however. For example, the authors state that OSLOM and the Leiden algorithm are based on a density heuristic and modularity, respectively, and that thus a comparison is unfair. I tend to disagree here. One of the main statements of the paper is that flow imbalance is a valid signal for community detection. To test this hypothesis, the performance of flow imbalance-based approaches should be compared with other approaches, including modularity- and density-based ones. If at all, at this stage one can exclude methods that are not supposed to work for directed graphs, as here a comparison would indeed be unfair.

### Experimental Setup and Results
- Connecting to the last paragraphs of Sec. 3.1: Since the soft assignments $\mathbf{P}$ are optimized during training, does that mean that $\mathcal{T}(\beta)$ changes during training? As a consequence, also $\mathcal{O}^\mathrm{sort}$ changes, which may lead to instabilities during training. If so, how is this accounted for?
- For the 10 methods introduced in the paragraph starting at line 273, it would be good to mention which of these approaches is based on GNNs, which requires providing $K$ as input, and which is designed to work with directed graphs.
- In line 280, the authors write that they use "the same hyperparameter settings stated in these papers" -- what does "these" refer to?
- In Table 1, it is shown that DIGRAC achieves the best performance in terms of $\mathcal{O}$. This is not surprising, however. Indeed, $\mathcal{O}$ is the objective of DIGRAC, therefore it would be surprising if a different method achieves better results. Instead, I would propose to use alternative measures for cluster validity if no ground truth is available.
- In Figure 3, why does Infomap perform so poorly? Does Infomap respond with a single cluster?
- In line 323-324, the number of clusters $K$ and the value $\beta$ for DIGRAC are set to specific values for each dataset. Appendix C.3 shows that these values are taken from the literature, which is fair. Which of the competing method also takes these specific values as inputs/hyperparameters? Or do all the competing methods determine the number of clusters $K$ automatically?
- In Figure 4 it is not clear what CI, LICE, and CE refer to. Does CI refer to self-supervised DiGCN, LICE to self- and supervised DiGCN, and CE to supervised DiGCN? Please clarify.
- Finally, there is a discrepancy between Figures 3e and 4a, and Figures 3f and 4b. Indeed, comparing the cyan triangles in Figure 3e with the orange markers (CE) in Figure 4a, one sees that while in Figure 3e the ARI increases with $\eta$ increasing from 0.2 to 0.25, we do not see such an increase in Figure 4a. Can you explain this?

### Further Comments
- I suggest to mention in the abstract and early in the introduction that the proposed approach is based on GNNs (this is not mentioned until line 43).
- Around eq. (2), I would appreciate a brief discussion about how the imbalance term $CI$ behaves for trivial cluster assignments, i.e., when $K=1$ or when $K=n$.
- In line 189, what does "noisy" refer to? Noisy attributes, or missing edges?
- In line 200, I assume $w_{ij}=w_{ij}$ should be replaced by $w_{ij}=w_{ji}$, right?
- In line 337, the loss should be $\mathcal{L}_{volsum}^{sort}$, right?

*EDIT* Update to Weak Accept after discussion period.

---

### Official Review · Reviewer_RnzU · 2022-10-21

**Overall Score:** 6
**Confidence:** 4

**Review:**

Summary:

The paper presents DIGRAC, an algorithm for clustering directed graphs.  The algorithm has several advantages such as not requiring node labels and its ability to compute node embeddings.  To accomplish this, DIGRAC introduces a directed flow imbalance measure and a graph neural network (GNN) architecture to encode the graph and assign cluster probabilities to nodes, allowing end-to-end optimization of the imbalance measure.  Performance is evaluated on Directed Stochastic Block Model (DSMB) and real-world graphs.  The method compares favorably with others on both the synthetic and real-world data, although no ground truth labels exist for the real-world graphs so the authors devise their own measures of success.  The stated contributions of the paper are (1) a new GNN framework for end-to-end clustering based on a specific objective, (2) a new self-supervised loss function measuring flow imbalance, (3) an unsupervised method extensible to a semi-supervised setting where labels are available.

Overall I believe the work represents a useful contribution; it is appropriate for the conference as it frames a new task on graphs and opens new avenues for graph research.  The shortcomings of the paper (lack of ground truth in real-world data) are relatively minor given the thorough benchmarking on synthetic data. If possible I believe it would help the paper to try for evaluation against some kind of ground truth cluster labels.

Recommendation: Weak accept.


Strong Points:
* The paper is well-written and clear and includes a thorough review of prior work. The benchmarks on DSMB are clear and appropriate.
* The paper introduces a novel algorithm for encoding directed graphs and clustering.  The end-to-end nature of embedding generation and clustering is useful for applications and seems to perform well.

Weak Points:
* The biggest weakness in the paper is the lack of real-world data.  The authors claim that real-world datasets with ground truth flow imbalances are not available, which confuses me given the stated importance of the problem.  The authors get around this issue by creating their own measures of optimality and evaluate against those; however, these  measures are similar to DIGRAC’s loss function and would seem to favor DIGRAC.  There are additional measures of optimality in the appendix but these are also confusing as they don’t uniformly favor DIGRAC.  Is there a way to utilize ground truth cluster labels to evaluate the method with these data or other datasets?  For example if the stocks in the lead/lag dataset were labeled by industry would DIGRAC perform well against these labels as measured by ARI?
* More theoretical explanation of the outperformance of DIGRAC would be helpful.  I’m not sure I understand why it performs well on the DSMB benchmarks and would like to understand this better.


Questions:
1. How does ARI look on real-world data if we do our best to construct some kind of real-world labels?
1. What insights were we able to derive from the results?  The stated example of the California to Arizona migration pattern was interesting to me.
1. Why not include the comparison to Louvain and Leiden as mentioned on page 3?

---

### Official Review · Reviewer_noPt · 2022-10-21

**Overall Score:** 5
**Confidence:** 2

**Review:**

In this paper, the authors proposed a GNN for node clustering in directed graphs based on flow Imbalance and evaluated its effectiveness on various datasets.   I have some major concerns.

1.  The paper is not self-contained and hard to follow. Some important concepts are used without being clearly described. For example, Meta-graph and meta-node should be formally defined. "directed core-periphery" structure in the introduction is used without explanations.
2.  The setting is similar to K-means. So It is more like an unsupervised learning method instead of a self-supervised learning method.
3. NMI is the most commonly used metric for clustering. In section 4, the shortcomings of NMI are not explicitly described. The reference provides an improved version of the NMI. Why not this one?   Besides, I could find NMI results in the appendix.
4. Some state-of-the-art clustering algorithms are missing.  For example:
[1] Bo, Deyu, et al. "Structural deep clustering network." Proceedings of The Web Conference 2020. 2020.
[2] Zhang, Xiaotong, et al. "Spectral embedding network for attributed graph clustering." Neural Networks 142 (2021): 388-396.
[3] Liu, Yue, et al. "Deep Graph Clustering via Dual Correlation Reduction." Proc. of AAAI. 2022.
[4] Xia, Wei, et al. "Self-supervised Contrastive Attributed Graph Clustering." arXiv preprint arXiv:2110.08264 (2021).
Since GNN can naturally handle directed graphs, these graph clustering methods should be cited and compared.

---

### Official Review · Reviewer_j4qv · 2022-10-22

**Overall Score:** 5
**Confidence:** 3

**Review:**


### Summary

This paper proposes a GNN framework for node clustering on directed graphs based on the directed flow imbalance measure.
The proposed framework, DIGRAC uses a self-supervised imbalance objective for node clustering which is designed to maximize the cut imbalance over clusters.



### Strengths & Weakness


1. The proposed framework is somewhat intuitive. Also, empirical studies were extensively conducted under a wide range of settings (sparsity, topologies). However, there are many statements/terminologies without sufficient explanation which make it difficult for readers to follow.

2. Although this method is the first work to maximize the flow imbalance using GNNs, there are already works using flow imbalance (directionality) for node clustering, so the idea is not entirely fresh.

3. The performance improvement of the proposed methodology seems marginal in many cases.

**Ethical Concerns:**

Yes

---

### Meta-Review · Area_Chair_cPiC · 2022-11-19

**Confidence:** 4
**Recommendation:** Borderline and needs further discussi…

**Meta Review:**

This paper studies self-supervised community detection for directed graphs, by considering GNN architectures trained using a directed flow imbalance objective. Empirical results across a variety of synthetic and real-world datasets show a systematic improvement over existing approaches.
Reviewers were generally torn on this submission. On one hand, they expressed concerns about the relevance of the empirical evaluation on real datasets, mostly due to the lack of ground truth measure of success. Additionally, reviewers also raised some doubts about novelty, with regards to existing approaches using flow imbalance for clustering. On the other hand, reviewers also acknowledged the importance of this topic, and the diligent work by the authors in producing an exhaustive set of baseline comparisons. All in all, this is a borderline submission, where the authors' ability to fix and address the clarification remarks raised during rebuttal will ultimately outweight the existing limitations.

---

### Decision · Program_Chairs · 2022-11-22

Accept (Poster)